# Deterministic positioning and alignment of a single-molecule exciton in plasmonic nanodimer for strong coupling

Renming Liu [1,2,7] ✉, Ming Geng[1,7], Jindong Ai[1,7], Xinyi Fan[1], Zhixiang Liu[1], Yu-Wei Lu[3], Yanmin Kuang[1], Jing-Feng Liu [4] ✉, Lijun Guo [1] ✉ & Lin Wu [5,6] ✉

Experimental realization of strong coupling between a single exciton and plasmons remains challenging as it requires deterministic positioning of the single exciton and alignment of its dipole moment with the plasmonic fields. This study aims to combine the host–guest chemistry approach with the cucurbit[7]uril-mediated active self-assembly to precisely integrate a single methylene blue molecule in an Au nanodimer at the deterministic position (gap center of the nanodimer) with the maximum electric field ($EF_{max}$) and perfectly align its transition dipole moment with the $EF_{max}$, yielding a large spectral Rabi splitting of 116 meV for a single-molecule exciton—matching the analytical model and numerical simulations. Statistical analysis of vibrational spectroscopy and dark-field scattering spectra confirm the realization of the single exciton strong coupling at room temperature. Our work may suggest an approach for achieving the strong coupling between a deterministic single exciton and plasmons, contributing to the development of room-temperature single-qubit quantum devices.

Strong coupling of a deterministic single exciton to electromagnetic modes at ambient conditions is highly desirable for exploiting room-temperature single-qubit quantum devices[1,2]. To achieve such an exciting goal, recent studies[3–6] have proposed a promising research direction for developing plasmonic nanocavities with an ultrasmall mode volume below 40 $nm^{3}$[7], broad tuning bandwidth, and room-temperature operations. In the past two decades, strong coupling between plasmons and a variety of emitters has been extensively investigated, as a result of which the number of emitters involved in the strong coupling has been dramatically reduced from the hundreds of thousands[8–12] to only a few[13–16] and even the single-emitter level in recent studies[7,17,18].

Nevertheless, strong coupling between a deterministic single exciton and the plasmons has yet to be realized.

As theoretically predicted[19–22], such strong coupling can be achieved by precisely integrating a single quantum emitter into a plasmonic nanocavity at a deterministic position with the maximum electric fields ($EF_{max}$) and perfectly aligning the excitonic transition dipole moment ($\mu_c$) with the field. However, simultaneous experimental implementation of these two conditions is challenging. For instance, we previously employed a cuboid Au (core)-Ag (shell) nanorod coated with monolayer dye molecule J-aggregates to study plexcitonic strong coupling[18], with uncertainties about the position and $\mu_c$ direction of the delocalized excitons in J-aggregates. In our

[1]School of Physics and Electronics, International Joint Research Laboratory of New Energy Materials and Devices of Henan Province, Henan University, Kaifeng 475004, China. [2]Institute of Quantum Materials and Physics, Henan Academy of Sciences, Zhengzhou 450046, China. [3]Quantum Science Center of Guangdong–Hong Kong–Macao Greater Bay Area (Guangdong), Shenzhen 518045, China. [4]College of Electronic Engineering, South China Agricultural University, Guangzhou 510642, China. [5]Department of Science, Mathematics and Technology, Singapore University of Technology and Design, 8 Somapah Road, Singapore 487372, Republic of Singapore. [6]Institute of High Performance Computing, Agency for Science, Technology, and Research (A*STAR), 1 Fusionopolis Way, No. 16-16 Connexis, Singapore 138632, Republic of Singapore. [7]These authors contributed equally: Renming Liu, Ming Geng, Jindong Ai. ✉e-mail: liurm@henu.edu.cn; liujingfeng@scau.edu.cn; juneguo@henu.edu.cn; lin_wu@sutd.edu.sg

hybrid systems, the cuboid Au (core)-Ag (shell) nanorod squeezed the plasmonic fields around its sharp corners into a region down to approximately 1 nm, which involved only a few excitons close to the sharp corners in the strong coupling. Even though we have recently further reduced the number of strongly-coupled excitons with plasmons to one and significantly improved its success rate using the hybrid plasmonic-photonic nanocavities, the exciton position and dipole direction remain uncertain in such a coupling system[23]. Similar cases are also observed in other plexcitonic systems, which employ two-dimensional (2D) transition-metal dichalcogenide monolayers as the exciton materials (Supplementary Table 1)[13,14]. However, the process for evaluating the number of excitons involved in the strong coupling based on such 2D excitonic materials remains debatable[24].

Efforts have been made toward achieving the deterministic positioning of a single quantum emitter. In 2016, Santhosh et al. achieved the strong coupling of the single-quantum dot (QD) to the gap plasmon mode, which is supported by a silver nanobowtie cavity containing a single QD at a relatively deterministic position in the nanogap, but whose $\mu_c$ direction cannot be well controlled to align with the mode fields[17]. Subsequently, more research groups have also demonstrated a similar strong coupling between the localized plasmon resonances and single QDs[25–30]. The main advantage for these works achieving the plexcitonic strong coupling in single QDs or J-aggregate chains lies in the large $\mu_c$ (0.5–1.0 e nm) of the QD or J-aggregate excitons, which can be considered as a collective effect of the presence of multiple excitons[31–33]. Therefore, it is uncertain that only one exciton is involved in the strong coupling in these systems. In

fact, due to the small $\mu_c$ of a single exciton ( < 0.1 e nm)[7,34] and serious damping of plasmons, realizing strong coupling between a deterministic single exciton and the plasmon mode remains elusive. Breakthrough work exploiting the single-exciton strong coupling with plasmons was first performed by Chikkaraddy et al. in the nanoparticle-on-mirror (NPoM) geometry[7]. The study successfully placed single methylene blue (MB) molecules in the gap of the NPoM and perfectly aligned the $\mu_c$ of the molecule along with the plasmonic fields using host–guest chemistry, while the number and location of the embedded molecules in the NPoM's gap remained random[7]. Taking advantage of DNA origami, Chikkaraddy et al. further deterministically positioned a single-Cy5 molecule at the center of the NPoM nanocavity; however, the $\mu_c$ of the molecule could not be deterministically aligned with the plasmonic fields, and the fit average dipole orientation was approximately 35° with respect to the gap plasmonic mode[35]. As the result of fine control either in the dipole position or in the dipole orientation of the single molecule, the authors observed spectral Rabi splitting (SRS) of approximately 60–90 meV, approaching the single-exciton strong coupling regime. At this juncture, the critical step to moving this field towards technological impact lies in the simultaneous achievement of deterministic molecule positioning and perfect dipole alignment with the mode field.

In this work, we propose a self-assembled nanodimer configuration constructed using the methods of host-guest chemistry and cucurbit[7]uril (CB[7])-mediated active self-assembly, which enable us precisely integrate a single methylene-blue (MB) molecule in an Au nanodimer (AuND) at the deterministic position (gap center) with the

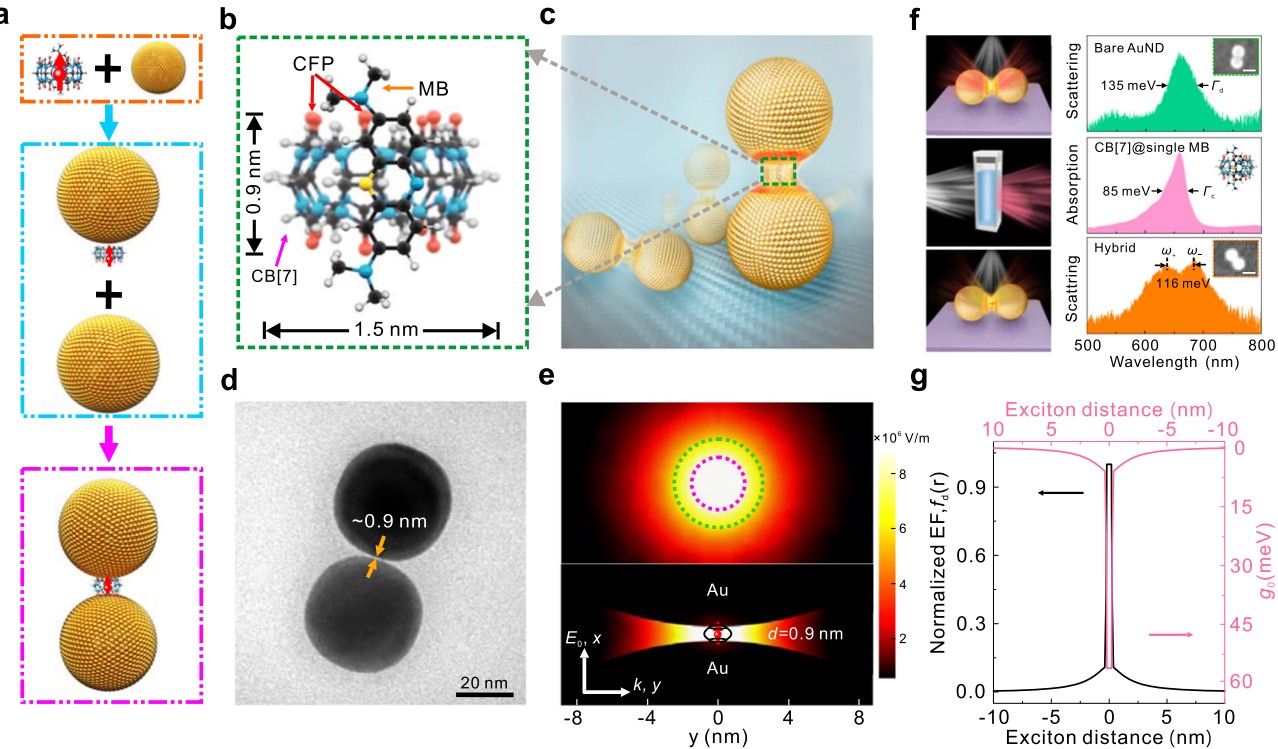

**Fig. 1 | Precise positioning and aligning of a single MB molecule in AuND for strong coupling. a** The process flow to fabricate the hybrid AuND containing a single MB molecule with deterministic position and dipole alignment. **b, c** Schematics of (**b**) a single MB molecule embedded in a CB[7] cage and (**c**) the single MB molecule captured in the gap center of the AuND via the CB[7] cage. CFP: carbonyl-fringed portals. **d** TEM image of an AuND/CB[7]@single MB molecule. **e** Simulated electric field (EF) distribution of the gap plasmon confined in the AuND with a 0.9-nm gap. The upper panel: in-plane vertical EF distribution in the center of the gap, with the 3 and 5 nm radius marked by dashed red and green circles, respectively. The lower panel: In-plane parallel EF distribution in the middle of the

gap. **f** Optical measurements. Left panel: Schematics of the optical measurements for a bare AuND, CB[7]@single MB emitters in water, and a hybrid AuND/CB[7]@single MB, repectively. Right panel: The upper and bottom curves are scattering spectra related to the measurements in the left panel, the insets show SEM images of the measured samples, and the scale bar is 50 nm; the middle curve is the absorption spectrum of CB[7]@single MB emitters in water. The molar ratio of MB and CB[7] was 1:2. **g** Normalized EF and calculated $g_0$ as a function of the exciton distance away from the gap center along the dimer direction (x-axis). In calculations, $\mu_c = 0.09$ e nm, $d = 0.9$ nm.

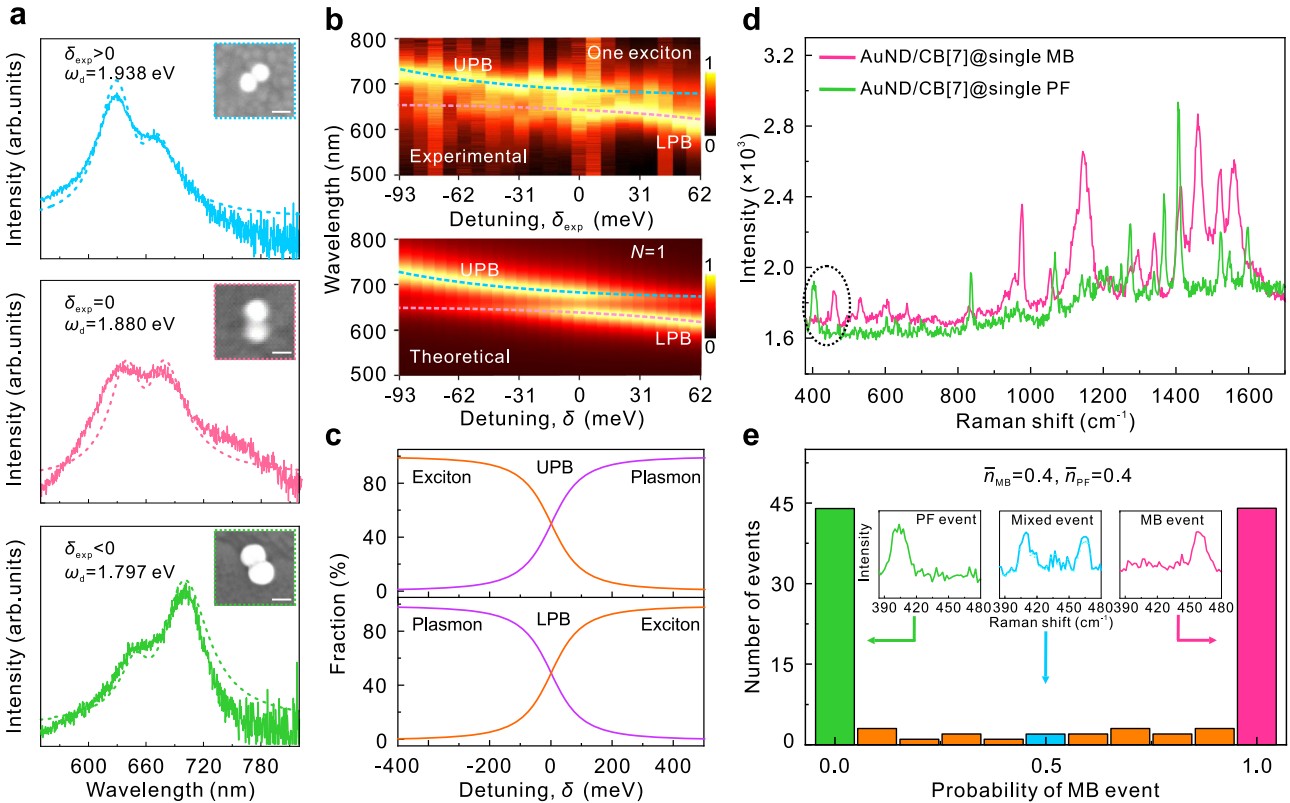

**Fig. 2 | Single-exciton strong coupling in individual AuND/CB[7]@single MB hybrids with different detunings. a** Scattering spectra of three individual AuND/CB[7]@single MB hybrids with different experimental detunings ($\delta_{exp}$>0 (upper), $\delta_{exp}$ = 0 (middle), and $\delta_{exp}$<0 (bottom)). The dashed curves show the theoretical spectra of the strongly coupled systems calculated using Eq. (3). The insets show SEM images of the corresponding measured samples. The scale bar is 50 nm. **b** Upper panel: Normalized experimental scattering spectra of the individual AuND/CB[7]@single MB hybrids arranged by the size of the experimental detuning ($\delta_{exp}$). Lower panel: Normalized theoretical spectra calculated for $N$ = 1 using Eq. (3), arranged by the size of the theoretical detuning ($\delta$). **c** The relative mixing fractions of the plasmon mode and exciton in the upper plexciton branch (UPB) and the lower plexciton branch (LPB) as a function of detuning. In calculations, $\hbar\omega_c$ = 1.880 eV, $g_{dc}$= 56.6 meV, $\bar{\Gamma}_d$ = 135 meV (Supplementary Fig. 12), and $\Gamma_c$ = 85 meV. **d** Representative SERS spectra of MB and PF molecules collected in single hybrid AuNDs with an integration time of 2 s and excitation at 633 nm. **e** Histograms of the relative contribution of MB dye $p_{MB} = I_{MB}/(I_{MB} + I_{PF})$ to the total intensity of MB and PF, demonstrating the fraction of the signal to which MB contributes. The insets show the signature of three different probabilities. Each spectrum was collected for 2 s, and the samples were excited at 633 nm. MB: methylene-blue, PF: proflavine.

$EF_{max}$ and perfectly align its transition dipole moment with the gap plasmon (along the electric field, EF). This hybrid plexcitonic system supports a gap plasmon mode with an ultrasmall mode volume $V_m$ of approximately 26 nm³, controlled by a single CB[7] molecule spacer with a thickness of approximately 0.9 nm. These advantages yield a large SRS of 116 meV for a single exciton interacting with the plasmons, which is large enough to exceed the system damping and confidently realize the single-exciton strong coupling in a deterministic manner. These results show that the hybrid AuND fabricated using host-guest chemistry combined with CB[7]-mediated active self-assembly is an ideal platform for addressing the deterministic positioning and alignment of the excitonic dipole in achieving single-exciton strong coupling.

## Results

### Precise positioning and alignment of a single molecule in AuND

Figure 1a summarizes our strategy for precisely positioning a single MB molecule at the gap center of an AuND and perfectly aligning the excitonic dipole moment with the plasmonic field. We embedded single MB molecules in individual macrocyclic CB[7] molecules using host-guest chemistry[7,36]. The pumpkin-shaped CB[7] molecules with hollow, hydrophobic internal volumes are water-soluble, and only one MB molecule can be accommodated in one CB[7] molecule (Fig. 1b), which allows for the construction of individual CB[7]@single MB emitters (see the Methods section for details). The encapsulation of

single MB molecules inside the CB[7]s was verified using experimental measurements, as highlighted in Supplementary Note 1 (Supplementary Figs. 1 and 2). Firstly, the absorption spectroscopy of MB dimers, shown by the small shoulder peak at 615 nm, disappeared on mixing MB with CB[7]. Secondly, the fluorescence intensity of the MB monomers encapsulated in CB[7] molecules was enhanced by more than three times (Supplementary Fig. 2a), owing to the enhancement in the fluorescence quantum yield of these monomers hosted in CB[7] cavities. The fluorescence quantum yield and brightness of MB molecules, well-known to have a low fluorescence quantum yield[37], are enhanced due to the low polarizability inside the CB[7] cavities and the suppression of some relaxation processes of the dye molecules[38]. Thirdly, the relaxation suppression of the encapsulated MB monomers significantly prolonged the fluorescence lifetimes of the bare MB molecules in aqueous solution from 0.48 ns to 2.11 ns (Supplementary Fig. 2b), which is in agreement with literature[38,39].

By embedding a single MB molecule in CB[7], the transition dipole moment of the MB molecule (along the longitudinal direction of MB molecule[40]) can be confined to the direction normal to the CB[7] torus (Fig. 1a (upper panel)). Next, we integrated a single MB molecule into the gap center of the AuND through CB[7]-mediated active self-assembly. As CB[7] has multiple carbonyl groups, it can interact with the Au atom and form coordination bonds via each carbonyl group[41] (Supplementary Fig. 3), causing the CB[7] molecule in the CB[7]@single MB emitter to strongly bind and hold two Au nanoparticles to

form an AuND with a fixed interparticle separation, which equals the thickness of CB[7] (Supplementary Fig. 4e-l). Increasing the amount of CB[7] and the hybridization time further improved the yield of AuNDs and other aggregations (Supplementary Fig. 5); this also indicates that the CB[7] molecule in CB[7]@single MB emitters plays a key role in the formation of AuNDs and other aggregations (Supplementary Fig. 6). After the CB[7]@single MB emitters are added to the colloidal Au nanoparticles, CB[7] can actively bind to the surface of the nanoparticle through its carbonyl-fringed portals on one side (Fig. 1a (middle panel)). When another Au nanoparticle moves close to this CB[7]@single MB emitter functionalized Au nanoparticle, the carbonyl-fringed portals on the other side of the CB[7] cage can grasp the second nanoparticle to form the hybrid AuND with a single MB molecule embedded in the gap center[36] (Fig. 1a (bottom panel)), i.e., the hot spot with $EF_{max}$ (Fig. 1c-e, and Supplementary Movie 1). Finally, the coupling system of AuND/CB[7]@single MB is constructed, in which the dipole moment of the encapsulated MB molecule can be fine-controlled along the gap plasmon mode (Fig. 1a (bottom panel)) and Fig. 1e (bottom panel)).

To tune the gap plasmon mode in resonance to the exciton absorption at approximately 660 nm (Fig. 1f), we selected Au nanoparticles with a diameter of approximately 40 nm to construct the hybrid AuNDs. Our numerical calculations show that the damping linewidth, $\Gamma_d$, of the gap plasmon mode supported by the bare AuND (without the MB molecule) displays a remarkable decrease from 274 to 126 meV, along with a reduction in the gap distance ($d$) from 10 to 0.9 nm (See Supplementary Note 2 and Supplementary Fig. 7), indicating relatively lower damping of the gap plasmon mode supported by the AuNDs with sub-nanometer gaps. It is also shown that the AuND with $d = 0.9$ nm supports a $Q$-factor of approximately 15, slightly higher than that of the NPoM configuration ($Q$ of approximately 11–12) constructed by placing a same-sized Au nanoparticle on an Au film[7] (Supplementary Fig. 8). Additionally, the plasmonic EF in the AuND is highly confined in its gap (Fig. 1e), giving rise to a giant Purcell factor of approximately $3.2 \times 10^6$ (at 659.8 nm) in the gap center of the AuND (Supplementary Note 3 and Supplementary Fig. 9). This facilitates the realization of the single-exciton strong coupling in such a plasmonic nanocavity[42].

Dark-field scattering measurements were performed to demonstrate the strong coupling of a single exciton and the AuND (Methods section, Supplementary Figs. 10 and 11). Figure 1f shows the scattering spectra of a bare AuND and a hybrid AuND/CB[7]@single MB. Compared to the bare AuND, a clear SRS of $\hbar\Omega_R \approx 116$ meV is observed in the scattering of this hybrid AuND coupling with a single MB molecule. By fitting the observed SRS using Eq. (1)[43] (Supplementary Note 4),

$$\hbar\Omega_R = 2\sqrt{\sqrt{N}g_{dc}(1 + \Gamma_c/\Gamma_d) \cdot (Ng_{dc}^2 + \Gamma_c\Gamma_d/4)^{1/2} - (Ng_{dc}^2 + \Gamma_c\Gamma_d/4) \cdot \Gamma_c/\Gamma_d},$$
(1)

with $N = 1$ ($N$ is the exciton number), $\Gamma_d = 135$ meV (Supplementary Fig. 12), and $\Gamma_c = 85$ meV, we extract a coupling constant of $g_{dc} = 56.6$ meV for a single exciton interacting with plasmons. $\Gamma_d$ ($\Gamma_c$) is the measured damping linewidth of the gap plasmon mode (exciton absorption) (Fig. 1f). The fitted coupling constant is very close to its theoretical limit of $g_0 = 56.9$ meV (Fig. 1g), which was calculated using Eq. (2):

$$g_0 = \frac{\hbar\omega_c}{\sqrt{2\hbar\varepsilon_0\omega_d V_{eff}}} \mu_c \cdot \hat{\mathbf{f}}_d(\mathbf{r}_c),$$
(2)

based on the plasmonic effective mode volume of $V_{eff} = 42.6$ nm³ (Supplementary Table 2) and $\mu_c = 0.09$ e nm[44], a typical value for single excitons[34]. In Eq. (2), $\hat{\mathbf{f}}_d(\mathbf{r}) = \mathbf{E}_d(\mathbf{r})/|\mathbf{E}_d(\mathbf{r})|_{max}$ is the normalized EF intensity at the position $\mathbf{r}$. In calculations, $\mu_c$ is set parallel to the mode field, $\hat{\mathbf{f}}_d(\mathbf{r}_c)$, and the high consistency between $g_{dc}$ and $g_0$ indicates a

fine alignment of $\mu_c$ along the plasmonic field. The extracted value of $g_{dc} = 56.6$ meV satisfies the strong coupling condition of $g_{dc}^2 > (\Gamma_d^2 + \Gamma_c^2)/8$ [18,45], indicating that the light-matter interaction observed in this hybrid system is in the strong coupling regime.

## Single-exciton strong coupling in AuNDs with different detuning

Figure 2a depicts the scattering spectra of three AuND/CB[7]@single MB hybrids with different detunings, achieved by integrating single CB[7]@single MB emitters in AuNDs, which are constructed using Au nanoparticles of different sizes (Supplementary Fig. 4a-d). In sample fabrication, an average dye concentration of $\bar{n}_{MB} = 0.8$ CB[7]@single MB emitter per nanoparticle was used. At such a low concentration, approximately 20–25% of the prepared samples correspond to AuNDs, out of which approximately 15% are in strong coupling, indicating a yield of 3–4% of the samples demonstrate strong coupling in dark-field scattering measurements. The results in Fig. 2a show an agreement between the measurements and the theoretical results calculated using Eq. (3), which was deduced from a quantum mechanical model for describing the plasmon-exciton strong coupling[18],

$$\sigma(\omega) \propto -Im \frac{(\hbar\omega - \hbar\omega_c + i\Gamma_c/2)}{(\hbar\omega - \hbar\omega_d + i\Gamma_d/2) \cdot (\hbar\omega - \hbar\omega_c + i\Gamma_c/2) - Ng_{dc}^2}.$$
(3)

Figure 2b depicts additional single-exciton strong coupling observed in single hybrid AuNDs with different detunings. The upper panel of Fig. 2b shows the normalized scattering spectra of the individual AuND/CB[7]@single MB hybrids arranged by the experimental detuning, $\delta_{exp} = \hbar\omega_+ + \hbar\omega_- - 2\hbar\omega_c$ [14], with the exciton absorption at $\hbar\omega_c = 1.880$ eV (Fig. 1f). Anti-crossing behavior can be observed in Fig. 2b, constituting the signature of the single-exciton strong coupling. The lower panel of Fig. 2b presents the normalized theoretical spectra calculated using Eq. (3) at different theoretical detunings $\delta = \hbar\omega_d - \hbar\omega_c$. We set $N = 1$ in calculations, while the rest of the parameters were extracted from the experimental measurements. The calculated results displayed a clear anti-crossing behavior with an SRS of $\hbar\Omega_R \approx 116$ meV (Fig. 2b (lower panel)), which is in good agreement with the experimental measurements (Fig. 2b (upper panel)). In the strong coupling regime, plasmons and excitons lose their identities and generate the hybrid quasiparticles of plexcitons. Figure 2c demonstrates the plasmon and exciton fractions in the upper plexciton branch (UPB) and lower plexciton branch (LPB) of the strongly coupled AuND/CB[7]@single MB hybrids, where a significant hybridization between the plasmon and exciton can be seen at the resonance ($\delta = 0$).

Strongly coupled AuND/CB[7]@single MB hybrids easily bleached under laser illumination (Supplementary Note 5 and Supplementary Fig. 13), indicating that the observed energy splitting is indeed due to the MB molecule. Additionally, owing to the extremely enhanced electric field intensity in the nanogap, the photoluminescence (PL) measurements also generated strong surface-enhanced Raman scattering (SERS)—consisting of sharp lines with a fluorescence background—that cannot be uniquely separated from photoluminescence[7]. All these factors obscured the use of the PL measurements to confirm further the single-exciton strong coupling in AuND/CB[7]@single MB hybrids. To further verify the presence of single MB molecules in individual AuND/CB[7]@single MB hybrids shown in Fig. 2a,b, we employed the bianalyte method, which has been well-proven in chemistry for confirming surface-enhanced Raman scattering (SERS) of single molecules[46,47]. Herein, we selected proflavine (PF) as a second molecule for the bianalyte SERS detection as it chemically and structurally resembles MB and binds inside CB[7] in the same way[7]. At the low dye concentration of $\bar{n}_{MB} \approx 0.4$ CB[7]@single MB and $\bar{n}_{PF} \approx 0.4$ CB[7]@single PF emitters per Au nanoparticle on average, which is the same as that used in Fig. 2a,b, the probability that two dye molecules

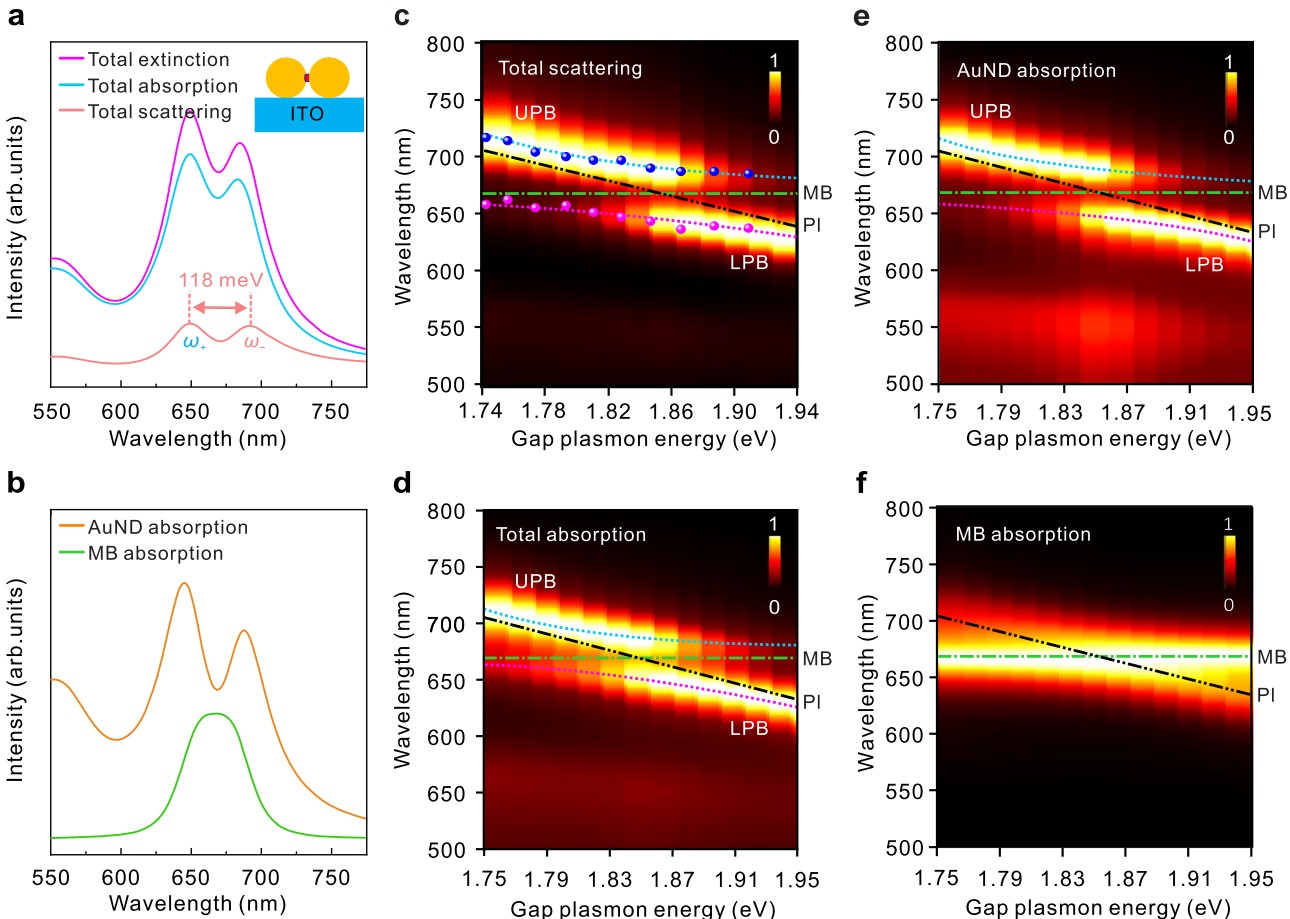

**Fig. 3 | Theoretical confirmation of strong coupling in the hybrid AuND containing a single MB molecule. a** Simulated scattering, absorption, and extinction cross-section spectra of the hybrid AuND/CB[7]@single MB system with a nanoparticle diameter of 42 nm. **b** Simulated absorption cross-section spectra of the coupling components of the AuND and MB molecule in the hybrid system. **c, d** Simulated total scattering and absorption cross-section spectra of the AuND/ CB[7]@single MB hybrids, arranged according to the different gap plasmon energies. The colored ball-shaped markers in (**c**) represent the typical experimental results extracted from Fig. 2b (upper panel). **e, f** Simulated absorption cross-section spectra of the coupling components of the AuND (**e**) and the MB molecule (**f**) in the hybrid AuND/CB[7]@single MB, arranged according to the gap plasmon energies. AuND: Au nanodimer, MB: methylene-blue, Pl: Plasmon.

simultaneously accommodate in the gap of an AuND is very low (Supplementary Fig. 14). Figure 2d demonstrates that MB and PF can be easily distinguished using the lowest two SERS bands at 400–410 cm$^{-1}$ for PF and 460–470 cm$^{-1}$ for MB, which allows us to construct a histogram that plots the probability of single-molecule signature[7,48]. The results show that at such a low dye concentration, two molecules are rarely detected simultaneously, which indicates that the results shown in Fig. 2a,b are truly in the single-molecule exciton strong coupling regime.

**Numerical simulations of the strong coupling in single AuND/ CB[7]@single MB**

The single-molecule exciton strong coupling observed in Fig. 2 can be reproduced through numerical simulations using the finite element method (FEM). Figure 3a shows the simulated scattering, absorption, and extinction cross-section spectra of the hybrid AuND/CB[7]@single MB, which demonstrate clear SRSs, where the CB[7]@single MB emitter was modeled as a dispersive medium cylinder (0.75 nm in radius and 0.9 nm in length, which is equal to the size of a CB[7]) with a permittivity described by the Lorentz model (discussed in the Methods section). Interestingly, such SRS can also be observed in the absorption cross-section spectrum of the coupling subsystem of AuND in the hybrid system (Fig. 3b and Supplementary Note 2), indicating that the strong coupling signal from the plasmonic channel is a hybridization of the plasmon mode and exciton transition, therefore further

supporting that single-exciton strong coupling occurred in the hybrid AuND/CB[7]@single MB[43]. By increasing the diameter of the Au nanoparticle from 30 to 54 nm, we can tune the gap plasmon resonance of the AuND from 1.94 eV to 1.74 eV (Supplementary Fig. 15), which covers the MB molecule absorption ($\hbar\omega_c$=1.880 eV) well. By tuning the resonance of the gap plasmon, we obtained the anti-crossing behavior with $\hbar\Omega_R \approx 118$ meV (larger than the system dissipation, $\hbar\Omega_R > (\Gamma_d + \Gamma_c)/2$) in the energy dispersions composed of the scattering cross-section spectra for the individual AuND/CB[7]@single MB hybrids with different particle sizes (Fig. 3c). The superimposed numerical anti-crossing curves in scattering with the experimental data extracted from Fig. 2b (upper panel) demonstrates a good agreement between the experimental measurements and the simulated results (Fig. 3c).

Aside from the anti-crossing dispersion observed in scattering, such dispersion behaviors can also be observed in the absorption from both the hybrid system and the coupling component of AuND with different detuning (Fig. 3d,e). This provides additional evidence of single-molecule exciton strong coupling achieved in our hybrid AuND system[49], even though the experimental single-nanostructure absorption measurement is more complex than the performed dark-field scattering[50,51] because the incident light is always scattered by the nanostructures[52]. In addition, we simulated the near-field distributions in the gap of the hybrid system at the upper $\omega_+$(649 nm) and lower $\omega_-$(692 nm) plexciton states, as well as the Rabi split valley at 665 nm

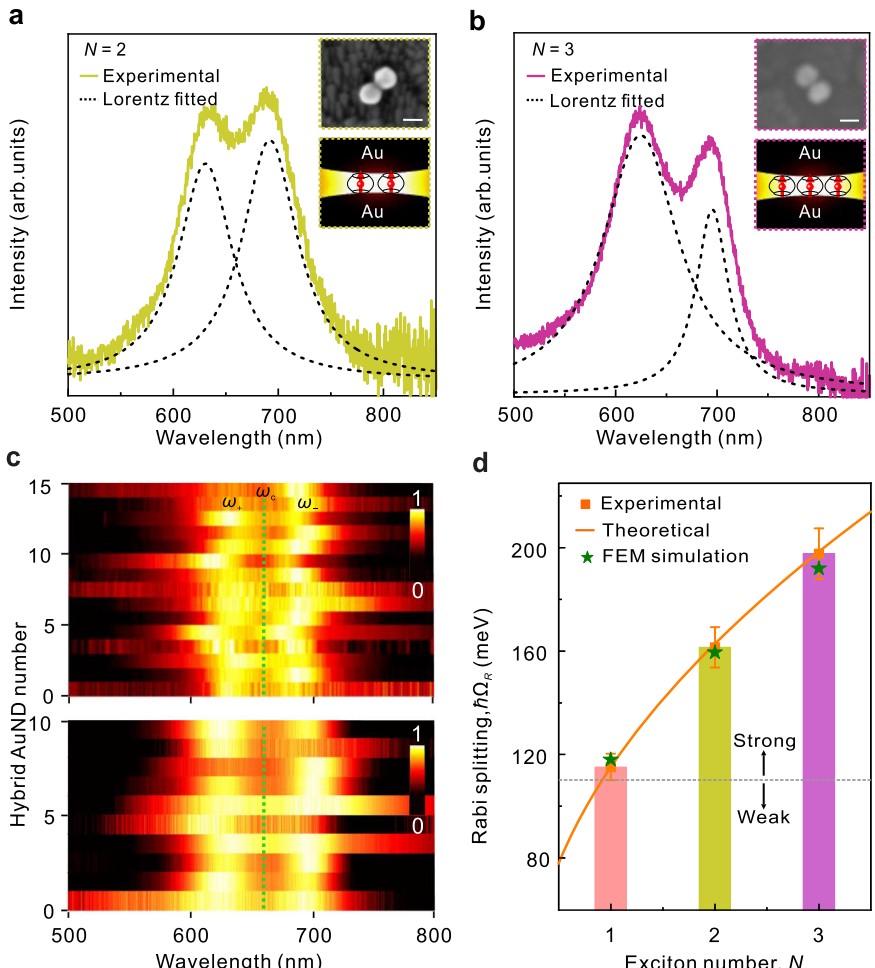

**Fig. 4 | Strong coupling of two- and three-molecule excitons with single AuNDs.** **a, b** Representative scattering spectra of the hybrid AuNDs interacting with (**a**) two and (**b**) three MB-molecule excitons. The insets show the SEM images (upper) of the measured samples and the schematics (bottom) of two and three CB[7]@single MB emitters embedded in the gap of the hybrid AuND. The dashed black lines represent the Lorentzian fits of the upper and lower plexcitonic states of the measured SRSs. **c** SRS observed in scattering spectra associated with different individual hybrid AuNDs, including (upper panel) two and (lower panel) three MB molecules. **d** Statistics of the experimentally measured SRSs as a function of the exciton number ($N$). The orange square with error bars (standard deviations, SDs) represents the statistics of the measured SRSs and the number of the sampled structures (40, 15, and 10, for exciton numbers $N = 1$, 2, and 3, respectively). The colored histogram represents the average values of the measured SRSs for $N = 1$, 2, and 3. The theoretical result (orange curve) was calculated using Eq. (1) with $g_{dc} = 56.2$ meV, $\bar{\Gamma}_d = 135$ meV, and $\Gamma_c = 85$ meV; the green stars represent SRSs simulated using the finite element method (FEM) for the hybrid AuNDs integrated with one, two, and three MB molecules, respectively. The dashed gray line is calculated according to $\hbar\Omega_R = (\Gamma_d + \Gamma_c)/2$.

(near the exciton absorption at approximately 660 nm) in the scattering spectrum (Supplementary Fig. 16). At the dye resonance, the EF shows a minimum around the junction between the two Au nanoparticles, corresponding to an enhanced molecular exciton conductance due to the plasmon-exciton strong coupling[7,53,54]. This interesting phenomenon originates from the extended coherence of the polaritonic states induced by the hybridization of the exciton state with the mode field at resonance ($\omega_d = \omega_c$)[53]. Then, the enhanced exciton conductivity improves the delocalization of the charge carriers, leading to a remarkable decrease in the EF intensity of the mode field around the junction between the two Au nanoparticles at the dye resonance.

## Discussion

Substantially stronger couplings between the AuND and two-molecule ($N = 2$) or three-molecule ($N = 3$) excitons can be achieved by increasing the amount of CB[7]@single MB emitters in the fabrication process of the coupling systems (as discussed in the Methods section). In the case of two or three CB[7]@single MB emitters, all emitters can potentially be simultaneously embedded

into the gap of the same AuND (shown in the insets of Fig. 4a,b), resulting in two- or three-molecule exciton strong coupling with the gap plasmon mode. Figure 4a and b, which show typical scattering spectra of two hybrid AuNDs containing two and three MB molecules, demonstrating an SRS of $\hbar\Omega_R \approx 160$ and 197 meV, respectively. By substituting these two SRS values into Eq. (3) with $g_{de} = 56.6$ meV, we estimate that $N \approx 2$ and 3 excitons were involved in these two strong coupling cases, respectively. Additional cases of two- and three-exciton strong coupling are shown in Fig. 4c, where the plexciton states corresponding to $\hbar\omega_{\pm} = \hbar\omega_c \pm \hbar\Omega_R/2$ are observed. In Fig. 4d, we plot the statistics of the measured SRSs for the strong coupling cases of $N = 1$, 2, and 3 with the statistical $\hbar\Omega_R \approx 115.2 \pm 5.1$, $161.5 \pm 7.8$, and $197.7 \pm 9.8$ meV, respectively. These experimental statistics are in accordance with the theoretical prediction (indicated by the orange curve in Fig. 4d), which was calculated using Eq. (1) by setting $g_{de} = 56.2$ meV. We also simulated the SRSs associated with the scattering of the hybrid AuNDs integrated with one, two, and three MB molecules, as indicated by the green stars in Fig. 4d (see the Methods section for simulation details). The experimental measurements correspond well with the analytically calculated

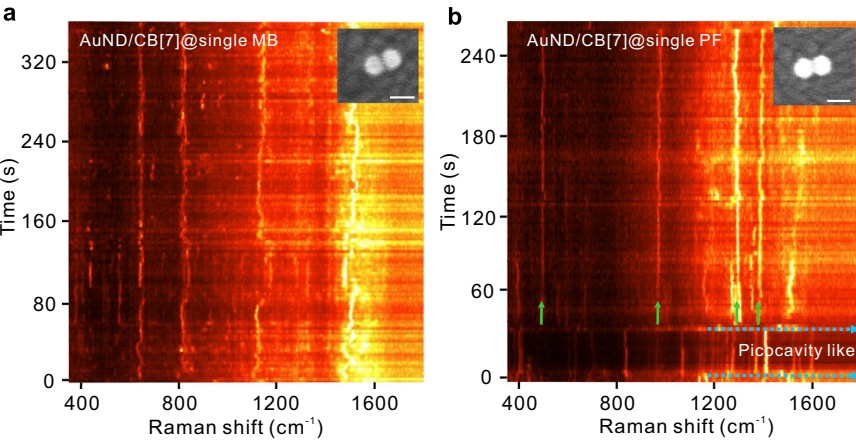

**Fig. 5 | Time-series SERS of single dye molecules embedded in single hybrid AuNDs. a, b** Time-dependent evolution of SERS signals from (**a**) a strongly coupled AuND/CB[7]@single MB and (**b**) a hybrid AuND/CB[7]@single PF. The insets show SEM images of the corresponding measured samples. The scale bar is 50 nm. Each spectrum was collected for 2 s, and the samples were excited at 633 nm. New vibrational lines between the blue arrows indicate the possible formation of a picocavity in the hybrid AuND. The green arrows indicate the nanocavity SERS lines. AuND: Au nanodimer, MB: methylene-blue, PF: proflavine.

results and the simulated Rabi splitting, demonstrating a signature of the strong coupling involved a quantized exciton number.

The single-molecule behaviors in the hybrid AuNDs can also be observed through the time-series SERS spectra of individual AuND/CB[7]@single MB hybrids. Figure 5a presents the time-dependent evolution of the SERS signals from a strongly coupled AuND/CB[7]@single MB, which exhibited the signature of single-exciton strong coupling in its scattering spectrum. These SERS signals demonstrate the clear single-molecule signature—spectral diffusion, which is well-proven evidence of single-molecule events both in the fluorescence community[55] and in the SERS community[7,56]. More time-series SERS spectra of the strongly coupled AuND/CB[7]@single MB hybrids are shown in Supplementary Fig. 17. Such single-molecule behavior can also be seen in the time-series SERS spectra of the hybrid AuND/CB[7]@single PF (Fig. 5b), where new vibrational lines can be observed over a certain period (between the dashed blue arrows) alongside the nanocavity SERS lines (indicated by green arrows), despite these nanocavity lines appearing less stable due to single molecules located in the AuND. This interesting phenomenon indicates the possible formation of a picocavity[57,58] in this hybrid AuND/CB[7]@single PF system.

To summarize, we have realized the strong coupling of a single-molecule exciton at room temperature in AuNDs by combining the extremely localized gap plasmon resonances with the methods of oriented host-guest chemistry and CB[7]-mediated active self-assembly. These methods enabled us to achieve plasmonic nanocavities with ultrasmall $V_m$ and guarantee that a single dye molecule is integrated into the plasmonic nanocavity at the deterministic position with $EF_{max}$. Meanwhile, the transition dipole moment of the exciton can also be finely aligned along the plasmonic field. These merits are crucial in achieving the deterministic single-exciton strong coupling with the plasmonic nanocavities. We successfully demonstrated a large and clear SRS of 116 meV for a single-molecule exciton, which is strongly coupled to the extremely confined gap plasmons. These strong coupling results achieved in our hybrid AuND systems remove the ambiguities in explaining the plasmon-exciton coupling, ranging from the uncertainties in the number, location, and dipole moment direction of the emitter involved in the coupling to a question of whether the plasmonic nanocavities are at all capable of realizing strong light-matter interactions at its fundamental limit[3]. In addition, because the plasmonic field in the gap of an AuND is parallel to the dimer direction, it is more suitable to investigate the polarization dependence of the plexcitons[59,60], compared to the NPoM structures[7,35]

where the plasmonic field is oriented perpendicular to the substrate (Supplementary Note 6). Supplementary Fig. 18 shows the polarization-resolved dark-field scattering of the strongly coupled AuNDs with MB-molecule excitons, indicating that the UPB and LPB modes in such strongly coupled systems have different polarization-dependent properties, thereby paving a hopeful way to manipulate the mixed plexcitonic states through control of the polarization. Therefore, such AuND-single molecule coupling systems are not only ideal for exploiting the single-molecule enhanced spectrum[61] and quantum chemistry[53,62,63], but are also crucial for quantum optics applications where a deterministic single qubit is needed. This work represents a significant step towards to single-emitter strongly coupled nanodevices, with interesting prospects for room-temperature quantum technologies.

## Methods

### Materials and synthesis of the Au nanoparticles
See Supplementary Note 7.

### Fabrication of AuND/CB[7]@single MB hybrids
We used MB molecules to fabricate the single-molecule emitters. We employed the host-guest chemistry of CB[7] molecules to avoid aggregation of the dye molecules and fabricate single-molecule emitters. Specifically, the CB[7] aqueous solution (5 mL) of $1.0 \times 10^{-7}$ M was prepared in a thermostat water bath at 50 °C, and then 2.5 mL of $1.0 \times 10^{-7}$ M MB aqueous solution was added and thoroughly mixed using ultrasonic wave for 10 min (the molar ratio of MB and CB[7] was 1:2). Following this process, single MB molecules embedded in individual CB[7] can be successfully obtained as CB[7]@single MB emitters. Next, 4 mL of as-prepared Au nanoparticles in CTAB were centrifuged and washed twice with deionized water, and subsequently the deposit was dispersed in 1 mL of deionized water. Next, 20 µL of the as-prepared CB[7]@single MB ($3.33 \times 10^{-8}$M) was added and hybridized with the washed Au nanoparticles for 1 h. The CB[7]@single MB emitters injected in the colloidal Au nanoparticles acts as a molecular glue to connect the individual Au nanoparticles and form the AuND/CB[7]@single MB hybrids and other aggregations via the carbonyl-fringed portals of CB[7].

Based on the amount of Au used in the preparation process, approximately $5.0 \times 10^{11}$ Au nanoparticles were used to hybridize with dye molecules (Supplementary Note 7 and Supplementary Fig. 19). Considering that approximately $4.0 \times 10^{11}$ MB monomers were added, there were on average $\bar{n}_{MB} \approx 0.8$ CB[7]@single MB emitter and $\bar{n}_0 \approx 0.8$

empty CB[7] per Au nanoparticle (the molar ratio of MB and CB[7] was 1:2). Such a low MB concentration enable us to obtain the single-MB exciton that is strongly coupling with a single AuND, despite the fact that the yield of the AuND was only 23% after the hybridization of CB[7]@single MB molecules and Au nanoparticles for 1 h (Supplementary Fig. 20a,b). Increasing the amount of CB[7]@single MB emitters or the hybridization time in the fabrication process can further improve the amount of hybrid AuNDs and other aggregations. Approximately 15% of the AuNDs demonstrated strong coupling in our measurements, and the rest did not feature energy splitting with and without MB molecules. In these strong-coupling cases, the AuNDs featured SRS consistent with that of one MB molecule, accounting for approximately 83.3% of the total strong-coupling cases, and approximately 13.3% of these strong coupling dimers demonstrate the two-MB strong coupling. A small part (approximately 3.3%) displayed three-MB molecule strong coupling (Supplementary Fig. 20c). In addition, the chances for two and three MB molecules to be simultaneously embedded in the same gap are also improved, which is more suitable for observing the two- and three-molecule strong coupling in individual AuND/CB[7]@single MB hybrids. Therefore, for the samples used in Fig. 4c, we have increased the dye concentrations to $\bar{n}_{MB} \approx 4$ (upper panel) and 8 (lower panel) CB[7]@single MB emitters per Au nanoparticle, respectively.

### Dark-field scattering and SERS measurements of the AuND/ CB[7]@single MB hybrids

The schematic of our dark-field scattering measurements is shown in Supplementary Figs. 10, 11. Before the dark-field scattering measurements, 10 µL of the AuND/CB[7]@single MB hybrids was dropped and cast onto the surface of the ITO-coated glass substrate. After 5 min, the droplet was removed, and the samples were washed with deionized water and dried under nitrogen. Then, the individual AuND/CB[7]@single MB hybrids isolated from the sample were formed and fixed on the surface of the ITO-coated glass substrate. To obtain the scattering spectra of these individual hybrids, we combined dark-field microscopy with SEM imaging of the same field of view to correlate elastic scattering spectra and the structures on the level of individual AuND/CB[7]@single MB hybrid. The scattering images and spectra of the individual hybrids were recorded on a dark-field optical microscope (Olympus BX53M, Olympus Inc., Japan) that was integrated with a monochromator (Acton Spectra Pro 2360, Acton Inc., USA), a quartz tungsten halogen lamp (150 W), and a charge-coupled device camera (Princeton Instruments Pixis 100B_eXcelon, Acton Inc., USA) which was cooled to −70 °C during the measurements. To obtain the scattering spectra of the individual hybrids, the light was launched from a dark-field objective (100×, numerical aperture 0.80), and the same objective collected the light scattered in the backward direction. The color scattering images were captured using a color digital camera (ARTCAM-300MI-C, ACH Technology Inc.) mounted on the imaging plane of the microscope.

The SERS signals of individual AuND/CB[7]@single MB hybrids were collected using a Raman spectrometer (NTEGRA Spectra PNL, NT-MDT Co., Russia), which combined the dark-field scattering imaging and the corresponding SEM characterization. The excitation source was a 633 nm laser, and each spectrum was collected for 2 s at a 60 µW/µm² pump intensity.

### Sample characterizations and optical measurements

The SEM images were taken on Zeiss Auriga-39-34 SEM and JEOL JSM-7610F Plus machines operated at 5 kV. The TEM images were taken on JEOL JEM-F200 machine operated at 100 kV. The absorption measurements were taken on an ultraviolet-visible-near-infrared spectrophotometer (Cary 5000), and the reference was 3 mL deionized water.

### Mode volume calculation

The AuND with a sub-nanometer gap can be viewed as a closed optical nanocavity, and its mode volume, $V_m$, can be calculated using[3,7]

$$V_m(\omega) = \frac{\int \varepsilon(\mathbf{r},\omega)|E(\mathbf{r},\omega)|^2 d\mathbf{r}}{\max[\varepsilon(\mathbf{r},\omega)|E(\mathbf{r},\omega)|^2]}, \quad (4)$$

via the FEM. To make this equation applicable to plasmons, the $\varepsilon(\mathbf{r},\omega)$ term was modified as[3,18].

$$\varepsilon(\mathbf{r},\omega) \rightarrow \mathrm{Re}[\varepsilon(\mathbf{r},\omega)] + 2\omega \mathrm{Im}[\varepsilon(\mathbf{r},\omega)]/\beta, \quad (5)$$

where $\varepsilon(\mathbf{r},\omega)$ is the permittivity of the metal at the position of $\mathbf{r}$, $\beta$ is the plasmonic damping term, and Re[] and Im[] are the real and imaginary parts, respectively. Recently, quasinormal mode theory (QNM) has been proposed as an alternative method of calculating the mode volume of non-Hermitian plasmonic nanocavities, even picocavities[64,65].

### Optical response simulations of hybrid AuND/CB[7]@single MB

We calculated the EF distribution, light scattering, absorption, and extinction of the hybrid AuND/CB[7]@single MB using the finite element method (FEM, COMSOL Multiphysics). The hybrid AuND/CB[7]@single MB was located on the surface of the ITO-coated glass substrate with a refractive index of 2.0. We also calculated the absorption cross-section spectra of the coupling components of AuND and MB molecule in the hybrid system. The AuND in the hybrid AuND/CB[7]@single MB was composed of two Au nanospheres (42 nm in diameter) with a nanogap of 0.9 nm, which equals the thickness of the molecular spacer of CB[7]. The single MB molecule in CB[7], i.e., CB[7]@single MB, was modeled as a dispersive medium cylinder with a diameter of 1.5 nm and a length of 0.9 nm, which equals the diameter and thickness of the molecular spacer of CB[7], respectively. The dielectric permittivity of the cylinder was described using the Lorentz model[59]:

$$\varepsilon(\omega) = \varepsilon(\infty) + \frac{f_0 \omega_c^2}{(\omega_c^2 - \omega^2 - i\Gamma_c\omega)}, \quad (6)$$

where $\varepsilon(\infty) = 1.96$ is the high-frequency component of the MB molecule matrix dielectric function, $f_0 = 0.27$ is the oscillator strength for a single MB molecule, and $\omega_c = 1.880$ eV and $\Gamma_c = 85$ meV are absorption energy and linewidth of the uncoupled MB-molecule exciton, respectively. In the calculations, an effective refractive index of 1.26 was set above the ITO substrate, owing to a significant roughness of the surface of ITO-coated glass substrate compared to the small Au nanoparticles, which ensured that the calculated resonant wavelength of the AuNDs agrees with the experimental measurement results under the same structural parameters (Supplementary Fig. 21). In simulations, we calculated the absorption cross-section spectrum of the MB molecule by integrating the total density of the power dissipation in the domain of the CB[7]@single MB, i.e., the dispersive medium cylinder between the two Au nanospheres, and then we normalized the integral result to the intensity of the incident field ($I_0$). Similarly, the absorption cross-section spectrum of the AuNDs in the hybrid system was calculated by integrating the total density of power dissipation in the domain of the two Au nanospheres (the dielectric permittivity of Au dimer was described by the Brendel-Bormann model); the integral result was also normalized to $I_0$.

In addition to the calculations of the hybrid AuND/CB[7]@single MB, we also calculated the SRS of the hybrid AuND integrated with two or three CB[7]@single MB emitters using the FEM (Fig. 4d). Considering the total volumes of two and three CB[7]@single MB emitters embedded in the dimer gap; we modeled these two cases as an

effective medium cylinder with the same volume as two and three CB[7] molecules, respectively. The effective medium cylinder used to model these two cases was 1.1 nm $(0.75 \times \sqrt{2}\,nm)$ and 1.3 nm $((0.75 \times \sqrt{3}\,nm)$ in radius, respectively, and 0.9 nm in length).

## Data availability

Relevant data supporting the key findings of this study are available within the article and the Supplementary Information file. The original data generated during the current study are available from the corresponding authors upon request.

## Code availability

The calculation data were obtained by evaluating the equations and simulating the nanostructures using standard technical software. The codes are available upon request to the corresponding authors.

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

## Acknowledgements

We thank Dr. Wei Li for helping with numerical simulations and Prof. Ying Yu for helping with optical measurements. This work was supported by the National Natural Science Foundation of China under Grant No. 12374326 (R.L. and J.-F.L.) and No. 12004101 (Y.K.), the Key Project of Natural Science Foundation of Henan under Grant No. 232300421141 (R.L.), the Start-Up Research Grant under Grant No. SRG SMT 2021 169 (L.W.) and Kickstarter Initiative (SKI) under Grant No. SKI 2021_02_14 (L.W.) from the Singapore University of Technology and Design, and the National Research Foundation Singapore under Grant No. NRF2021-QEP2-02-P03 (L.W.), No. NRF2021-QEP2-03-P09 (L.W.), and No. NRF-CRP26-2021-0004 (L.W.).

## Author contributions

R.L., J.-F.L., L.G., and L.W. were responsible for the idea and supervised the project. J.-F.L and R.L. contributed to the theoretical calculations. M.G., Y.-W.L., and J.-F.L. performed the FEM numerical simulations. M.G. and Z.L. performed the sample preparations. J.A., X.F., and Y.K. contributed to the experimental setup and the optical measurements. J.A. and M.G. contributed to SERS measurements. R.L., J.-F.L., L.G., and L.W. were responsible for the manuscript's interpretation, writing, revision, and finalization. All authors discussed the results and commented on the paper.

## Competing interests

The authors declare no competing interests.
