## [Peer Review File · Nature Communications]

Deterministic positioning and alignment of a single-molecule exciton in plasmonic nanodimer for strong couplingEditorial Note: Parts of this Peer Review File have been redacted as indicated to remove third-party material where no permission to publish could be obtained.

REVIEWER COMMENTS

Reviewer #1 (Remarks to the Author):

In “Deterministic positioning and alignment of a single-molecule exciton in plasmonic nanodimer for strong coupling”, R. Liu et al. report that they observe the strong coupling regime between a single molecular exciton and a plasmonic resonator. Observing this coupling regime is an essential step for the development of quantum devices working at room temperature without the need for costly and energy-intensive cryogenic equipment. However, the current manuscript does not provide any unequivocal demonstration that the strong coupling regime is observed with a single exciton and would require extensive extra information, as well as reference and additional experiments before it can be considered for publication. If the authors can provide all the required data to answer the following points, then their report would be of interest to a broad scientific community and could justify publication in Nature Communications.

1-The first issue is that the authors never prove that there is indeed one and only one methylene blue (MB) molecule in the plasmonic gap. They mention that “only one MB molecule can be accommodated in one CB[7] molecule” without proving it: can they justify it with, at the very least, a reference discussing this point? More importantly, they never prove that only one CB[7] is located in the gap. Can they prove unequivocally that it is the case? At most, I believe that the authors can say that their data are consistent with one molecule in strong coupling regime but that this remains a strong hypothesis.

2-What is the percentage of dimers in strong coupling? What is the percentage of dimers with coupling consistent with one, two or three MB molecules in the gap? What is the percentage of dimers in the sample? There is absolutely zero quantitative information on the sample that is produced and analyzed in the manuscript. The authors must clearly say what is the percentage of single particles, dimers or larger aggregates in their sample. And

they must say how many dimers are in strong coupling compared to the number of dimers that do not feature energy splitting with and without MB molecules.

3-How can the authors ensure that the CB[7] molecule is responsible for the formation of dimers? They mention that “the CB[7] molecule can actively bind to the Au atoms on the surface of the nanoparticle through its carbonylfringed portals” but can they actually prove it? Do they still observe dimers without the CB[7] molecule? When gold particles are dried on substrates, dimers always appear spontaneously: how can we be sure that the CB[7] molecule is indeed in the gap?

Furthermore, since they do not control the number of CB[7] molecules per particle, why do they only observe dimers? In Figure S3, they show an image with a lot of dimers but this does not make sense: if they introduce a very limited number of CB[7] molecules, most particles should remain as single particles and only a few dimers should appear; while, if they put a lot of CB[7] molecules, more than one will be found per particle and large aggregates should appear. It is also very strange that the darkfield images in figure S7 are so different with and without the MB molecule: why is there more aggregation with MB molecules than without them?

Overall, the authors must perform extra reference measurements by preparing samples with varying concentrations of CB[7] molecules mixed with the gold particles and prove that this concentration controls the level of aggregation observed in the final samples, which would be consistent with the fact that the CB[7] molecule acts as a bridge between gold particles.

4-How can the authors ensure that the observed energy splitting is indeed due to the MB molecule? In past reports on strong coupling (for instance, refs 3, 18 or 33 of the manuscript), the molecules were actively or passively bleached to ensure that the energy splitting disappears after bleaching. Since the authors do not directly prove that the CB[7] molecule is indeed in the gap or that there are actually MB molecules in the dimer, they must absolutely ensure that the energy splitting disappears after bleaching the MB molecules. If they cannot bleach them optically, they could maybe reduce them as the reduced state of MB is colorless. Without this demonstration, it seems impossible for me to ensure that the energy splitting observed by the authors is indeed strong coupling due to

MB molecules.

5-The authors perform numerous simulations that are confronted to experiments. First of all, these simulations do not prove anything from an experimental point of view and only mean that the experiments might be consistent with simulations of strong coupling with one exciton: it is essential to rewrite the manuscript to make this point clearer.

Furthermore, the field enhancement maps that are shown in the manuscript are strongly pixelated: how did the authors ensure convergence of their simulations?

6-There are a number of typos, especially the word “bare” confused with the word “bear” (Fig. 1-fIV written in green and several times in the SI). Please correct.

Reviewer #2 (Remarks to the Author):

The field of strong coupling in plasmonic nanosystems has made remarkable progress in recent years, pushing the limits down to the level of a few to single molecules, yet the deterministic positioning of single molecules remains a significant challenge. In this study, the authors aim to address this using guest-host self-assembly with super spherical nanoparticles to deterministically position a single molecule exciton and demonstrate strong coupling. They aim to place a single molecular emitter in the gap centre of an Au nanodimer system and claim the simultaneous achievement of both deterministic molecular positioning and dipole alignment with the mode field. Previous studies have employed similar assembly processes, mostly in systems such as nanoparticle-on-a-mirror geometry. Utilizing dimers for strong coupling could be useful and might find more potential.

Nonetheless, there are some serious concerns regarding the claims made about the possibility of single-molecule involvement in strong coupling and the accuracy of the reported coupling strength. Further detailed work is necessary at this stage, as the reported results seem preliminary. Hence, I suggest major revisions.

- Firstly, the authors describe the self-assembly process, but no experimental data is

presented to support their assertion of deterministic positioning of molecules. The process of "active" self-assembly and how it necessarily leads to a non-random outcome is unclear. It is quite speculative to assume that an individual CB[7] molecule is holding two large Au nanoparticles without dissociation. The forces required for such a holding would be substantial, given the gap size is approximately 0.9 nm. How are the authors claiming that only one molecule is present in the gap?

- When a MB molecule is enclosed within a CB[7] cage, it is very likely for some parts of the MB molecule to extend beyond the hydrophobic portal of CB[7], as MB is lengthier, also seen in Fig.1. This further obstructs the binding of CB[7] to Au nanodimers. This further raises the question of how an individual CB[7] can hold not just the MB, but also binds to Au nanoparticles at the same time. Additionally, the C=O functional groups of CB[7], which typically bind to metals, are not readily accessible when MB is present inside, as it is lengthier.

- In addition, despite the nanoparticles being spherical, the assembly process could result in dimers having a small facet size, as evident from the SEM images provided in the supplementary information. This suggests that there may be multiple CB[7]-MB systems present in the gap, potentially contributing to the observed strong coupling.

- Detailed quantification of the yield of formation of dimers and how many of those demonstrate strong coupling is required. Further, the variability in coupling strength among these dimer systems should be thoroughly studied to test and validate the number of molecules contributing to the strong coupling.

- Furthermore, the authors' use of coupling strength to calculate the dipole angle is unrealistic and does not accurately reflect the experimental conditions. This is due to several assumptions made, including the number of emitters, the estimation of mode volume, and the dipole strength of an emitter in close proximity to a metal.

- The authors claim that the coupling strength value corresponds to a single exciton, but the SEM images in Fig. 2a only suggest a small facet size, which again suggests the potential

presence of multiple CB-MB constructs in the gap. Moreover, the angle is calculated based on theoretical assumptions assuming a single molecule in the gap, which is unrealistic and potentially unreliable since the interpretation has already presupposed the single-molecule scenario.

- Lastly, the authors rely on single-exciton scattering spectral results with different detuning values to validate the findings. However, recent studies (<https://doi.org/10.1038/s41467-018-06450-4>, <https://doi.org/10.1021/acsphotonics.9b01079>) have shown that scattering spectra tend to overestimate the values and therefore, cannot be used to accurately validate whether the system is indeed in the single-exciton regime. Thus, further PL-based experiments are required to verify and corroborate the reported results.

In light of these concerns, I would recommend major revisions to the article for it to be considered.

Reviewer #3 (Remarks to the Author):

The manuscript by Liu et al. presents the experimental investigation of the coupling between a gold nanoparticle dimer and methylene blue molecules encapsulated in cucurbit[7]uril molecules. The latter are functionalized with carbonyl-fringed portals, enabling them to stick the two nanoparticles together while controlling a gap size of 0.9 nm, and ensure both the precise orientation of the molecule and the coupling of the dimer with a single MB molecule. This method is not new and was also used in Chikkarady et al., *Nature* 535 (2016) to couple a single MB molecule with a nanoparticle-on-mirror (NPoM) construct. The authors of the present manuscript adapted the method to the formation of gold nanoparticle dimers encapsulating one or a few more molecules.

This constitutes another platform to investigate the coupling of few molecules to gap plasmons, with a dimer geometry instead of a NPoM geometry. In a (sadly not cited here) review article by Baumberg et al., *Nature Material* 18, 668 (2019), the following statement was made:

“Coupling nanoparticle plasmons together represents an alternative approach to field localization, as light can be tightly confined to the gaps between nanoparticles. However,

the same limitations in fabrication capabilities also lead to inconsistent control over gap dimensions in nanoparticle clusters”

which, to me, seemed to rule out the investigation of the precise coupling of few molecules with dimer geometry, in favor of the celebrated NPoM structure. In the present manuscript, the authors seem to prove that this can be achieved with the very same method, i.e., controlling the gap size and ensuring that a single molecule is coupled using a CB molecule with host-guest chemistry. This achievement, in my opinion, represents a significant step in the investigation of single-emitter strongly coupled nanodevices, with interesting prospects for room-temperature quantum technologies.

The coupling of a single emitter to a nanoparticle dimer was reported in e.g., the work of Santhosh et al., *Nature Comm.* 7, 11823 (2016), but the authors of the current manuscript relevantly raised the point that multiple excitons could couple to the gap plasmon even though only one quantum dot is localized within the gap. Consequently, the strong coupling of single molecules to plasmons provide another (probably more rigorous) testbed to explore single electronic transitions optically at room temperature. Therefore, even though the current work does not represent the first ever reported strong coupling between a single molecule and a plasmon mode (done earlier with a NPoM structure in Chikkarady et al. (2016)), it would represent the first ever single molecule strong coupling with a nanoparticle dimer, and open further investigations using dimer constructs in the nanophotonic community.

I, however, have some reserve on the results presented here, and I cannot accept the manuscript in the present form as a publication in *Nature Communication*. In the following paragraphs, I detail a few points that, to me, should be successfully addressed in order to provide strong enough evidence that the reported physical interpretations are correct. As far as my knowledge can go, the experimental methods employed by the author are well-documented and rigorously detailed. My comments mostly concern the simulations provided in the text and the supplementary file, as well as the interpretations drawn from them.

1) By varying the size of the nanoparticles and isolating Au dimers/CB[7]@MB hybrids, the authors obtained Fig. 2b(l), showing an anticrossing behaviour and superimposed with the

eigenenergies obtained with Eq. (1). Next, the authors perform a theoretical test to verify that the observed anticrossing indeed corresponds to single-molecule strong coupling and not enhanced absorption or Fano resonance. A finite element method using a full-wave solver is then used to provide extinction, absorption and scattering spectra of a dimer made of two perfect gold spheres. The spectral splitting obtained numerically in the absorption spectrum of the total system and subsystem is then interpreted as an evidence of single-molecule strong coupling. I am wondering why did the authors only confirm this matter theoretically, and why not measure absorption, photoluminescence or SERS spectra of the investigated uncoupled and coupled systems? Only the absorption and PL spectra of MB molecules, CB[7], and CB[7]@single MB molecules are provided, but the corresponding data for the empty dimers as well as the hybridized systems were not provided. Such data, to me, would strengthen the authors statements about the single-molecule strong coupling interpretation extracted (here solely) from scattering spectra. It would have even more significance than the current theoretical investigation.

2) This theoretical investigation using the FEM lacks clarity to me, and as presented, I do not trust it fully:

a. Nowhere in the manuscript or in the supplementary information, I could find what dielectric function was used to model the Au dimer. Was it a Drude model or data from e.g., Johnson and Christy?

b. I came across the comment (5 times counting main text and SI) that an effective index of 1.26 was set above the ITO glass substrate to account for significant surface roughness of the surface. I am not really convinced by this statement and have the feeling that this refractive index is there mostly to adjust the observed resonances with the experimental one. I would gladly like to have more details on where such resonance mismatch could come from. For example, it is well known in the literature that quantum surface effects (see e.g. Yang et al., Nature 576, 248 (2019)) strongly shifts the resonance of plasmonic systems. Could this somehow also explain the resonant wavelength mismatch between observed and numerical data?

c. I also have reserve on the method presented here, consisting in modeling the molecule with a cylinder whose dielectric function is a fitted Lorentz model. The authors mention the oscillator strength $f_0=0.27$. How does this relate to the single molecule dipole moment $\mu_c=0.09$ e.nm, which is roughly the single MB molecule transition dipole moment? Also, even if it may present some difficulties, it is usually preferable to model a molecule as a point dipole with a chosen position and orientation. Here, the cylindrical shape spreads in the gap, even if small and sized as a CB molecule. In terms of numerical simulation, any finite volume geometry whose refractive index differs from the medium's supports confined modes, in addition to the Lorentz resonance in the dielectric function. How do the authors make sure that the supported modes of this cylinder do not interfere in their calculations? To defend the authors, it is true that this method was used in Chikkarady's work, but I am especially careful because I have earlier seen misinterpretations related to hybridized modes interpreted with this method.

d. In the Methods section, I did not understand either why the radii of the cylinders used to model the coupled molecules were 1.1 and 1.4 nm for 2 and 3 molecules in the gap, respectively. This requires some clarifications.

e. The method to extract the coupling components of the absorption in Fig. 3b was also too succinctly detailed.

3) Related to Fig. 3f (but also Fig. 3b), the authors mention a "spectrum broadening instead of a clear SRS of the exciton absorption [...] further demonstrating that the intermediate-strong coupling regime has been reached for the single exciton interacting with the gap plasmons". This statement, as well as the data shown in these figures, weakens the strong statement made in the first sentence of the Discussion section: "we have realized the strong coupling of a single exciton at room temperature in plasmonic nanodimers [...]". Could the authors elaborate a bit more on the intermediate strong coupling regime? Also, what does this imply with respect to the SRS observed in the total absorption?

4) The mode volume defined in Eq. (S1) is valid for Hermitian systems, i.e., closed cavities. For lossy, dispersive and absorbing nanoresonators such as the investigated gold dimers, the

formula for the mode volume is different, see e.g. Wu et al., ACS Photonics 8, 307 (2021). Because such resonators are non-Hermitian, the mode volume is complex and this can result in the Purcell factor for a single molecule not being necessarily maximal on resonance, for instance. An online freeware, which can be found in the following link (<https://doi.org/10.5281/zenodo.7400937>, see also Wu et al., Comp. Phys. Comm. 284, 108627 (2023) for a tutorial), provides matlab codes that can be used together with COMSOL to determine such complex mode volumes. I would highly recommend the authors to determine the mode volumes with the formula from Wu et al., at least to check whether their computed mode volume (formula (S1), which is wrong), is similar with the real part of the complex mode volume. If so, the imaginary part also provides interesting physical content, e.g. a non-Hermitian phase. I understand that this may require some work, and in the defense of the authors, another debatable (yet complex-valued) formula was used in the work of Chikkarady (Eq. (S3) of their SI). I expect at least a significant discussion on the complex mode volume of such structures, and a strong justification of why the authors used Eq. (S1).

5) As a suggestion for improving the manuscript, I would recommend that the authors compute the Purcell factor of a point dipole placed in the center of the gap and oriented along the dimer axis, versus wavelength or frequency (see e.g., (S6) in Chikkarady's work). If the Purcell factor is strongly peaked ($\sim 10^6$) around the resonance of the dimer, that would further bring evidence of the strong coupling regime at the single molecule level, as shown in Rousseaux et al., PRB 98, 045435 (2018). In this work, a Lorentzian fit of the Purcell factor is used to extract the coupling strength, knowing the dipole moment of the emitter. This was done with the authors' theoretical model that they self-cite, with a closed-form expression for the scattering, but not for the Purcell factor formula.

6) I already mentioned quantum surface corrections in comment 2b. Other complexity may arise with e.g., dark plasmonic modes in these structures. For example, many strongly confined modes can appear in NPoM structures (see e.g. Kongsuwan et al., ACS Photonics 7, 463 (2020)). Additionally, it was recently suggested that dark modes, that are usually thought as a dissipative channel that quench the emitter's fluorescence, could result in a pseudomode hybridizing with the emitter's transition and promoting strong coupling

dynamics seen in the spectral Rabi splitting of the bright dipolar mode of a nanosphere (Rousseaux et al., Phys. Rev. Research 2, 033056 (2020)). Since in the present case, the molecule is so tightly confined in such a small metallic gap, what about the coupling to such strongly confined dark modes?

7) Similarly, the authors may be familiar with recent works on picocavities in NPoM geometries, whose main investigator, J. Baumberg, recently published a short review article (Baumberg, Nano Letters 22, 5859 (2022); see also Benz et al. Science 354, 726 (2016)). Could such dimers be used as a platform to investigate the coupling of picocavities, i.e. atomic protrusions coupling to molecules close to the metallic surface?

8) Finally, I would suggest strengthening the Discussion section a little. As far as I understood by reading the paper, the only advantage in the investigation of single-molecule strong coupling with Au dimers instead of NPoM would be that they present (hardly) less dissipation: $Q \sim 14$ while $Q \sim 11-12$ for NPoM, as mentioned by the authors in the text. To me, this seems quite irrelevant. The quality factor is still low, and anyway, the main interest of nanocavities does not lie in the ability to obtain higher Q-factors, owing to large dissipation due to the metal. However, maybe such structure can present other significant interest? I would like to see what the opinion of the authors on this matter is.

In addition, I also suggest the following additions/changes:

1) The statistics presented in Fig. 4d is too succinctly detailed. Please specify the number of sampled structures (which, I guess, corresponds to the one presented in Fig. 4c, but this is not explicitly written), and what the error bars represent (I guess, standard deviation?).

2) In the introduction, the authors mentioned a former work with “a cuboid Au@Ag nanorod coated with dye molecule J-aggregate excitons”. At this stage in the text, for better readability, I would suggest relaxing the use of notations such as Au@Ag and describe the device in a few words instead.

3) On page 4, “CB[7]” was used without being defined before as cucurbit[7]uril molecule.

4) On page 8, I believe the statement “We set $N=1$ and $g_{dc}=56.6$ meV in calculations, and the rest of the parameters were extracted from the experimental measurements” is incorrect, since g_{dc} was extracted with a fit of SRS with Eq. (1). So, the only assumption is $N=1$.

5) In Eq. (1) (and several other times), h should be replaced by \hbar .

6) On page 12, I found the sentence “At the dye resonance, [...] strong coupling” a little too short, just referring to two other works. I think this point is interesting and deserve a little more explained physics. Why is the conductance of the molecule enhanced due to the coupling?

7) In Fig. 4 a and b, the dashed lines, which I assume are Lorentzian fits, are not described.

8) In Fig. 4 d, the green stars do not appear in the legend.

9) In the Methods sections, the word “field” is written “filed” several times.

10) In addition to ref. 45, I would suggest adding a more recent reference, that is even more relevant since it corresponds to TDDFT investigations of particle dimers coupled to single molecules: Kuisma et al., ACS Photonics 9, 1065 (2022).

11) Fig. S12 from the supplementary info seems useless to me, since it is already shown as insets in Fig. 4 and b.

12) Related to Table S2, emitter numbers are shown for WS2 monolayer and J-aggregate. The very idea of defining emitter numbers for excitons in these structures is debated, and I would recommend citing the work of Tserkesis et al., Rep. Prog. Phys. 83, 082401, 2020.

A list of Changes Made in the Revised Version of Manuscript

Chang 1. We have listed Lijun Guo as a co-corresponding author and added Dr. Jindong Ai as a co-first author in the present version to account for their important contributions to the revised manuscript (see the revised lines 3, 4, and 16 on page 1 in the revised manuscript).

Chang 2. We have removed the specific value of the dipole angle estimated in the revised manuscript and revised the expression of this point (see the revised lines 25-30 on page 2; the revised lines 96 and 97 on page 5; the revised lines 198 and 199 on page 10 in the revised manuscript).

Chang 3. We have added one of our recent works in the Introduction section. We also discussed the exciton number estimation in the revised manuscript (see the revised lines 54-58, 60-62 on page 3; the added refs. 23 and 24 in the revised manuscript; and the revised caption of Supplementary Table S1).

Chang 4. We have defined *cucurbit [7]uril molecule* as *CB[7]* before it is used in our revised manuscript (see the revised lines 94 on page 5 in the revised manuscript).

Chang 5. We have added more discussions and references to verify the mention of “*only one MB molecule can be accommodated in one CB[7] molecule*” in the revised manuscript (see the revised lines 115-125 on page 6 in the revised manuscript; and the revised Supplementary Fig. S1).

Chang 6. We have added extra experiments and discussions, as well as references to support our argument of “*the CB[7] molecule can actively bind to the Au atoms on the surface of the nanoparticle through its carbonylfringed portals*” (see the revised lines 129 on page 6; the revised lines 129 on page 6; the revised lines 130-137 on page 7; the added ref. 38 in the revised manuscript, and the added Supplementary Figs. S3-S6 in the current Supplementary Information, SI).

Chang 7. We have calculated the Purcell factor of a point dipole placed in the gap center of the AuND and oriented along the dimer axis, versus wavelength. We have also given a discussion in the revised Section S3 of the revised SI (see the revised lines 158-161 on page 8; the added ref. 40 in the revised manuscript; and the revised Section S3 and the added Fig. S9 in the revised SI).

Chang 8. We have given the dye concentration used in our samples (see the revised lines 204-207 on page 10; the revised lines 433-437 on page 22; and the revised lines 438-449 on page 23 in the revised manuscript).

Chang 9. We have revised the sentence “*We set $N=1$ and $g_{dc}=56.6$ meV in calculations, and the rest of the parameters were extracted from the experimental measurements*” as “*We set*

N=1 in calculations, and the rest of the parameters were extracted from the experimental measurements” (see revised lines 219 and 220 on page 11 in the revised manuscript).

Chang 10. We have added the laser-induced bleaching experiment of the strongly coupled AuNDs with single MB molecules and explained why we have not measured the AuND/CB[7]@single MB coupling systems in PL or absorption spectra (see the revised lines 229-230 on page 11; the revised lines 231-237 on page 12 in the revised manuscript; the added Section S4 and Fig. S13 in the revised SI).

Chang 11. We have added extra bianalyte SERS and time-series SERS spectra of the hybrid AuNDs to further verify single dye molecules in the gap of the AuNDs (see the revised lines 237-251 on page 12; the revised line 1 on page 13; the revised lines 267 and 268 on page 13; the revised lines 269-272 on page 14; the revised lines 388-391 on page 20; the revised lines 477-479 on page 24; the revised lines 480 and 481 on page 25; the added Fig. 2(e, d) and Fig. 5; the added refs. 44-46, 51-54 in the revised manuscript; and the added Fig. S17 in the revised SI).

Chang 12. We have rewritten the part of numerical simulations in our revised manuscript and replaced the field enhancement maps in Fig. 1(e) (see the revised lines 273-280 and 282 on page 14; the revised lines 299-302 and 305-311 on page 15; the revised lines 312 and 313 on page 16; and the revised Fig. 1(e) in the revised manuscript).

Chang 13. We have rewritten and strengthened the Discussion section in the revised manuscript (see the revised lines 369-383 on page 19, the revised lines 384-386 on page 20, and the added Fig. 5 in the revised manuscript).

Chang 14. We have added the percentages of dimers in the sample, dimers in the strong coupling, and the dimers with coupling consistent with one, two, or three MB molecules in the gap (see the revised lines 433-437 on page 22; the revised lines 438-449 on page 23 in the revised manuscript; and the added Supplementary Figs. S19 in the revised SI).

Chang 15. We have added the method to calculate the coupling components of the absorption in Fig. 3b (see the revised lines 504-512 and 516-520 on page 26 in the revised manuscript; and the added part of *simulations of the absorption cross-section spectra for the coupling components* in the revised Section S2 in the revised SI).

Chang 16. We have added the refs. 23, 24, 37-40, 44-46, 48, 50-54, and 56 in the revised manuscript.

Chang 17. We have also added the Figs. S3-S6, S9, S13, S14, and S17-S20 in the revised SI; other revisions to the SI can found in the parts are highlighted in red.

Replies to the Reviewer's Comments

Reviewer #1

In “Deterministic positioning and alignment of a single-molecule exciton in plasmonic nanodimer for strong coupling”, R. Liu et al. report that they observe the strong coupling regime between a single molecular exciton and a plasmonic resonator. Observing this coupling regime is an essential step for the development of quantum devices working at room temperature without the need for costly and energy-intensive cryogenic equipment. However, the manuscript does not provide any unequivocal demonstration that the strong coupling regime is observed with a single exciton and would require extensive extra information, as well as reference and additional experiments before it can be considered for publication. If the authors can provide all the required data to answer the following points, then their report would be of interest to a broad scientific community and could justify publication in Nature Communications.

We deeply appreciate Reviewer #1 for his or her positive comments on the novelty and importance of our manuscript. We have tried our best to provide the required data and to address all the concerns of Reviewer #1 as follows.

Comment 1: *The first issue is that the authors never prove that there is indeed one and only one methylene blue (MB) molecule in the plasmonic gap. They mention that “only one MB molecule can be accommodated in one CB[7] molecule” without proving it: can they justify it with, at the very least, a reference discussing this point? More importantly, they never prove that only one CB[7] is located in the gap. Can they prove unequivocally that it is the case? At most, I believe that the authors can say that their data are consistent with one molecule in strong coupling regime but that this remains a strong hypothesis.*

Our Response: Regarding mentioning “only one MB molecule can be accommodated in one CB[7] molecule”, R. Chikkaraddy et al. have verified with experimental evidence in *Nature* 2016, 535, 127–130. From their experimental results, one can see that the cucurbit[7]uril (CB[7]) is water-soluble and can indeed accommodate only one methylene-blue (MB) molecule inside. Inspired by them, the evidence of encapsulating single MB inside CB[7] in our experiment has also been confirmed by the absorption spectroscopy of MB molecules (Supplementary Fig. S1 in the original Supplementary Information, SI): MB dimers (shown by the left small ‘shoulder’ peak at ~614 nm) largely decrease and even disappear on mixing lower methylene-blue concentrations with CB[7] (the molar ratio of MB and CB[7] was 1:2), and the MB monomer peak (at ~664 nm) has a clear blue shift, Supplementary Fig. S1 and the following Fig. R1]. To

confirm, we have reperformed an additional control experiment with the smaller CB[5] (which MB cannot fit, the dashed red line in Fig. R1). It is found that the experiment cannot remove this shoulder peak (Fig. R1, dashed red line), and the MB monomer peak has not any blue shift, ruling out parasitic binding [*Nature* 2016, 535, 127–130]. In addition, we have also studied the photoluminescence (PL) spectra of the MB molecules and the single MB molecules embedded in CB[7] (Supplementary Fig. S2(a) in the original SI). It was found that the fluorescence intensity of the MB monomers can be largely enhanced by 18 times when accommodated in CB[7] molecules, arising from a dramatic increase of the luminous MB monomers. On the other hand, the lifetime of encapsulated MB monomer in CB[7] was significantly prolonged to 2.11 ± 0.02 ns from 0.48 ± 0.01 ns of the MB molecules in solution (Supplementary Fig. S2(b)), indicating that some of the relaxation processes of the encapsulated MB monomers are severely hindered by the host cavities [*Chem. Eur. J.* 2019, 15, 5215–5219]. All these experimental results indicate that we have indeed placed single MB molecules into CB[7] molecules, which can avoid the aggregation of MB monomers.

Fig. S1 in the original Supplementary Information. Normalized absorption spectra of MB molecules (black line), CB[7]@single MB emitters (red line), and CB[7] in water (blue line). The molar ratio of MB and CB[7] is 1:2.

Fig. R1. Normalized absorption spectra of MB in water, with (solid red) and without (black) encapsulation in cucurbit[*n*]urils of different diameters (dashed and solid red lines). Icons show individual molecules (in orange; line centered at ω_c) and paired molecular dimers (in blue). The molar ratio of MB and CB[*n*] is 1:2.

Fig. S2. (a) Black and green lines are PL spectra of MB molecules in aqueous solution with different concentrations of 1.2×10^{-3} and 4×10^{-4} M, respectively. Pink line: PL spectrum of MB molecules embedded in CB[7]; the concentration of the MB molecules in 4×10^{-4} M. (b) Fluorescence decay profiles for MB molecules (green diamonds) and single MB molecules embedded in CB[7] (red circles) in aqueous solution. The concentrations of the MB molecule in these two cases are all 4×10^{-4} M. The solid green and red curves are the exponential fitted results of the experimental data.

To further prove only one CB[7]@single MB is in the gap, we have adopted the bianalyte method, which has been well-studied and well-proven in chemistry for confirming surface-enhanced Raman scattering (SERS) of single molecules [*Nature* 2016, 535, 127–130; *Nat. Commun.* 2014, 5, 4357; *Phys. Chem. Chem. Phys.* 2011, 13, 4500–4506; *J. Phys. Chem. B* 2006, 110, 1944–1948]. The principle is to use two molecules with well-distinguished Raman spectra and to show that mixtures at low concentrations give signals which arise from one or

the other but not both molecules. Herein, we use proflavine (PF) as a second molecule which is very similar chemically and structurally to MB and binds inside CB[7] in the same way [Nature 2016, 535, 127–130]. At a low dye concentration ($\bar{n} \sim 0.8$ dye molecules, i.e., $\bar{n}_{MB} \sim 0.4$ CB[7]@single MB and $\bar{n}_{PF} \sim 0.4$ CB[7]@single PF per Au nanoparticle (on average), the same dye concentration as we have used in Fig. 2a, b in the manuscript, see the revised Method section), one can see that there is a very low probability that two dye molecules can simultaneously accommodate in the same gap of an Au nanodimer (AuND).

Fig. R2. (a) SERS spectra from single MB and PF molecules collected in individual AuNDs with an integration time of 2s and excitation at 632 nm. (b) Histograms of the relative contribution of MB dye $p_{MB} = I_{MB} / (I_{MB} + I_{PF})$ to the total intensity of MB and PF for $\bar{n}_{MB} = \bar{n}_{PF} = 0.4$ molecules, demonstrating the fraction of the signal contributed by MB. Insets show the signature of the three different probabilities. (c, d) Time-dependent evolution of SERS signals from two strongly coupled AuND/CB[7]@single MB hybrids. (e) Time-dependent evolution of SERS signals from an AuND/CB[7]@single PF hybrid. Each spectrum is collected for 2s at 633 nm, 60 $\mu\text{W}/\mu\text{m}^2$ pump. Insets are the SEM images of the measured hybrid AuNDs.

MB and PF can be easily distinguished in their SERS spectra in the 400-460 cm^{-1} region (Fig. R2(a)). The SERS of this low-concentration 50:50 mixtures are measured. The lowest two SERS bands at $\sim 400\text{-}410$ cm^{-1} for PF and $\sim 460\text{-}470$ cm^{-1} for MB were used to distinguish each molecule alone, allowing us to construct a histogram where the probability of single-molecule

signature is plotted in the following Fig. R2(a, b), and accompanying them are the relevant bianalyte spectra for three different probabilities of single-molecule events (0 corresponds to single PF event, 0.5 for the mixed event and 1 for single MB event) [*Anal. Chem.* 2007, 79, 8411–8415]. The histogram indicates that, at such a low dye concentration, two dye molecules are rarely found simultaneously, and the measure AuND/CB[7]@single MB hybrids in Fig. 2 are truly in the single molecule regime.

On the other hand, we have also recorded the time-series SERS spectra of individual strongly coupled AuND/CB[7]@single MB (PF) hybrids, as shown in Fig. R2(c-e), from which one can see the single-molecule behavior —‘spectral diffusion’ of the vibrational peaks of single dye molecules. Such spectral diffusion is well-studied well-proven evidence of single-molecule events both in the fluorescence community [*Nature* 1991, 349, 225–227] and in the SERS community [*Science* 1997, 275, 1102–1104; *Nature*, 2016, 535,127–130]. The vibrational shifts for different bonds are seen to be correlated but can be in opposite directions, which is only possible to account for if they are from single molecules. All these experimental results have further confirmed the presence of single dye molecules in the gap of the measured hybridized AuNDs.

Corresponding Revision: We have added more discussions and references to verify the mention of “*only one MB molecule can be accommodated in one CB[7] molecule*” in the revised manuscript (see the revised lines 115-125 on page 6 and the revised Supplementary Fig. S1). We have also added extra bianalyte SERS and time-series SERS spectra of the hybrid AuNDs to further verify single dye molecules in the gap of the hybrid AuNDs (see the added Fig. 2(e, d) and Fig. 5; the added refs. 37, 38 44-46, 51-54; the revised lines 237-251 on page 12; the revised lines 267 and 268 on page 13; the revised lines 269-272 on page 14; the revised lines 369-383 on page 19; the revised lines 384-386 on page 20; the revised lines 477- 479 on page 24; and the revised lines 480 and 481 on page 25 in the revised manuscript; and the added Fig. S17 in the revised SI).

Comment 2: *What is the percentage of dimers in strong coupling? What is the percentage of dimers with coupling consistent with one, two or three MB molecules in the gap? What is the percentage of dimers in the sample? There is absolutely zero quantitative information on the sample that is produced and analyzed in the manuscript. The authors must clearly say what is the percentage of single particles, dimers or larger aggregates in their sample. And they must say how many dimers are in strong coupling compared to the number of dimers that do not feature energy splitting with and without MB molecules.*

Our Response: In the sample fabrication process, the number of Au nanoparticles can be calculated as $\sim 5.0 \times 10^{11}$ (see the Method Section in the revised manuscript). Considering the number of the added CB[7]@single MB emitters $\sim 4.0 \times 10^{11}$, there are $\bar{n}_{MB} \sim 0.8$ CB[7]@single MB emitters and $\bar{n}_0 \sim 0.8$ empty CB[7] without MB per Au nanoparticle (the molar ratio of MB and CB[7] is 1:2 in CB[7]@single MB emitters' preparation). Such a low MB concentration facilitates us to obtain the single-MB exciton strong coupling with a single AuND. The morphological characterization experiments show that the percentages of the single Au nanoparticle, nanodimer, and nanoclusters (≥ 3 nanoparticles) are $\sim 62\%$, 23% , and 15% , respectively, after the Au nanoparticles mixed with CB[7]@single MB emitters for 1h (Fig. R3(a), *i.e.*, Fig. S5(a) in the revised SI). On the other hand, with increasing the amount of CB[7]@single MB emitters or the hybridization time, the percentages of the AuNDs and other aggregations can be further improved.

Fig. R3. (a) Representative TEM images of the Au nanoparticles mixed with CB[7]@single MB emitters (at a concentration of $\bar{n}_{MB} \sim 0.8$ CB[7]@single MB emitter per Au nanoparticle) for 1h. (b) Statistical histograms of the single Au nanoparticles, nanodimers, and other aggregations (*i.e.*, clusters, ≥ 3 nanoparticles). (c) Quantum steps of the coupling strength observed from the strongly coupled hybrid AuNDs isolated from the Au nanoparticle ensemble treated with CB[7]@single MB emitters at a concentration of $\bar{n}_{MB} \sim 0.8$ CB[7]@single MB emitter per Au nanoparticle.

We have measured more than 200 individual AuNDs isolated from this sample, and it is found that $\sim 5\%$ of these measured dimers demonstrate strong coupling, and the rest of the measured dimers do not feature energy splitting with and without MB molecules. Compared to that ($\sim 1\%$) commonly achieved at present [*Nat. Commun.* 2018, 9, 4012; *Nano Lett.* 2022, 22, 4686–4693], such a percentage is acceptable. According to the mole ratio of MB and CB[7] of 1:2 in the fabrication of CB[7]@single MB emitters, there are at least half of AuNDs (on average) formed via empty CB[7] without MB molecule inside. On the other hand, there are also some NDs ($\sim 5\%$) formed naturally (the following Fig. R4(a-c) in the reply for **Comment**

3), and there are also some dimers that may have morphological defects which reduce the coupling strength between the gap plasmon mode and exciton. All these factors reduce the percentage of the dimers showing spectral Rabi splitting.

It also should be mentioned that, in these strong-coupling NDs, the NDs featured energy splitting consistent with one MB molecule accounting for ~83.3%, and ~13.3% of these strong coupling dimers demonstrate the two-MB strong coupling (Fig. R3(c)). There was also a very small part (~3.3%) of the strong-coupling samples showing three-MB molecule strong coupling, owing to some cases of the non-uniform adsorption of CB[7]@single MB emitters on the Au nanoparticles. The quantum steps in Fig. R3(c) demonstrate that most of the individual AuND/CB[7]@single MB hybrids are in the single-exciton strong coupling regime for the sample treated with a low dye concentration ($\bar{n}_{MB} \sim 0.8$ CB[7]@single MB emitter per Au nanoparticle).

Corresponding Revision: We have added the percentages of dimers in the sample, dimers in the strong coupling, and the dimers with coupling consistent with one, two or three MB molecules in the gap (see the revised lines 433-437 on page 22; the revised lines 438-449 on page 23; the revised lines 470-476 on page 24 in the revised manuscript; and the added Supplementary Figs. S19 in the revised SI).

Comment 3: *How can the authors ensure that the CB[7] molecule is responsible for the formation of dimers? They mention that “the CB[7] molecule can actively bind to the Au atoms on the surface of the nanoparticle through its carbonylfringed portals” but can they actually prove it? Do they still observe dimers without the CB[7] molecule? When gold particles are dried on substrates, dimers always appear spontaneously: how can we be sure that the CB[7] molecule is indeed in the gap? Furthermore, since they do not control the number of CB[7] molecules per particle, why do they only observe dimers? In Figure S3, they show an image with a lot of dimers but this does not make sense: if they introduce a very limited number of CB[7] molecules, most particles should remain as single particles and only a few dimers should appear; while, if they put a lot of CB[7] molecules, more than one will be found per particle and large aggregates should appear. It is also very strange that the darkfield images in figure S7 are so different with and without the MB molecule: why is there more aggregation with MB molecules than without them? Overall, the authors must perform extra reference measurements by preparing samples with varying concentrations of CB[7] molecules mixed with the gold particles and prove that this concentration controls the level of aggregation observed in the*

final samples, which would be consistent with the fact that the CB[7] molecule acts as a bridge between gold particles.

Our Response: We divide this comment into four parts (I-IV) and address them individually.

Part (I) About the questions of *“How can the authors ensure that the CB[7] molecule is responsible for the formation of dimers? They mention that “the CB[7] molecule can actively bind to the Au atoms on the surface of the nanoparticle through its carbonylfringed portals” but can they actually prove it?”*

The construction of AuND and other aggregations by using CB[n] molecules (through a strong multivalence interaction between CB[n] and gold) is a common approach that has been employed and discussed in previous literature [*Chem. Commun.*, 2008, 17, 1989–1991; *Chem. Commun.*, 2011, 47, 9867–9869; *ACS Nano*, 2011, 5, 3878–3887; *Sci. Rep.* 2014, 4, 6785]. As is well known that CB[n] are molecules with multiple carbonyl groups that could interact with gold and bind the Au nanoparticles together as molecular glue [*Nature*, 2016, 535, 127–130; *Chem. Commun.*, 2008, 17, 1989–1991; *Langmuir* 2003, 19, 6483–6491; *ACS Nano*, 2011, 5, 3878–3887; *Chem. Commun.*, 2011, 47, 9867–9869]. To demonstrate the key role of CB[7] in constructing AuNDs, we have also performed extra reference measurements by preparing samples with varying concentrations of CB[7] mixed with colloidal Au nanoparticles, according to the Reviewer’s good suggestion. The experimental results show that the CB[7] concentration indeed controls the level of particle aggregation observed in the final samples. Figures R4(d-l) demonstrate TEM images of the Au nanoparticles ($\sim 5.0 \times 10^{11}$) mixed with CB[7] (6.66×10^{-8} M, the same CB[7] concentration as we have used in CB[7]@single MB emitter fabrication in our original manuscript) of 10, 50, and 100 μL (corresponding to ~ 0.8 , 4 and 8 CB[7] molecules per Au nanoparticle) for 1, 10, and 24 h respectively. Compared to the yield ($\sim 5\%$) of AuNDs naturally formed in the control sample without CB[7] (see insets in Fig. R4(a-c)), the yields of AuNDs and other aggregations in the samples mixed with CB[7] are significantly improved (see the insets of Figs. R4(d-l)). Additionally, for samples mixed with CB[7] at a fixed concentration, the yields of AuNDs and other aggregations can be improved by increasing the hybridization time. All these results are consistent with the fact that the CB[7] molecule acts as a bridge in forming the AuNDs and other aggregations.

Fig. R4. (a-c) TEM images of Au nanoparticles isolated from the sample without mixing CB[7]@single MB emitters at different times (after the sample was fabricated 1, 10, and 24 h, respectively). (d-l) TEM images of Au nanoparticles isolated from the samples (2 mL, $\sim 5.0 \times 10^{11}$) mixed with CB[7] molecules (6.66×10^{-8} M) with varying volumes of 10, 50, and 100 μ L at different hybridization times (1, 10, and 24 h, respectively). The insets are statistical histograms of the single Au nanoparticles, nanodimers, and other aggregations in these samples (*i.e.*, clusters, ≥ 3 particles). S: Single, D: Dimer, C: Cluster.

Part (II) About the mention of *“the CB[7] molecule can actively bind to the Au atoms on the surface of the nanoparticle through its carbonylfringed portals”* in our manuscript, this conclusion has been verified in previous literature [*Phys. Chem. Chem. Phys.*, 2010, 12, 10429–10433; *Chem. Commun.*, 2008, 17, 1989–1991]. For example, An et al. have carried out the Fourier transform IR reflection absorption spectroscopy (FT-IRRAS) measurements of the self-assembled CB[7] monolayers on a gold surface (the following Fig. R5(a)). The Two characteristic peaks of CB[7] at 1751 cm^{-1} and 1474 cm^{-1} corresponding to C=O and C-N

stretching vibrations (*Chem. Commun.*, 2004, 7, 848-849), respectively, were clearly detected. Compared with the infrared spectrum of CB[7] obtained using bulk KBr pellet (Fig. R5(b)), the more intense peak at 1751 cm^{-1} (amide I) suggests that CB[7] molecules rest with their carbonyl groups perpendicular with respect to the gold surface (the inset in Fig. R5(a)), which is in accordance with the FT-IRRAS surface selective rule [*J. Phys. Chem. A*, 2004, 108, 9673-9681]. In addition, we have also performed extra SERS measurements of the CB[7] molecules embedded in single AuNDs (red line in Fig. R6); it can be seen that the stretch vibration of the C=O ($\sim 1756\text{ cm}^{-1}$) indeed significantly enhanced compared to other SERS peaks and that in the normal Raman spectrum of the CB[7] molecules in powder (black line in Fig. R6), which also indicates that the binding of CB[7] on gold is indeed through its carbonylfringed portals.

[FIGURE REDACTED]

Fig. R5. (a) FT-IRRAS spectrum of the CB[7] monolayer on a gold surface. (b) FTIR spectrum of the CB[7] obtained using bulk KBr pellet. These two figures are taken from the work by An et al. [*Chem. Commun.*, 2008, 17, 1989–1991].

Fig. R6. Red line: SERS spectrum of a CB[7] molecule embedded in the AuND. Black line: Normal Raman spectrum of the CB[7] molecules in powder. The inset is the SEM image of the measure AuND/single CB[7]. The scale bar is 100 nm.

Part (III) About the questions of *“Do they still observe dimers without the CB[7] molecule? When gold particles are dried on substrates, dimers always appear spontaneously: how can we be sure that the CB[7] molecule is indeed in the gap? Furthermore, since they do not control the number of CB[7] molecules per particle, why do they only observe dimers?”*

Indeed, it is inevitable that there are some naturally formed AuNDs without CB[7] molecules can also be observed. Because when the Au nanoparticles dropped and dried on the substrate, there are some AuNDs formed randomly, but their yield was very low ($\sim 5\%$, see the above Fig.

R4(a-c)); thus, the influence of the naturally formed AuNDs on the measurement of AuND/CB[7]@single MB hybrids is limited.

In our original manuscript, we controlled the number of CB[7] molecules per Au nanoparticle, despite not estimating the specific number of CB[7] molecules per particle. According to the amounts of the MB molecules ($\sim 4.0 \times 10^{11}$), CB[7] molecules ($\sim 8.0 \times 10^{11}$, the molar ratio of MB and CB[7] was 1:2), and Au nanoparticles ($\sim 5.0 \times 10^{11}$) used in our fabrication process (see the Methods Section in the revised manuscript), there were $\bar{n}_{MB} \sim 0.8$ CB[7]@single MB emitter and $\bar{n}_0 \sim 0.8$ empty CB[7] (*i.e.*, $\bar{n}_{CB[7]} \sim 1.6$ CB[7] molecules) per Au nanoparticle.

On the other hand, as the Reviewer has pointed out, we cannot guarantee there are only dimers because the trimers and other aggregations can also be formed when the number of CB[7] molecules adsorbed on a nanoparticle is more than one. However, it does not affect our measurements of the AuNDs by combining the dark-field scattering and SEM imaging to determine whether what we have measured are indeed AuNDs (Supplementary Fig. S7 in the original SI).

Fig. S7 in the original Supplementary Information. (a) I: Dark-field scattering images of the measured AuND/CB[7]@single MB hybrids; II and III: the corresponding SEM images of the measured AuND/CB[7]@single MB hybrids; IV: The corresponding scattering measurement result for the AuND/CB[7]@single MB hybrid marked as "A" in I and II. (b) I: Dark-field scattering images of the measured AuND/CB[7] without MB molecules; II and III: the corresponding SEM images of the measured AuND/CB[7] without MB; IV: The corresponding scattering measurement result for the AuND/CB[7] marked as "B" in I and II.

Part (IV) About the question “*It is also very strange that the darkfield images in figure S7 are so different with and without the MB molecule: why is there more aggregation with MB molecules than without them?*”, we argue that the SEM images in Fig. S7(a, b) II (i.e., Fig. S11(a, b) II in the revised SI) only reflect the aggregation situation of the Au nanoparticles deposited in some local regions, which has somewhat randomness. We have rechecked the SEM images of the Au nanoparticles isolated from the two samples and deposited them in other locations on the ITO-coated glass substrate (Figs. R7(a) and R7(b)). These Au nanoparticles come from the same two samples we have used in Figs. S7(a) and S7(b) of the original SI. From Fig. R7(a), it seems that there are more Au aggregations from the sample without MB molecules than with them (Fig. R7(b)). The two samples have a similar overall aggregation because they were treated with the same CB[7] concentration of $\bar{n}_{CB[7]} \sim 1.6$ CB[7] molecules per Au nanoparticle. To give a much clearer proof, we have also reperfomed TEM characterizations of such two samples (one sample without MB molecules (Fig. R7(c) and the other with them (Fig. R7(d)), but they have the same CB[7] concentration, $\bar{n}_{CB[7]} \sim 1.6$ CB[7] molecules per Au nanoparticle), from which one can see that the two samples indeed have a similar aggregation (see the insets in Fig. R7(c, d)). It also indicates that the CB[7] molecule in CB[7]@single MB emitter is key in forming AuNDs and other aggregations.

Fig. R7. (a, b) SEM images of the Au nanoparticles isolated from the samples mixed with CB[7] molecules (a) without and (b) with MB molecules at the same concentration of CB[7], i.e., $\bar{n}_{CB[7]} \sim 1.6$ CB[7] molecules per Au nanoparticle. (c, d) TEM images of the Au nanoparticles

isolated from the samples mixed with CB[7] molecules (c) without and (d) with MB molecules at the same concentration of CB[7], *i.e.*, $\bar{n}_{CB[7]} \sim 1.6$ CB[7] molecules per Au nanoparticle. The insets are statistical histograms of the single Au nanoparticles, nanodimers, and other aggregations (*i.e.*, nanoclusters, ≥ 3 nanoparticles). S: Single, D: Dimer, C: Cluster.

Corresponding Revision: We have added the TEM images for Au nanoparticles mixed with CB[7] in varying concentrations at different hybridization times, the TEM images of the typical gaps formed in individual AuND/CB[7]@single MB hybrids, and the TEM images of the Au nanoparticles mixed with CB[7] molecules and CB[7]@single MB emitters with a same concentration of CB[7]; we also have added extra reference of *Chem. Commun.*, 2008, 17, 1989–1991 to further support our argument of “*the CB[7] molecule can actively bind to the Au atoms on the surface of the nanoparticle through its carbonylfringed portals*” (see the revised lines 115-125 on page 6; the added ref. 38 in the revised manuscript; and the added Supplementary Figs. S3-S6 in the revised SI).

Comment 4: *How can the authors ensure that the observed energy splitting is indeed due to the MB molecule? In past reports on strong coupling (for instance, refs 3, 18 or 33 of the manuscript), the molecules were actively or passively bleached to ensure that the energy splitting disappears after bleaching. Since the authors do not directly prove that the CB[7] molecule is indeed in the gap or that there are actually MB molecules in the dimer, they must absolutely ensure that the energy splitting disappears after bleaching the MB molecules. If they cannot bleach them optically, they could maybe reduce them as the reduced state of MB is colorless. Without this demonstration, it seems impossible for me to ensure that the energy splitting observed by the authors is indeed strong coupling due to MB molecules.*

Our Response: According to the Reviewer’s suggestion, we have carried out the laser-induced bleaching of the strongly coupled nanosystems between the individual AuNDs and single MB molecules. It is well known that when the plasmonic nanoparticle-dye molecule strong coupling systems experienced laser illumination, the bleaching would easily occur in dye molecules [*Phys. Rev. Lett.* 2015, 114, 157401; *ACS Nano* 2021, 15, 14732–14743; *Phys. Rev. Lett.* 2017, 118, 237401]. In our revised manuscript, we utilized this bleaching method to destroy the exciton states in the MB molecule via illumination from a femtosecond laser. It was found that the transparency dips in spectral Rabi splitting (SRS) are remarkably weakened and even completely disappeared under a relatively weak irradiance ($10 \mu\text{W}/\mu\text{m}^2$ for 20 s) by a 514-nm femtosecond laser (Ti: sapphire laser, Mira 900) (Fig. R8). Such an optical bleaching

phenomenon in these strong coupling systems indicates that the observed energy splitting in Fig.2 in our manuscript arises from the MB molecule.

Fig. R8. Scattering spectra of individual AuND/CB[7]@MB hybrids before (I) and after (II) photobleaching induced by laser illumination. Insets show SEM images of the measured hybrid AuNDs.

Corresponding Revision: We have demonstrated the laser-induced bleaching of the strongly coupled AuNDs with single MB molecules (see the revised lines 229 and 230 on page 11; the revised lines 1 on page 12 in the revised manuscript; and the added Section S4 and Fig. S13 in the revised SI).

Comment 5: *The authors perform numerous simulations that are confronted to experiments.*

First of all, these simulations do not prove anything from an experimental point of view and only mean that the experiments might be consistent with simulations of strong coupling with one exciton: it is essential to rewrite the manuscript to make this point clearer. Furthermore, the field enhancement maps that are shown in the manuscript are strongly pixelated: how did the authors ensure convergence of their simulations?

Our Response: According to the Reviewer's suggestion, we have rewritten the manuscript, especially the part of numerical simulations in our revised manuscript. These numerical simulations are important to supplement and verify our experiments, especially those that are difficult to perform directly. For example, the absorption spectra observed from the strongly coupled system and its components are difficult to measure directly in our

AuND/CB[7]@single MB coupling system. From the spectral Rabi splitting observed in the simulated absorption spectra, one can further confirm the ability of the AuND to achieve the single-exciton strong coupling. In addition, these simulated results also provide a theoretical limit to our experimental measurements. We have also recalculated the field enhancement maps using the finite elementary method (FEM, COMSOL Multiphysics 5.4) with the iterative solver of GMRES and the refined tetrahedral mesh in the gap of AuND, which guarantees a good convergence (Fig. R9) and much smoother field distribution map (see the following Fig. R10, *i.e.*, Fig. 1(e) in the revised manuscript).

Fig. R9. The convergence graph for each wavelength in the region of 400-700 nm (at a step of 2 nm) in the FEM simulations using the iterative solver of GMRES, from which one can see a good convergence.

Fig. R10. Simulated EF distribution of the gap plasmon confined in a 0.9-nm gap of the AuND. (I) The upper panel: in-plane vertical EF distribution in the center of the nanogap, with 3 and 5 nm radius marked as red and blue dashed circles, respectively. (II) The lower panel: In-plane parallel EF distribution in the middle of the gap.

Corresponding Revision: We have rewritten the part of numerical simulations in our revised manuscript. We have also recalculated the field enhancement maps using FEM with the iterative solver of GMRES and the refined tetrahedral mesh in the gap of AuND (see the revised lines 273-280 and 290 on page 14; the revised lines 299-302 and 305-311 on page 15; the revised lines 312 and 313 on page 16; the revised Fig. 1(e) in the revised manuscript; and the added part of “*Simulations of the absorption cross-section spectra for the coupling components*” in the revised Section S2).

Comment 6: *There are a number of typos, especially the word “bare” confused with the word “bear” (Fig. 1-f IV written in green and several times in the SI). Please correct.*

Our Response and Corresponding Revision: We have carefully checked and revised these typos (the revised parts are highlighted in red, see the revised Fig. 1f(IV) in the revised manuscript and the revised SI).

Reviewer #2

The field of strong coupling in plasmonic nanosystems has made remarkable progress in recent years, pushing the limits down to the level of a few to single molecules, yet the deterministic positioning of single molecules remains a significant challenge. In this study, the authors aim to address this using guest-host self-assembly with super spherical nanoparticles to deterministically position a single molecule exciton and demonstrate strong coupling. They aim to place a single molecular emitter in the gap centre of an Au nanodimer system and claim the simultaneous achievement of both deterministic molecular positioning and dipole alignment with the mode field. Previous studies have employed similar assembly processes, mostly in systems such as nanoparticle-on-a-mirror geometry. Utilizing dimers for strong coupling could be useful and might find more potential. Nonetheless, there are some serious concerns regarding the claims made about the possibility of single-molecule involvement in strong coupling and the accuracy of the reported coupling strength. Further detailed work is necessary at this stage, as the reported results seem preliminary. Hence, I suggest major revisions.

We deeply appreciate Reviewer#2 for his or her positive comments and constructive suggestions that greatly help improve the quality and presentation of this manuscript. We have carefully addressed all the concerns from Reviewer#2 as follows.

Comment 1: *The authors describe the self-assembly process, but no experimental data is presented to support their assertion of deterministic positioning of molecules. The process of "active" self-assembly and how it necessarily leads to a non-random outcome is unclear. It is quite speculative to assume that an individual CB[7] molecule is holding two large Au nanoparticles without dissociation. The forces required for such a holding would be substantial, given the gap size is approximately 0.9 nm. How are the authors claiming that only one molecule is present in the gap?*

Our Response: It is well known that CB[n] molecules (including CB[7]) have been extensively used as a molecular glue to connect Au nanoparticles through a strong interaction between the CB[n] and gold [*Chem. Commun.* 2008, 17, 1989–1991; *Chem. Commun.* 2011, 47, 9867–9869; *ACS Nano* 2011, 5, 3878–3887]. Just as J. J. Baumberg and his co-workers have pointed out, “the aggregation of gold nanoparticles with CB[n] produces a repeatable, fixed, and rigid interparticle separation of 0.9 nm, and thus such assemblies possess distinct and exquisitely sensitive plasmonics” [*ACS Nano* 2011, 5, 3878–3887]. Since CB[n] are molecules with multiple carbonyl groups, they could interact with gold via each carbonyl group [*Chem. Commun.* 2008, 17, 1989–1991], and therefore the CB[n] molecule can strongly bind and hold

two Au nanoparticles to form a dimer with a fixed and rigid interparticle separation, which equals the thickness of the molecular spacer of CB[n] [*ACS Nano*, 2011, 5, 3878–3887].

Therefore, even a single CB[7] molecule can hold two gold nanoparticles enough to construct a dimer without dissociation, which our TEM characterizations can verify. The following Fig. R11(a) shows the TEM images of the gold nanoparticles mixed with CB[7]@single MB emitters at a low concentration of $\bar{n}_{MB} \sim 0.8$ CB[7]@single MB emitter per Au nanoparticle, enabling us to embed one MB molecule in the gap of an Au nanodimer (AuND). After the colloidal Au nanoparticles were mixed with CB[7]@single MB emitters for 1h, the TEM characterization of this sample was carried out. It can be seen that with the active self-assembly induced by CB[7], AuNDs and other Au clusters (≥ 3 nanoparticles) can be formed, and the dimers (insets) demonstrate a fixed interparticle separation of ~ 0.9 nm (see Fig. R13(e-f) in the following reply for **Comment 3**). **It indicates that the binding between CB[7] and gold is strong enough to hold two Au nanoparticles without dissociation.**

To verify a single MB molecule in the gap of the AuND, we have also performed extra experiments of the two-analyte (MB and Proflavine, PF) surface-enhanced Raman scattering (SERS). This allows us to construct a histogram (Fig. R2(b), *i.e.*, Fig. 2(e) in the revised manuscript), which gives the probability that the two molecular spectra are detected simultaneously in different fractions (*Anal. Chem.* 2007, 79, 8411–8415; *Nature*, 2016, 535,127–130). As clearly evident, at such a low dye concentration ($\bar{n}_{MB} \sim 0.4$ CB[7]@single MB and $\bar{n}_{PF} \sim 0.4$ CB[7]@single PF per Au nanoparticle, the same dye concentration as we have used in Fig. 2a, b in the manuscript, see the revised **Method section**), **two molecules are rarely found at the same time. We are truly in the single-molecule regime.** Although this does not guarantee a direct correlation with single-molecule strong-coupling situations, it does prove the statistical probability of single molecules at this dye concentration.

On the other hand, we have also recorded time-series SERS spectra from the individual hybrid AuND/CB[7]@single MB (or PF) isolated from the sample treated with a low dye concentration of $\bar{n}_{MB} \sim 0.8$ CB[7]@single MB (or PF) per Au nanoparticle the AuNDs, as shown in the above Fig. R2(c-d) (Fig. 5 and Supplementary Fig. S17 in the revised manuscript and Supplementary Information, SI, respectively), from which one can see the single-molecule behavior—‘spectral diffusion’ of the vibrational peaks of single dye molecules [*Nature*, 2016, 535,127–130 (2016)]. Such spectral diffusion is well-studied well-proven evidence of single-molecule events both in the fluorescence community [*Nature* 1991, 349, 225-227 (1991)] and in the SERS community [*Science* 1997, 275, 1102-1104]. The vibrational shifts for different bonds are seen to be correlated but can be in opposite directions, which is only possible to

account for if they are from individual molecules. All these experimental results confirm the presence of single dye molecules in the gap of the measured AuNDs.

Lastly, once a hybrid AuND/CB[7]@single MB has been constructed via the CB[7]@single MB emitter, the single MB molecule (enclosed within CB[7]) can be deterministically positioned at the center of the gap. Its dipole moment can be well aligned along the gap plasmon, which is guaranteed by the unique way of CB[7]-assisted active self-assembly via carbonyl groups on the two sides of CB[7] [*ACS Nano*, 2011, 5, 3878–3887; *Nature*, 2016, 535,127–130].

Fig. R11. Typical TEM image of the Au nanoparticles hybridized with CB[7]@single MB emitters for 1 h. The insets are TEM images of the typical AuNDs constructed via single CB[7] molecules in CB[7]@single MB emitters.

Corresponding Revision: We have added TEM images of Au nanoparticles mixed with CB[7]@single MB emitters and TEM images of the typical gaps formed in individual AuND/CB[7]@single MB hybrids. We have added extra bianalyte SERS and time-series SERS spectra of the hybrid AuNDs to verify further single dye molecules accommodated in the gap of the AuNDs. We have also added extra references to support the assertion of deterministic positioning of molecules in the gap center of the AuNDs (see the revised lines 115-125 on page 6; the revised lines 237-251 on page 12; the revised lines 369-383 on page 19; the revised lines 384-386 on page 20; the added Fig. 2(d, e) and Fig. 5; the added refs. 38 and 39 in the revised manuscript, and the Supplementary Figs. S4(e-h), Fig. S17, and S19(a) in the revised SI).

Comment 2: *When a MB molecule is enclosed within a CB[7] cage, it is very likely for some parts of the MB molecule to extend beyond the hydrophobic portal of CB[7], as MB is lengthier, also seen in Fig.1. This further obstructs the binding of CB[7] to Au nanodimers. This further*

raises the question of how an individual CB[7] can hold not just the MB, but also binds to Au nanoparticles at the same time. Additionally, the C=O functional groups of CB[7], which typically bind to metals, are not readily accessible when MB is present inside, as it is lengthier.

Our Response: It indeed seems that when a MB molecule is enclosed within a CB[7] cage, some parts of the MB molecule will extend beyond the hydrophobic portal of CB[7], as shown in Fig.1(b). When the C=O functional groups of CB[7] bind to metals, it is a strong interaction between multiple carbonyl groups and metal atoms [*Chem. Commun.*, 17, 1989–1991 (2008)], as discussed in the above Reply for **Comment 1**. In this binding process, the MB molecule chain could be bent or compressed to some extent in the gap of the AuND, making the MB inside CB[7] cage cannot hamper the access of C=O functional groups to the surface of the metal and the formation of the AuND with repeatable and fixed interparticle separation of 0.9 nm [*ACS Nano*, 2011, 5, 3878–3887]. In reply to the following **Comment 3**, we show the typical TEM images and the statistics of gap distance observed in individual AuND/CB[7]@single MB hybrids, from which one can see that the CB[7]@single MB emitters can effectively bind to gold and hold two Au nanoparticles to form the hybrid NDs with a fixed gap distance of about 0.9 nm, equaling to the thickness of the CB[7] molecule. It also means that the MB molecule in CB[7] cannot hamper the access of C=O functional groups to the surface of the gold.

Comment 3: *In addition, despite the nanoparticles being spherical, the assembly process could result in dimers having a small facet size, as evident from the SEM images provided in the supplementary information. This suggests that there may be multiple CB[7]-MB systems present in the gap, potentially contributing to the observed strong coupling.*

Our Response: We acknowledge that this question is an important issue in our manuscript. Although the small facet of the AuNDs makes the possibility of multiple CB[7]@single MB emitters in the gap higher than that of the ND constructed by perfect spherical Au nanoparticles, such a small facet plays a very limited effect on our single-molecule exciton strong coupling conclusions, especially at low dye concentrations.

In sample fabrication, the number of Au nanoparticles was $\sim 5.0 \times 10^{11}$. By considering the number ($\sim 4.0 \times 10^{11}$) of CB[7]@single MB emitters added in these colloidal Au nanoparticles, there was $\bar{n}_{MB} \sim 0.8$ CB[7]@single MB emitter and $\bar{n}_0 \sim 0.8$ empty CB[7] molecule per Au nanoparticle (the molar ratio of MB and CB[7] was 1:2, a half of CB[7] molecules were empty without MB, see Methods Section in the revised manuscript). Here, we take $\bar{n}_{MB} = 1$ and $\bar{n}_0 = 1$ for an example, i.e., there are 2 CB[7]@single MB emitters and 2 empty CB[7] molecules on average can be bound to an AuND, one can estimate the possibilities for

one and two CB[7]@single MB emitters embedded in the gap of the ND are ~50% (the nanodimer is formed due to CB[7] binding, either through the CB[7]@single MB emitter or the empty CB[7]) and 0.28%, respectively. In specific, considering the cross-section area ($\sim 1.8 \text{ nm}^2$) of the CB[7] and the surface area ($\sim 5024 \text{ nm}^2$) of an Au nanoparticle with a diameter of 40 nm, the probability of two CB[7]@single MB emitters simultaneously accommodated in a same gap (i.e., located in the gap region with a cross-sectional area of $28.3 \text{ nm}^2 = 3.14 \times 3^2 \text{ nm}^2$, where has almost the same most intensive electric fields, Fig. 1(e) in the revised manuscript) can be estimated as $\sim 0.28\%$ according to probability calculation (Fig. R12(a)). Assuming that two small facets construct the nanogap in an AuND, each facet is 10 nm in diameter (i.e., $3.14 \times 5^2 = 78.5 \text{ nm}^2$ in area, Fig. R12(b)), the possibility for two CB[7]@single MB emitters simultaneously embedded in this gap will be improved to $\sim 0.78\%$. However, it remains very small. Therefore, such a small facet does not affect our single molecule results, especially the samples treated with dye molecules at low concentrations. If we increase the dye concentration, the possibility of multiple CB[7]@single MB emitters embedded in the same gap will improve correspondingly. Our manuscript also demonstrated the two- and three-MB strong coupling by increasing the added CB[7]@single MB emitters in the sample fabrication process (see Fig. 4 and the Method section in the revised manuscript).

Fig. R12. (a) Schematic of CB[7]@single MB emitters and empty CB[7] molecules bound on an AuND constructed by two perfect Au spherical nanoparticles. The possibility for two CB[7]@single MB emitters embedded in the gap of the nanodimer is $p = 50\% \times (C_3^1 \times 1/3 \times 28.3/5024) \approx 0.28\%$. (b) Schematic of CB[7]@single MB emitters and empty CB[7] molecules bound on an AuND constructed by two Au nanoparticles with small facets. The possibility for two CB[7]@single MB emitters embedded in the gap of the AuND is $p = 50\% \times (C_3^1 \times 1/3 \times 78.5/5024) \approx 0.78\%$. It can be seen that the small facets of Au nanoparticles have a very limited influence on the single molecule accommodation in the nanodimer.

We have noticed this issue in our experiments. In sample preparation, we have managed to fabricate the Au nanoparticles with a super spherical shape by using the method of oxidation etching (Fig. R13(a-d)) [*Nano Lett.* 2015, 15, 1012–1017]. Figure R13 (e-h) shows that the gaps constructed by these super spherical Au nanoparticles have good smoothness. Although a small part of AuNDs has nanogaps with small-size facets, it does not significantly influence the single molecule results in our manuscript due to the low dye concentration we have used.

Fig. R13. (a-d) TEM images of the super spherical Au nanoparticles with different diameters of (a) 27.0 ± 2.2 nm, (b) 38.4 ± 1.8 nm, (c) 45.4 ± 0.8 nm and (d) 57.0 ± 2.8 nm, respectively. (e-h) TEM images of the representative AuNDs with a fixed gap of about 0.9 nm are constructed by super spherical Au nanoparticles via individual CB[7]@single MB emitters.

On the other hand, to further verify the single dye molecule embedded in the gap of the AuND, we have also performed extra experiments of the two-analyte SERS and time-series SERS signals from the individual AuND/CB[7]@single MB (or PF) hybrids, just as we have replied for the **Comment 1**. These extra experiments show that the single-molecule strong coupling in our AuND/CB[7]@single MB hybrids can be guaranteed at a low concentration of dye molecules.

Corresponding Revision: We have added the possibility estimation for two CB[7]@single MB emitters simultaneously embedded in the gap of single AuNDs (with and without small facets) at a low dye concentration and TEM images for the super spherical Au nanoparticles with different sizes as well as the typical AuNDs constructed by these spherical Au nanoparticles are provided (see the added Supplementary Figs. S4 and S14 in the revised SI).

Comment 4: *Detailed quantification of the yield of formation of dimers and how many of those demonstrate strong coupling is required. Further, the variability in coupling strength among these dimer systems should be thoroughly studied to test and validate the number of molecules contributing to the strong coupling.*

Our Response: We have counted the yield of the AuNDs in the samples employed in our manuscript. The above Fig. R3(a, b) show that the yield of AuNDs is about 23% after the hybridization of CB[7]@single MB emitters and Au nanoparticles for 1h at a low dye concentration of $n \sim 0.8$ CB[7]@single MB emitter per Au nanoparticle. Aside from AuNDs, single Au nanoparticles and nanoclusters (≥ 3 nanoparticles) can also be seen with percentages of about 62% and 15%, respectively. In addition, the yields of the AuNDs and nanoclusters can be improved by increasing CB[7] concentration or hybridization time.

We have also investigated the variability in coupling strength among the measured AuNDs. For the sample used to detect the single-molecule exciton strong coupling ($\bar{n}_{MB} \sim 0.8$ CB[7]@single MB emitter per Au nanoparticle), we measured more than 200 individual AuNDs isolated from this sample. There were $\sim 15\%$ of these dimers that demonstrated strong coupling (featured energy splitting), and the rest of the nanodimers did not feature energy splitting with and without MB molecules. Compared to that ($\sim 1\%$) reported in the previous literature [*Nat. Commun.* 2018, 9, 4012, *Nano Lett.* 2022, 22, 4686], such a percentage is understandable and acceptable. According to the mole ratio of MB and CB[7] of 1:2 in the fabrication process of CB[7]@single MB emitters, there were at least half of AuNDs (on average) formed via the empty CB[7] without MB molecule inside. On the other hand, there are some AuNDs ($\sim 5\%$) that formed naturally without CB[7]@single MB emitters (see the above Fig. R4(a-c)), and other NDs may have some defects which reduce the coupling strength between the dye molecule and plasmons. All these factors further reduced the yield of the AuNDs showing strong coupling. It also should be mentioned that, in these strong-coupling nanodimers, the AuNDs featured energy splitting consistent with one MB molecule accounting for $\sim 83.3\%$, and $\sim 13.3\%$ of these strong coupling dimers demonstrate the two-MB strong coupling. There is also a small part ($\sim 3.3\%$) of the measured nanodimers show three-MB molecule strong coupling owing to the non-uniform adsorption of CB[7]@single MB emitters to Au nanoparticles in some cases (see the above Fig. R3(c)). **The quantum steps in Fig. R3(c) demonstrate that most strongly coupled AuNDs are in the single-molecule exciton strong coupling regime.**

Corresponding Revision: We have clarified the yield of AuNDs, and those demonstrate strong coupling. In addition, the variability in coupling strength among the measured NDs has also

been counted (see the revised lines 433-437 on page 22; the revised lines 438-449 on page 23 in the revised manuscript; and the added Supplementary Fig. S19 in the revised SI).

Comment 5: *Furthermore, the authors' use of coupling strength to calculate the dipole angle is unrealistic and does not accurately reflect the experimental conditions. This is due to several assumptions made, including the number of emitters, the estimation of mode volume, and the dipole strength of an emitter near a metal.*

Our Response: According to the Reviewer's suggestion, we have removed the specific value of the dipole angle estimated in the revised manuscript. **The perfect alignment of the dipole moment with the gap plasmon in our hybrid nanostructure can be guaranteed by CB[7] mediated guest-host self-assembly, which has been reported in the previous literature** [*Nature* 2016, 535, 127–130; *ACS Nano* 2011, 5, 3878–3887; *Nano Lett.* 2012, 12, 5924–5928; *Nano Lett.* 2013, 13, 5985–5990].

Just as the Reviewer has pointed out, utilizing AuNDs for the single emitter strong coupling could be more useful and might find more potential. Yet, the deterministic positioning of single molecules remains a significant challenge. **This manuscript attempts to address this challenge by using the guest-host self-assembly with super spherical nanoparticles to position a single molecule and demonstrate strong coupling deterministically.** Regarding the questions raised by Reviewer#2, we reply as follows.

We have discussed the number of emitters in the single AuND in the reply for **Comments 1 and 3**. According to our extra experiments of two-analyte SERS and time-series SERS spectra of the individual AuND/CB[7]@MB hybrids, we further verified the experimental results in our manuscript are indeed in the single molecule regime, especially when the sample was fabricated at low dye concentrations. Therefore, the number of emitters in our manuscript is not just an assumption. Once the single MB molecule in the AuND can be determined, one can estimate the angle of μ_c based on the single-molecule coupling strength with plasmons.

Indeed, the determination of coupling strength depends on the accurate calculation of the mode volume and the dipole strength of an emitter in close proximity to a metal. We have employed the common approach in our manuscript for mode volume calculation, which has been extensively used in the previous literature [*Nature* 2016, 535, 127-130; *Phys. Rev. Lett.* 2015, 114, 157401; *Phys. Lett. A* 2022, 299, 309-312; *Phys. Rev. Lett.* 2017, 118, 237401; *Phys. Rev. Lett.* 2023, 130, 143601]. To further verify the accuracy of our calculations, we have also calculated the Purcell factor of a point dipole in the gap center of the AuND (Fig. R14) using the following formula [*Phys. Rev. B* 2018, 98, 045435],

$$F_p(\omega) = \frac{\rho_n(\mathbf{r}_E, \omega)}{\rho_n^0(\mathbf{r}_E, \omega)}, \quad (\text{R1})$$

where $\rho_n(\mathbf{r}_E, \omega)$ and $\rho_n^0(\mathbf{r}_E, \omega)$ is the local density of optical states (LDOS) in the gap center of the AuND and free space, respectively. Our calculations show that the Purcell factor in the gap center of the AuND is up to $\sim 3.2 \times 10^6$ at the resonant wavelength of 659.8 nm (resonant to the exciton absorption at ~ 660 nm). Based on this value, we calculated the mode volume of the AuND via the following formula with $Q \sim 12.96$,

$$F_p = \frac{3}{4\pi^2} \frac{Q}{V_m} \left(\frac{\lambda_0}{n} \right)^3 \quad (\text{R2})$$

$V_m \sim 29 \text{ nm}^3$, agreeing with that ($\sim 26 \text{ nm}^3$, Supplementary Table S2 in the revised SI), we have calculated using Eq. (S1) in the Supplementary Materials. **Notice that Eq. (S1) is more suitable for the case of plane wave excitation, just as we have used in our manuscript.** Therefore, the mode volume of the AuND calculated in our manuscript is accurate and can be used to calculate the coupling strength.

Fig. R14. Purcell factor simulation of the AuND constructed by a single CB[7] molecule (gap = 0.9 nm), with a classical emitter placed in the center of the gap at the position of maximum field and spontaneous emission rate plotted as a function of wavelength.

Lastly, we indeed use a dipole strength of the MB molecule according to the work of K. Patil et al. [*Phys. Chem. Chem. Phys.*, 2000, 2, 4313–4317], which can be used to reproduce the experimental results observed in our AuND/CB[7]@single MB hybrids. Thus, we argue that the dipole strength of the MB molecule used in our manuscript is reasonable. Undeniably, the mode volume and the corresponding coupling strength (g_{dc}) are calculated physical quantities that cannot be measured directly in the experiment. Maybe, as the Reviewer pointed out, the

dipole angle calculated based on these physical quantities does not accurately reflect the experimental conditions. **To be more rigorous, we have removed the specific value of the dipole angle estimated in the revised manuscript. It should be mentioned that the perfect alignment of the transition dipole moment with the gap plasmon can be guaranteed using the CB[n] mediated guest-host self-assembly [Nature 2016, 535, 127–130].**

Corresponding Revision: We have removed the specific value of the dipole angle estimated in our revised manuscript; we have also recalculated the mode volume using the Purcell factor and reperformed experiments to verify the single-molecule scenario in our AuND/CB[7] coupling systems (see the revised line 25 and 26 on page 2; the revised lines 96 and 97 on page 5; the revised lines 197-199 on page 10; the added Fig. 2(d, e), Fig. 5 in the revised manuscript; and the revised Section S3; the added Figs. S9 and S17 in the revised SI;).

Comment 6: *The authors claim that the coupling strength value corresponds to a single exciton, but the SEM images in Fig. 2a only suggest a small facet size, which again suggests the potential presence of multiple CB-MB constructs in the gap. Moreover, the angle is calculated based on theoretical assumptions assuming a single molecule in the gap, which is unrealistic and potentially unreliable since the interpretation has already presupposed the single-molecule scenario.*

Our Response: As discussed in the above reply for **Comment 3**, the small facets of Au nanoparticles have a very limited influence on the single molecule accommodation in single AuNDs when we use a low dye concentration. In addition, we have also performed extra experiments on the two-analyte SERS and time-dependent evolution of SERS signals collected from individual AuND/CB[7]@single MB hybrids, which indicates that at a low dye concentration, two molecules are rarely found at the same time, indicating that we are truly in the single-molecule regime. Therefore, the single-molecule scenario in our manuscript can be determined, which is not just a presupposition. After the single dye molecule in the AuND was determined, one can estimate the angle of μ_c based on this single-molecule scenario, the calculated coupling strength, and the observed spectral Rabi splitting (SRS). **Even so, combined with the reply in comment 5 above, we have removed the specific value of the dipole angle estimated in our revised manuscript.**

Comment 7: *Lastly, the authors rely on single-exciton scattering spectral results with different detuning values to validate the findings. However, recent studies (<https://doi.org/10.1038/s41467-018-06450-4>, <https://doi.org/10.1021/acsphotonics.9b01079>) have shown that scattering spectra tend to overestimate the values and therefore, cannot be*

used to accurately validate whether the system is indeed in the single-exciton regime. Thus, further PL-based experiments are required to verify and corroborate the reported results.

Our Response: The single MB molecules embedded in the AuNDs are very easy to be photobleached under the laser exposition (see the above Fig. R8), even at a relatively weak excitation ($10 \mu\text{W}/\mu\text{m}^2$) and a short duration time (20 s). Such a bleaching approach is usually employed to verify the energy splitting observed in the hybrid system is indeed strong coupling due to the dye molecules [*Phys. Rev. Lett.* 2015, 114, 157401; *ACS Nano* 2021, 15, 14732–14743; *Phys. Rev. Lett.* 2017, 118, 237401].

In addition, for single MB molecules accommodated in the sub-nanometer gap of AuNDs and attached to the gold surface, the non-radiative quenching of the dye molecules will be drastic [*Phys. Rev. Lett.* 2002, 89, 203002; *ACS Nano* 2011, 5, 5823–5829], also increasing difficulties in observing the strong-coupling signals through the PL measurements. On the other hand, due to the extremely enhanced electric field intensity in the sub-nanometer gap, the PL measurements also generate strong surface-enhanced Raman scattering (SERS)—consisting of sharp lines with a fluorescence background—that cannot be uniquely separated from photoluminescence [*Nature* 2016, 535, 127–130]. **Therefore, observing the energy level splitting through PL measurements is unrealistic in our single-molecule strong coupling systems.** So, we have turned to conducting experiments on measuring the SERS spectra of such single-molecule coupled systems (see the above Fig. R1) to strengthen our conclusions.

Indeed, plasmon-quantum dot (by H. Leng et al. <https://doi.org/10.1038/s41467-018-06450-4>) and plasmon-J aggregates (formed by many dye molecules, by M. Wersäll, et al. <https://doi.org/10.1021/acsp Photonics.9b01079>) strong coupling systems are more stable in PL measurement than our single-molecule strong coupling system. H. Leng et al. and M. Wersäll, et al. have also shown that scattering spectra tend to overestimate the values than the PL spectra in their studies. **We think that the differences in SRS observed from the PL and scattering mainly arise from the coupling signals coming from different channels (coupling subsystems) with different damping** [*Phys. Rev. B* 2021, 103, 235430]. For example, the SRS of the plasmon-exciton coupling system observed in scattering mainly comes from the plasmonic channel, while the SRS signals observed in PL are mainly from the exciton channel. **The origin of the PL spectral profiles of those plasmonic strong coupling systems remains ambiguous.** For instance, the PL from the excitons not coupling with the plasmonic structure can also contribute to the detected PL signal. **Due to the complexity of the PL emission process, it is difficult to extract the strong coupling information from PL spectra** [*Phys. Rev. Lett.* 2022, 128, 167402]. Recently, H. Wei and her co-workers have proved that the scattering and PL spectra for the strong plasmon-exciton coupling are unified if one can detect

the optical responses from the same channel and avoid the influence of emission from the uncoupled excitons [Phys. Rev. Lett. 2022, 128, 167402]. Even though we cannot separate the plasmonic channel (AuND in the hybrid system) to experimentally detect the scattering singles in the AuND/CB[7]@single MB coupling system, the single molecule contributes very little to the scattering compared to the plasmonic channel. Therefore, the SRS in the scattering of the AuND mainly comes from the plasmonic channel.

Fig. R15. The calculated LS and SRS observed in scattering and emission of the AuND/CB[7]@single MB hybrid, respectively. The LS is calculated according to Eq. (3) in [Phys. Rev. Lett. 2023, 130, 143601]; the SRS in scattering is calculated according to Eq. (1) in our manuscript; the SRS in emission is calculated $\hbar\Omega_R^{em} = 2\sqrt{g_{dc}^2 - (\Gamma_d^2 + \Gamma_c^2)}/8$ in [Phys. Rev. A 2006, 73, 053807]. In calculation, $\Gamma_d = 135$ meV and $\Gamma_c = 85$ meV, and $g_{dc} = 56.6$ meV.

Fig. R15 gives the level splitting (LS, black line) and SRS observed in scattering from the plasmon channel (blue line) and in emission from the exciton channel (red line). **Compared to the LS (Ω_{LS}), the SRS observed in the scattering of AuND/CB[7]@single MB is only overestimated ~6% at the condition of $g_{dc} = 56.6$ meV and $\Gamma_d/\Gamma_c = 135/85 \approx 1.59$ (as we used in the manuscript). Even so, the interaction is in the strong coupling regime because the LS occurs and the LS is larger than the damping of the hybrid system ($\hbar\Omega_{LS} > (\Gamma_d + \Gamma_c)/2$, i.e., $g_{dc}^2 > (\Gamma_d^2 + \Gamma_c^2)/8$ in the manuscript). However, very small and even no SRS will be observed in emission from the exciton channel because it is much smaller than LS in the coupled plasmon-exciton systems. From the perspective of precise measurement of LS, the means of scattering is more suitable to detect the strong coupling that occurred in our AuND/CB[7]@single MB hybrids.**

Corresponding Revision: We have explained the difficulty in PL measurement of our AuND/CB[7]@single MB hybrids with single dye molecules embedded in sub-nanometer gaps.

As an alternative, we have performed extra measurements of bianalyte SERS and time-series SERS spectra for individual AuND/CB[7]@single MB hybrids to verify further the single molecule strong coupling reported in our manuscript (see the revised lines 229 and 230 on page 11; the revised lines 231 and 237 on page 12; and the reply to the above **Comment 1**).

Reviewer #3

The manuscript by Liu et al. presents the experimental investigation of the coupling between a gold nanoparticle dimer and methylene blue molecules encapsulated in cucurbit[7]uril molecules. The latter are functionalized with carbonyl-fringed portals, enabling them to stick the two nanoparticles together while controlling a gap size of 0.9 nm, and ensure both the precise orientation of the molecule and the coupling of the dimer with a single MB molecule. This method is not new and was also used in Chikkarady et al., Nature 535 (2016) to couple a single MB molecule with a nanoparticle-on-mirror (NPoM) construct. The authors of the present manuscript adapted the method to the formation of gold nanoparticle dimers encapsulating one or a few more molecules. This constitutes another platform to investigate the coupling of few molecules to gap plasmons, with a dimer geometry instead of a NPoM geometry. In a (sadly not cited here) review article by Baumberg et al., Nature Material 18, 668 (2019), the following statement was made: “Coupling nanoparticle plasmons together represents an alternative approach to field localization, as light can be tightly confined to the gaps between nanoparticles. However, the same limitations in fabrication capabilities also lead to inconsistent control over gap dimensions in nanoparticle clusters” which, to me, seemed to rule out the investigation of the precise coupling of few molecules with dimer geometry, in favor of the celebrated NPoM structure. In the present manuscript, the authors seem to prove that this can be achieved with the very same method, i.e., controlling the gap size and ensuring that a single molecule is coupled using a CB molecule with host-guest chemistry. This achievement, in my opinion, represents a significant step in the investigation of single-emitter strongly coupled nanodevices, with interesting prospects for room-temperature quantum technologies. The coupling of a single emitter to a nanoparticle dimer was reported in e.g., the work of Santhosh et al., Nature Comm. 7, 11823 (2016), but the authors of the current manuscript relevantly raised the point that multiple excitons could couple to the gap plasmon even though only one quantum dot is localized within the gap. Consequently, the strong coupling of single molecules to plasmons provide another (probably more rigorous) testbed to explore single electronic transitions optically at room temperature. Therefore, even though the current work does not represent the first ever reported strong coupling between a single molecule and a plasmon mode (done earlier with a NPoM structure in Chikkarady et al. (2016)), it would represent the first ever single molecule strong coupling with a nanoparticle dimer, and open further investigations using dimer constructs in the nanophotonic community.

I, however, have some reserve on the results presented here, and I cannot accept the manuscript in the present form as a publication in Nature Communication. In the following paragraphs, I detail a few points that, to me, should be successfully addressed in order to

provide strong enough evidence that the reported physical interpretations are correct. As far as my knowledge can go, the experimental methods employed by the author are well-documented and rigorously detailed. My comments mostly concern the simulations provided in the text and the supplementary file, as well as the interpretations drawn from them.

We deeply appreciate Reviewer #3 for his or her positive comments on the novelty and importance of our manuscript. We also appreciate the Reviewer's constructive suggestions to help us greatly improve the technical quality and the presentation of this manuscript. We have carefully addressed all the concerns from Reviewer #3 as follows.

Comment 1: *By varying the size of the nanoparticles and isolating Au dimers/CB[7]@MB hybrids, the authors obtained Fig. 2b(I), showing an anticrossing behaviour and superimposed with the eigenenergies obtained with Eq. (1). Next, the authors perform a theoretical test to verify that the observed anticrossing indeed corresponds to single-molecule strong coupling and not enhanced absorption or Fano resonance. A finite element method using a full-wave solver is then used to provide extinction, absorption and scattering spectra of a dimer made of two perfect gold spheres. The spectral splitting obtained numerically in the absorption spectrum of the total system and subsystem is then interpreted as an evidence of single-molecule strong coupling. I am wondering why did the authors only confirm this matter theoretically, and why not measure absorption, photoluminescence or SERS spectra of the investigated uncoupled and coupled systems? Only the absorption and PL spectra of MB molecules, CB[7], and CB[7]@single MB molecules are provided, but the corresponding data for the empty dimers as well as the hybridized systems were not provided. Such data, to me, would strengthen the authors statements about the single-molecule strong coupling interpretation extracted (here solely) from scattering spectra. It would have even more significance than the current theoretical investigation.*

Our Response: According to the Reviewer's suggestion, we have conducted experiments on measuring the surface-enhanced Raman scattering (SERS) for the single-molecule coupled systems to strengthen our conclusions. In the above Fig. R2(a, b), we have demonstrated the experimental results of two-analyte (MB and Proflavine, FP) SERS in the individual coupled Au nanodimers (AuNDs). This allows us to construct a histogram (Fig. R2(b), *i.e.*, Fig. 2(e) in the revised manuscript) which gives the probability that the two molecular spectra are detected simultaneously in different fractions [*Anal. Chem.* 2007, 79, 8411–8415; *Nature*, 2016, 535,127–130] (Fig. 2(d, e) in the revised manuscript). As clearly evident, at a low dye concentration ($\bar{n}_{MB} \sim 0.4$ CB[7]@single MB and $\bar{n}_{PF} \sim 0.4$ CB[7]@single PF per Au nanoparticle, the same dye concentration as we have used in Fig. 2(a, b) in the manuscript, see

the revised Method section), a very low probability can be seen that two dye molecules are simultaneously embedded in the same gap of an AuND. Although this does not guarantee a direct correlation with single-molecule strong-coupling situations, it does prove the statistical probability of single molecules at this concentration.

On the other hand, we have also recorded the time-series SERS spectra from the individual strongly coupled AuND/CB[7]@single MB isolated from the sample treated with a low dye concentration of $\bar{n}_{MB} = 0.8$ CB[7]@single MB per Au nanoparticle, as shown in the above Fig. R2(c-e) (Fig. 5 and Supplementary Fig. S17(a, b) in the revised manuscript), from which the single-molecule behavior ---“spectral diffusion” of the vibrational peaks of single MB molecules can be observed. Such spectral diffusion is well-studied well-proven evidence of single-molecule events both in the fluorescence community [*Nature* 349, 1991, 225-227] and the SERS community [*Science* 1997, 275, 1102-1104; *Nature*, 2016, 535, 127-130]. The vibrational shifts for different bonds are seen to be correlated but can be in opposite directions, which is only possible to account for if they are from single molecules. All these measurements further confirm the presence of single dye molecules in the hybrid AuNDs, which leads to strong coupling at room temperature.

It should be mentioned that we have not performed the experiments of measuring PL and absorption in our manuscript. It is well known that dye molecules (especially only a few) in the plasmonic nanocavities are very easy to photodegrade under laser illumination, even at a relatively weak excitation [*Phys. Rev. Lett.* 2015, 114, 157401; *Phys. Rev. Lett.* 2017, 118, 237401; *ACS Nano* 2021, 15, 14732-14743; *Phys. Rev. Lett.* 2023, 130, 143601]. Just as we have demonstrated in the above Fig. R8, the single MB molecules in the hybrid AuND/CB[7]@single MB samples are very easy to be bleached, making such single-molecule exciton coupling systems not suitable for performing the PL measurements. In addition, for single MB molecules embedded in the sub-nanometer gap of AuNDs and attached to the gold surface, the non-radiative quenching of the dye molecules will be drastic [*Phys. Rev. Lett.* 2011, 89, 203002; *ACS Nano* 2011, 5, 5823-5829], also increasing the difficulties in observing the strong-coupling signals from the PL measurements from such hybrid systems. On the other hand, due to the extremely enhanced electric field intensity in the nanogap, the PL measurements also generate strong surface-enhanced resonant Raman scattering (SERS)—consisting of sharp lines with a fluorescence background—that cannot be uniquely separated from photoluminescence [*Nature* 2016, 535, 127-130].

About the question of absorption measuring for the individual AuND/CB[7]@single MB hybrids, to the best of our knowledge, such single-nanoparticle absorption measurement has remained a challenge in experiments because the incident light is always scattered by

nanoparticles—for example, Zengin *et al.* [*Sci. Rep.* 2013, 3, 3074] have attempted to measure the absorption spectra of single hybrid nanoparticles for the first time to provide further evidence of strong coupling in their plasmon-exciton interacting systems. They have measured single plasmonic nanoparticles hybridized with J-aggregates using $100 \times \text{NA} = 1.3$ oil immersion objective; while the measured extinction spectra show poor signal-to-noise ratios and the relative contribution of absorption to the measured extinction is only $\sim 1/3$, the other $2/3$ of the measured signal is still from scattering. At present, we are not able to perform such single-nanoparticle absorption measurements based on our experimental setup. Therefore, we have simulated these absorption spectra of our plasmon-single molecule hybrid system to strengthen our statements (Fig. 3 in our Manuscript).

Corresponding Revision: We have performed extra measurements of bianalyte SERS and time-series SERS spectra for individual AuND/CB[7]@single MB hybrids to verify further the single-molecule strong coupling reported in our manuscript. We have also added an explanation about the difficulty in absorption and PL measurements of our AuND/CB[7]@single MB hybrids with single dye molecules embedded in sub-nanometer gaps (see the revised lines 229 and 230 on page 11; the revised lines 231-251 on page 12; the revised lines 369-383 on page 19; the added ref. 48; the added Fig. 2(d, e) and Fig. 5 in the revised manuscript; and the added Supplementary Fig. S17 in the revised Supplementary Information, SI).

Comment 2: *This theoretical investigation using the FEM lacks clarity to me, and as presented, I do not trust it fully:*

a. Nowhere in the manuscript or in the supplementary information I could find what dielectric function was used to model the Au dimer. Was it a Drude model or data from e.g., Johnson and Christy?

Our Response: We have clarified the dielectric function used for modeling the Au dimer in the revised manuscript. The dielectric function we have used for modeling the Au dimer is the Brendel-Bormann model.

Corresponding Revision: The revised manuscript clarifies the dielectric function for modeling the Au dimer (see revised lines 510 and 511 on page 26).

b. I came across the comment (5 times counting main text and SI) that an effective index of 1.26 was set above the ITO glass substrate to account for significant surface roughness of the surface. I am not really convinced by this statement and have the feeling that this refractive index is there mostly to adjust the observed resonances with the experimental one. I would gladly like

to have more details on where such resonance mismatch could come from. For example, it is well known in the literature that quantum surface effects (see e.g. Yang et al., Nature 576, 248 (2019)) strongly shifts the resonance of plasmonic systems. Could this somehow also explain the resonant wavelength mismatch between observed and numerical data?

Our response: We have read carefully the literature of *Yang et al. Nature 2019, 576, 248-252*. The authors have established an experimental procedure to measure these complex dispersive surface-response functions, using quasi-normal-mode perturbation theory and observations of pronounced nonclassical effects. They have also demonstrated the nonclassical spectral shifts over 30% and the breakdown of Kreibig-like broadening in a quintessential. Indeed, such quantum surface effects can strongly shift the plasmonic resonances.

Fig. R16. The normalized experimental scattering (red curve) of an AuND on the ITO-coated glass substrate. The dashed red curve is the simulated scattering of the AuND located on the ITO-coated glass substrate using an effective refractive index $n=1.26$ above and around the ND. The dashed black curve is the simulated scattering of this AuND using the refractive index of air ($n=1.00$) above the substrate.

As reported by Yang et al., the quantum surface effects can significantly shift the plasmonic resonances towards the high energy (blue shifts), compared to the classical predictions. From the measurements in our manuscript, however, the measured plasmonic resonances of AuNDs have a significant redshift (more than 50 nm) relative to the theoretical results obtained by the classical simulations when we do not consider the dielectric environment changes above the substrate and around the metal nanostructures (the dashed black curve in Fig. R16). Since the surface roughness of the ITO-coated glass substrate is comparable to (and even larger than) the size of Au nanoparticles (see the inset of Fig. R16), the nanoparticles have possibilities residing in some sunken locations on the substrate's surface, making the refractive index around the nanoparticles cannot be simply used as that of air ($n=1.00$). Therefore, an

effective refractive index should be considered above the substrate when we simulated it as a perfectly smooth surface.

Some quantum surface effects may exist in the AuNDs; however, we think that the refractive index changes caused by the surface roughness of the substrate play a key role in the redshift of the plasmonic resonances in our cases.

Corresponding Revision: We have added Supplementary Fig. S20 in the revised SI to demonstrate why we have used an effective refractive index (1.26) in FEM simulations of the AuND located on ITO-coated glass substrate (see the revised line 504 and 505 on page 26 and the added Fig. S20 in the revised SI).

c. I also have reserve on the method presented here, consisting in modeling the molecule with a cylinder whose dielectric function is a fitted Lorentz model. The authors mention the oscillator strength $f_0=0.27$. How does this relate to the single molecule dipole moment $\mu_c=0.09$ e.nm, which is roughly the single MB molecule transition dipole moment? Also, even if it may present some difficulties, it is usually preferable to model a molecule as a point dipole with a chosen position and orientation. Here, the cylindrical shape spreads in the gap, even if small and sized as a CB molecule. In terms of numerical simulation, any finite volume geometry whose refractive index differs from the medium's supports confined modes, in addition to the Lorentz resonance in the dielectric function. How do the authors make sure that the supported modes of this cylinder do not interfere in their calculations? To defend the authors, it is true that this method was used in Chikkarady's work, but I am especially careful because I have earlier seen misinterpretations related to hybridized modes interpreted with this method.

Our Response:

(I) In our manuscript, we adopted $\mu_c = 0.09$ e nm for a single MB-molecule exciton in calculations according to *Phys. Chem. Chem. Phys.*, 2000, 2, 4313-4317, which is also a typical value for single excitons [*Nano Lett.* 2022, 22, 2177–2186]. According to the Reviewer's suggestion, we have attempted to estimate the single MB molecule oscillator strength, f_0 , using [*Phys. Chem. Chem. Phys.*, 2000, 2, 4313-4317],

$$f_0 = \frac{(2\pi)^2 m_e \omega_c |\mu_c|^2}{3e^2 h}, \quad (\text{R3})$$

where, m_e and e are the mass of and charge of the electron; μ_c is the transition dipole moment; $\omega_c = 2\pi c / \lambda_c$ is the frequency at maximum absorbance. Substituting $\mu_c = 0.09$ e nm into Eq. (R3), one can get $f_0 \approx 0.41$ for individual MB emitters. This value seems to be larger than what

we utilized in our manuscript. It should be noted that we simulated the whole CB[7]@single MB as a uniform dispersive medium cylinder with the same volume as that of the CB[7]. However, the volume of an emitter (*i.e.*, MB monomer) is much smaller than that of the CB[7]. Therefore, we have adopted a smaller $f_0 = 0.27$ to describe the whole CB[7]@single MB, as Chikkarady et al. have done in *Nature* 2016, 535, 127–130. This value used in our coupling systems is reasonable because one can estimate that there was about 0.6 MB molecule/nm³ in the dispersive medium in Chikkarady's model, according to the result of their FDTD simulations (Supplementary Fig. 3 and Fig. 4(a) in *Nature* 2016, 535, 127–130). In our AuND/CB[7]@single MB hybrid system, there is about 0.63 MB molecule/nm³ (1/1.59 nm³, 1.59 nm³ is the volume of a dispersive medium cylinder, *i.e.*, the CB[7] molecule in the gap) in the dispersive medium. On the other hand, it is found that such numerical simulations can well reproduce our experimental results. Therefore, we argue that adopting such an oscillator strength is acceptable for simulating the strong coupling in our hybrid systems.

(II) As the Reviewer has suggested, one can model a single molecule as a point dipole with a chosen position and orientation in simulations. We think such an approach is suitable for simulating the scattering/absorption spectrum of the plasmon mode and the dipole (emitter) emission. However, for directly simulating the strong coupling between two interacting components, manifesting as spectral Rabi splitting (SRS) in scattering or absorption of the hybrid system (or the coupling subcomponents) driven by a plane wave (just as that we have employed in our experiment), this approach does not seem appropriate. On the contrary, the common method that has been extensively used is modeling the excitonic material as a dispersive medium described with the Lorentz model, as what has been done in our manuscript and other previous literature [*Nano Lett.* 2013, 13, 3281–3286; *Nature* 2016, 535, 127–130; *Nature Commun.* 2016, 7, 11823; *Nature Commun.* 2018, 9, 4012; *Phys. Rev. Lett.* 2015, 114, 157401; *Nano Lett.* 2017, 17, 3809–3814; *Phys. Rev. Lett.* 2017, 118, 237401].

(III) We have calculated the absorption cross-section spectrum of the dispersive medium cylinder, which was used to model the CB[7]@single MB emitter with the dielectric permittivity described by the Lorentz model (Eq. (4) in our manuscript), and this medium cylinder (0.75 nm in radius and 0.9 nm in thickness) was placed in a uniform dielectric environment with a refractive index of 1.26 (as we have used in our manuscript). The calculated result (Fig. R17) shows that no other modes are supported by this dispersive medium cylinder which can interfere with our calculations.

Fig. R17. The calculated absorption cross-section spectrum of the dispersive medium cylinder using Eq. (4) (i.e., $\varepsilon(\omega) = \varepsilon(\infty) + \frac{f_0 \omega_c^2}{(\omega_c^2 - \omega^2 - i\Gamma_c \omega)}$) in our manuscript. In calculations, $\varepsilon(\infty) = 1.96$, $f_0 = 0.27$, $\omega_c = 1.880$ eV, and $\Gamma_c = 85$ meV were utilized.

d. In the Methods section, I did not understand either why the radii of the cylinders used to model the coupled molecules were 1.1 and 1.4 nm for 2 and 3 molecules in the gap, respectively. This requires some clarifications.

Our Response: We acknowledge that we have not given a clear description in the original manuscript about why the radii of the cylinders used to model the coupled molecules were 1.1 and 1.4 nm for two and three CB[7]@single MB emitters in the gap. Considering the total volumes of two and three CB[7]@single MB emitters embedded in the gap, we modeled these two cases as two effective medium cylinders with the same volumes as the two and three CB[7] molecules, respectively. The dispersive medium cylinders used to model these two cases are 1.1 nm ($\approx 0.75 \times \sqrt{2}$ nm) and 1.3 nm ($\approx 0.75 \times \sqrt{3}$ nm) in radii and 0.9 nm in length, respectively. Notice that we have recalculated and renewed the SRS for the strong coupling case of three MB molecules (Fig. 4(d) in our manuscript) by modeling three CB[7]@single MB emitters in the gap as a dispersive medium cylinder with 1.3 nm in radius and 0.9 nm in length.

Corresponding Revision: We have explained why the radii of the cylinders used to model the coupled molecules were 1.1 and 1.3 nm for 2 and 3 molecules in the gap, respectively; we have recalculated and renewed the SRS for the strong coupling case of three MB molecules (see the revised lines 515-520 on page 26; and revised Fig. 4(d) in the revised manuscript).

e. The method to extract the coupling components of the absorption in Fig. 3b was also too succinctly detailed.

Our Response: We acknowledge that we have not clearly described the method to obtain the absorption cross-section spectra of the two coupling components (Fig. 3(b)). According to the Reviewer's suggestion, we have added the calculation details in Supplementary Section S2 of the revised SI. In the simulations, we used the finite element method (FEM) to calculate the absorption cross-section of the MB molecule by integrating the total density of power dissipation in the domain of the CB[7]@single MB, i.e., the dispersive medium cylinder between the two Au nanoparticles (the dielectric permittivity of the cylinder was described by the Lorentz model, as we have mentioned above), and then the integral result was normalized to the intensity of the incident field (I_0). Similarly, the absorption cross-section of the AuND in the hybrid system was calculated by integrating the total density of power dissipation in the domain of the two Au nanoparticles. Then the integral result was also normalized to the intensity of the incident field (I_0). It should be mentioned that such a calculation method is the standard approach Comsol Multiphysics provides. It enables us to conveniently calculate the absorption cross-section of any domain in the hybrid system.

Corresponding revision: We have added the detailed method to extract the coupling components of the absorption in Fig. 3(b) (see the revised lines 504-512 on page 26 in the revised manuscript; the added part of *simulations of the absorption cross-section spectra for the coupling components* in the revised Section S2 in the revised SI.

Comment 3: *Related to Fig. 3f (but also Fig. 3b), the authors mention a “spectrum broadening instead of a clear SRS of the exciton absorption [...] further demonstrating that the intermediate-strong coupling regime has been reached for the single exciton interacting with the gap plasmons”. This statement, as well as the data shown in these figures, weakens the strong statement made in the first sentence of the Discussion section: “we have realized the strong coupling of a single exciton at room temperature in plasmonic nanodimers [...]”. Could the authors elaborate a bit more on the intermediate strong coupling regime? Also, what does this imply with respect to the SRS observed in the total absorption?*

Our Response: This is indeed an important question. Next, we will briefly introduce the intermediate-strong coupling regime that we have defined and explain the relationship between the intermediate-strong coupling and the SRS observed in the total absorption and the absorption of the coupling components.

In our recent work [*Phys. Rev. B* 2021, 103, 235430], we have demonstrated the relativity and diversity of strong coupling in the coupled plasmon-emitter systems. As is well known, the eigenlevels (E_{\pm}^l) of the strong coupling system can be obtained as:

$$E_{\pm}^l = (\varepsilon_d + \varepsilon_c) / 2 - i(\Gamma_d + \Gamma_c) / 4 \pm \hbar\Omega_{LS} / 2, \quad (\text{R4})$$

with the level splitting (LS) of $\hbar\Omega_{LS} = 2\sqrt{g_{dc}^2 - [(\Gamma_d - \Gamma_c)/4 + i\delta/2]^2}$, and $\delta = \varepsilon_d - \varepsilon_c$ is the detuning between the two coupling components. **Theoretically**, as shown in the following Fig. R18(a), when $g_{dc} < (\Gamma_d - \Gamma_c)/4$ the system is in the weak coupling regime, and no LS is occurring; when $g_{dc} > (\Gamma_d - \Gamma_c)/4$ the eigenlevel splits into two branches, the precondition for light-matter strong coupling. **Experimentally**, the level splitting of a strongly coupled system is usually probed by the absorption or emission spectrum Rabi splitting (SRS). **However, there has been a controversy about whether or not the SRS exactly reflects the LS of the strongly coupled plasmon-emitter systems.**

[FIGURE REDACTED]

Fig. R18. (a) Real (solid curve) and imaginary (dashed curve) parts of the mixed eigenlevels shown in Eq. (R4) as a function of g_{dc} at resonance. (b) The detailed interaction regimes classified for the plasmon-emitter interaction $\Gamma_d > \Gamma_c$ are derived according to the analytical classification criteria in Eqs. (16), (27), and (28) in *Phys. Rev. B* 2021, 103, 235430. SC: strong coupling. The two Figures are taken from Fig. 1(b, d) of *Phys. Rev. B* 2021, 103, 235430.

Our investigations have confirmed that the critical criteria and the width of LS differ from those of the SRS observed from the coupling components. These critical criteria strongly depend on coupling strength and the damping line widths of the coupling subsystems, manifesting the plasmon-emitter interactions as interesting relativity and diversity. According to these critical criteria, the plasmon-emitter interactions can be classified into five different coupling regimes (Fig. R18(b)): **(I)** the weak coupling regime, in which no LS occurs; **(II)** the pseudo-strong coupling regime, where no LS occurs but the SRS can be observed in absorption from the plasmon channel; **(III)** the dark strong coupling regime, in which the LS exists, but there is not the SRS cannot be observed in the absorption spectrum from any channel; **(IV)** the intermediate-strong coupling regime, where the LS occurs, and the SRS can be seen in the absorption signal from the plasmon channel, but it cannot be seen in the absorption signal from the exciton channel; and **(V)** the super-strong coupling regime, in which the LS and the SRSs observed in the absorption spectra from the two channels coexist.

Related to Fig. 3(b, f) in the manuscript, the SRS can be seen in the absorption signal from the plasmon channel but not in the absorption signal from the exciton channel. Thus, we

classified this interaction as an intermediate strong coupling. However, combined with the reply to **Comment 5** raised by Reviewer #1, we have revised the presentation of this part of the simulations in the revised manuscript, and we have reclassified this interaction as a ‘strong coupling regime’ based on the commonly used strong-coupling criterion of $\hbar\Omega_R=118 \text{ meV} > (\Gamma_d + \Gamma_c)/2=107.5 \text{ meV}$.

Regarding “*what does this imply with respect to the SRS observed in the total absorption?*” Our studies show that the absorption spectrum of the coupled plasmon-emitter system can be written as [Phys. Rev. B 2021, 103, 235430],

$$\sigma(\omega) \propto A_d \sigma_d(\omega) + A_c \sigma_c(\omega), \quad (\text{R5})$$

with

$$\sigma_d(\omega) = \langle\langle d; d^+ \rangle\rangle \propto -\text{Im} \frac{\omega - \varepsilon_c + i\gamma_c}{(\omega - \omega'_+)(\omega - \omega'_-)}, \quad (\text{R6})$$

$$\sigma_c(\omega) = \langle\langle c; c^+ \rangle\rangle \propto -\text{Im} \frac{\omega - \varepsilon_d + i\gamma_d}{(\omega - \omega'_+)(\omega - \omega'_-)}, \quad (\text{R7})$$

where $\sigma_d(\omega)$ and $\sigma_c(\omega)$ are the absorption spectra of the plasmon- and QE-subsystem, respectively, $A_d \propto \Gamma_d$ and $A_c \propto \Gamma_c$ are the weight factors of the photon absorption from the two channels. Eq. (R5) indicates that the total absorption spectrum of the coupled system is the superposition of the absorption spectra from the two channels. Due to $A_d \propto \Gamma_d$ and $A_c \propto \Gamma_c$, it was found that the main contribution to the total absorption of the coupled system comes from the component with larger decay.

Corresponding Revision: We have rewritten the part of FEM simulations in Fig. 3 and replaced the “*intermediate-strong coupling regime*” with the *strong coupling regime* according to the commonly used strong-coupling criterion $\hbar\Omega_R > (\Gamma_d + \Gamma_c)/2$ (see the revised lines 273-280 and 290 on page 14).

Comment 4: *The mode volume defined in Eq. (S1) is valid for Hermitian systems, i.e., closed cavities. For lossy, dispersive and absorbing nanoresonators such as the investigated gold dimers, the formula for the mode volume is different, see e.g. Wu et al., ACS Photonics 8, 307 (2021). Because such resonators are non-Hermitian, the mode volume is complex and this can result in the Purcell factor for a single molecule not being necessarily maximal on resonance, for instance. An online freeware, which can be found in the following link (<https://doi.org/10.5281/zenodo.7400937>, see also Wu et al., Comp. Phys. Comm. 284, 108627 (2023) for a tutorial), provides matlab codes that can be used together with COMSOL to*

determine such complex mode volumes. I would highly recommend the authors to determine the mode volumes with the formula from Wu et al., at least to check whether their computed mode volume (formula (S1), which is wrong), is similar with the real part of the complex mode volume. If so, the imaginary part also provides interesting physical content, e.g. a non-Hermitian phase. I understand that this may require some work, and in the defense of the authors, another debatable (yet complex-valued) formula was used in the work of Chikkarady (Eq. (S3) of their SI). I expect at least a significant discussion on the complex mode volume of such structures, and a strong justification of why the authors used Eq. (S1).

Our Response: We are very grateful to the Reviewer for recommending us to learn the new and more accurate method for calculating the mode volume of plasmonic nanocavities. **In fact, due to the AuND having a sub-nanometer gap, most of the field $|E(\mathbf{r}, \omega)|$ and energy density are confined only in the gap between the nanoparticles,** the AuND with a sub-nanometer gap can be deemed as an optical cavity, and its mode volume V_m can be calculated by using the formula employed by Chikkarady et al. (i.e., Eq. (S3) of their SI in *Nature* 2016, 535, 127–130),

$$V_m(\omega) = \frac{\int w(\mathbf{r}, \omega) d^3\mathbf{r}}{\max[w(\mathbf{r}, \omega)]}. \quad (\text{R8})$$

To make the equation applicable to plasmons, the $w(\mathbf{r}, \omega)$ was modified as,

$$w(\mathbf{r}, \omega) = \frac{1}{2} \left[\text{Re}[\varepsilon(\mathbf{r}, \omega)] + \frac{2\omega \text{Im}[\varepsilon(\mathbf{r}, \omega)]}{\beta} \right] |E(\mathbf{r}, \omega)|^2 \quad (\text{R9})$$

where the term $\varepsilon(\vec{r}, \omega)$ is the permittivity of the metal at position \vec{r} . β is the plasmonic damping term, and $\text{Re}[\]$ and $\text{Im}[\]$ are, respectively, the real and imaginary parts. Notice that these two formulae (Eqs. (R8) and (R9)) are Supplementary Eqs. (S1) and (S2) in our SI,

$$V_m(\omega) = \frac{\int \varepsilon(\mathbf{r}, \omega) |E(\mathbf{r}, \omega)|^2 d\mathbf{r}}{\max[\varepsilon(\mathbf{r}, \omega) |E(\mathbf{r}, \omega)|^2]}, \quad (\text{S10})$$

with the modification of $\varepsilon(\mathbf{r}, \omega) \rightarrow \text{Re}[\varepsilon(\mathbf{r}, \omega)] + 2\omega \text{Im}[\varepsilon(\mathbf{r}, \omega)]/\beta$. **It should be mentioned that this method is used to calculate the real part of the complex mode volume, and it has been extensively utilized in the previous literature to calculate the V_m of the plasmonic nanostructures** [*Nature* 2016, 535, 127-130; *Phys. Lett. A* 2002, 299, 309-312; *Opt. Lett.* 2010, 35, 4208-4210; *Phys. Rev. Lett.* 2015, 114, 157401; *Nano Lett.* 2022, 22, 4686–4693; *Phys. Rev. Lett.* 2023, 130, 143601]. Despite that such a calculation approach for the plasmonic V_m is somewhat debatable, we argue that it is reasonable for calculating the V_m of the AuND with a sub-nanometer scale gap **because most of the field $|E(\mathbf{r})|$ and energy density are confined**

only in the gap between the nanoparticles (Fig.1(e)), the permittivity $\epsilon(\mathbf{r})$ is still dominated by the refractive index in the gap, and the imaginary part of V_m can be neglected [Nature 2016, 535, 127-130].

To further verify the rationality of the above V_m calculated using Eq. (S1) in our Supplementary Materials, we have attempted to calculate the V_m using the formula [Phys. Rev. Lett. 2013, 110, 237401; Nature 2016, 535, 127-130],

$$F_p = \frac{3}{4\pi^2} \frac{Q}{V_m} \left(\frac{\lambda}{n} \right)^3, \quad (\text{R10})$$

F_p is the Purcell factor of a point dipole in the gap center, which can be obtained by,

$$F_p(\omega) = \frac{\Gamma}{\Gamma_0} = \frac{\rho_n(\mathbf{r}_E, \omega)}{\rho_n^0(\mathbf{r}_E, \omega)}, \quad (\text{R11})$$

where $\rho_n^0(\mathbf{r}_E, \omega)$ is the local density of optical states (LDOS) in free space. It can also be written in terms of the modified spontaneous emission rate Γ of the QE, divided by its free space spontaneous emission rate Γ_0 . Our calculation shows that the Purcell factor in the gap center is up to $\sim 3.2 \times 10^6$ at the resonant wavelength of 659.8 nm (resonant to the exciton transition in MB molecule). Based on this Purcell factor, we can calculate the Q factor, here $Q = 12.96$ at 659.8 nm, so we can then extract the mode volume at 659.8 nm is of $V_m \sim 29 \text{ nm}^3$, agreeing with that ($V_m \sim 26 \text{ nm}^3$) we have calculated by using the Supplementary Eqs. (S1) and (S2) in our manuscript. It should be mentioned that **the method of Eqs. (S1) and (S2) are more suitable for calculating the mode volume of a nanocavity excited by the plane wave, as we have utilized in our experiments.** Thus, the calculations and obtained V_m for the AuND in our manuscript are reasonable. We have also added this second method to estimate the V_m of the AuND cavity in our revised SI.

Corresponding revision: We have discussed the reasonability of the mode volume calculated by Eq. (S1) in our Supplementary Information (SI), and we also have added another estimation method by using the Purcell factor in the revised SI (see the revised Section S4 in the revised SI).

Comment 5: *As a suggestion for improving the manuscript, I would recommend that the authors compute the Purcell factor of a point dipole placed in the center of the gap and oriented along the dimer axis, versus wavelength or frequency (see e.g., (S6) in Chikkarady's work). If the Purcell factor is strongly peaked ($\sim 10^6$) around the resonance of the dimer, that would further bring evidence of the strong coupling regime at the single molecule level, as shown in Rousseaux et al., PRB 98, 045435 (2018). In this work, a Lorentzian fit of the Purcell factor is*

used to extract the coupling strength, knowing the dipole moment of the emitter. This was done with the authors' theoretical model that they self-cite, with a closed-form expression for the scattering, but not for the Purcell factor formula.

Our Response: As we have demonstrated in the above reply for **Comment 4**, we have calculated the Purcell Factor of a point dipole placed in the center of the gap and oriented along the dimer axis versus the wavelength (see the above Fig. R14 and Supplementary Fig. S9 in our revised SI). The calculation result shows that the Purcell Factor of the point dipole placed in the gap center of an AuND (constructed by a single CB[7], gap = 0.9 nm) can be up to about 3.2×10^6 at 659.8 nm, resonating to the exciton transition at about 660 nm in MB molecule, indicating that this AuND construction is fully capable of achieving the single molecule strong coupling [*Phys. Rev. B* 2018, 98, 045435; *Nature* 2016, 535, 127-130].

Corresponding revision: We have calculated the Purcell factor of a point dipole placed in the gap center of the AuND and oriented along the dimer axis, versus wavelength. We have also discussed the revised Section S3 of the revised SI (see the revised lines 158-161 on page 8; the revised Section S3 and the added Fig. S9 in the revised SI; and the added ref. 40 in the revised manuscript).

Comment 6: *I already mentioned quantum surface corrections in comment 2b. Other complexity may arise with e.g., dark plasmonic modes in these structures. For example, many strongly confined modes can appear in NPoM structures (see e.g. Kongsuwan et al., ACS Photonics 7, 463 (2020)). Additionally, it was recently suggested that dark modes, that are usually thought as a dissipative channel that quench the emitter's fluorescence, could result in a pseudomode hybridizing with the emitter's transition and promoting strong coupling dynamics seen in the spectral Rabi splitting of the bright dipolar mode of a nanosphere (Rousseaux et al., Phys. Rev. Research 2, 033056 (2020)). Since in the present case, the molecule is so tightly confined in such a small metallic gap, what about the coupling to such strongly confined dark modes?*

Our Response: We have read the references recommended by Reviewer #3. Indeed, there will be strongly confined dark modes in the famous NPoM structures, and the confined dark modes may strongly couple to the exciton's transitions, which significantly influences the interaction between the bright dipolar mode and the exciton. According to the Reviewer's suggestion, we have also investigated the modes highly confined in the gap (0.9 nm) of the AuND.

Fig. R19. Plasmonic modes confined in the nanogap of an AuND constructed by a single CB[7] molecule (gap = 0.9 nm), the plasmonic modes are excited by a classical emitter placed in the center of the gap at the position of the maximum field. Its spontaneous emission enhancement factor (Purcell factor) is plotted as a function of wavelength.

As we have discussed above, the calculated Purcell factor of a point dipole (placed in the center of the gap and oriented along the dimer axis) versus wavelength show that there are indeed some strong dark modes with much higher intensities compared to that of the bright dipolar mode (Fig. R19), despite these dark modes cannot be observed in the light scattering of the AuND. Unlike that of the single Au (or Ag) nanoparticles (arXiv:2211.03247; *Phys. Rev. Research* 2020, 2, 033056), **the energy of the dark plasmonic modes in the AuND is far away from that of the bright dipolar mode and the transition frequency of the MB molecule (~660 nm), indicating that the interactions between the excitonic transition and the dark plasmonic modes will be in the weak coupling regime.** Therefore, we argue that the dark plasmonic modes could not result in a pseudo mode hybridizing with the exciton's transition and promoting strong coupling dynamics seen in the spectral Rabi splitting of the bright dipolar mode of an AuND, which is unlike the nanosphere case theoretically demonstrated in *Phys. Rev. Research* 2020, 2, 033056. It is a merit for the AuNDs with sub-nanometer gaps to exploit the strong coupling of single emitters.

While, the weak interactions between the dark mode and the emitter can induce a finite frequency shift (Lamb shift) of the emitter transition, which can be reflected by a shift of the splitting valley position in the scattering/absorption spectrum of the AuND-single molecule system (arXiv:2211.03247). So, one can investigate the Lamb shift based on this single-molecule strong coupling system, even though the frequency shift of the emitter in such a sub-nanometer plasmonic gap has a very complex mechanism. We have also discussed this point in Section S3 of the revised SI.

Corresponding revision: We have discussed in the revised Section S3 of the revised SI (see the revised Section S3, the added ref. S[6] and Fig. S9 in the revised SI, and the added ref. 40 in the revised manuscript).

Comment 7: *Similarly, the authors may be familiar with recent works on picocavities in NPoM geometries, whose main investigator, J. Baumberg, recently published a short review article (Baumberg, Nano Letters 22, 5859 (2022); see also Benz et al. Science 354, 726 (2016)). Could such dimers be used as a platform to investigate the coupling of picocavities, i.e. atomic protrusions coupling to molecules close to the metallic surface?*

Our Response: We appreciate this constructive suggestion. Accordingly, we have carried out extra SERS measurements of single proflavine (PF) molecules embedded in the nanogap of individual AuNDs (i.e., AuND/CB[7]@single PF hybrid). The following Fig. R20 (i.e., Fig. 5(b) in the revised manuscript) shows the time-series SERS spectra from a coupled AuND/CB[7]@single PF hybrid, from which we see that aside from the ‘spectral diffusion’ of the SERS peaks, new strong vibrational lines can be observed over a certain period. Such an interesting phenomenon agrees well with the single molecule behavior in picocavity reported by J. J. Baumberg and his co-workers (*Science* 2016, 354, 726–729; *Nano Lett.* 2022, 22, 5859–5865), indicating that there may be a picocavity has been formed in our AuND. Therefore, as the Reviewer has pointed out, the AuND will provide another promising platform to investigate the coupling of picocavities. Of course, many investigations still need to be further performed to exploit this interesting phenomenon in such individual hybrid AuNDs. Also, we are very grateful to Reviewer#3 for raising this important question which gives us an interesting research direction.

Fig. R20. Time-dependent evolution of SERS signals from the typical AuND/CB[7]@single PF hybrids. Each spectrum is collected for 2s at 633 nm, 60 $\mu\text{W}/\mu\text{m}^2$ pump.

Corresponding Revision: We have added extra SERS measurements of AuND/CB[7]@single PF hybrids, demonstrating that a picocavity has been formed in the hybrid AuND (see the added Fig. 5(b); the revised lines 377-383 on page 19; the revised lines 384-386 on page 20 in the revised manuscript).

Comment 8: *Finally, I would suggest strengthening the Discussion section a little. As far as I understood by reading the paper, the only advantage in the investigation of single-molecule strong coupling with Au dimers instead of NPoM would be that they present (hardly) less dissipation: $Q \sim 14$ while $Q \sim 11-12$ for NPoM, as mentioned by the authors in the text. To me, this seems quite irrelevant. The quality factor is still low, and anyway, the main interest of nanocavities does not lie in the ability to obtain higher Q -factors, owing to large dissipation due to the metal. However, maybe such structure can present other significant interest? I would like to see what the opinion of the authors on this matter is.*

Our Response: According to the Reviewer's good suggestion, we have removed the statement, "Aside from processing an ultrasmall mode volume, the gap plasmon mode supported by the Au nanodimer has a lower dissipation compared to that of the NPoM configuration, which has been used for the first time to investigate the single-molecule exciton strong coupling" in the Discussion section of the revised manuscript. In addition, we have also strengthened the discussion's attempt to reveal other significant interests of the hybrid AuNDs. For example, we added and discussed extra experiments (Fig. 5(a) in the revised manuscript) to demonstrate the 'spectral diffusion' of the vibrational lines observed from a single MB molecule in individual AuNDs. This section further discussed the single molecule events in the strongly coupled AuND/CB[7]@single MB hybrids at a low dye concentration. On the other hand, as we have replied in the above **Comment 7**, we have also discussed that such strong coupling AuND-single molecule nanosystem maybe provides another useful platform for exploiting the coupling of picocavities (Fig. 5(b)), i.e., the generation and evolution of Au clusters in the sub-nanometer gap of the dimer under laser irradiation and single molecule action. Therefore, such AuND/CB[7]@single dye molecule strong coupling system will exploit pico-chemistry, pico-tweezers, and in-situ single-site photocatalysis.

As the Reviewer has pointed out, the strong coupling of single-molecule excitons to plasmonic NDs represents a significant step in investigating single-emitter strongly coupled nanodevices, with interesting prospects for room-temperature quantum technologies. Therefore, the strong coupling of the single exciton to AuND may provide a more rigorous testbed to explore single electronic transitions optically at room temperature, representing a significant step towards single-emitter strongly coupled nanodevices, with interesting prospects for room-

temperature quantum technologies. Also, this strongly coupled system provides an ideal testbed for exploiting the single-molecule Raman detection, quantum chemistry, and quantum optical devices where a single qubit is needed. We have discussed these points in the recent manuscript.

Corresponding Revision: We have removed the sentence “*Aside from processing an ultrasmall mode volume, the gap plasmon mode supported by the Au nanodimer has a lower dissipation compared to that of the NPoM configuration, which has been used for the first time to investigate the single-molecule exciton strong coupling*” in the Discussion section of the original manuscript. We have also strengthened the Discussion section in the revised manuscript (see the revised lines 369-383 on page 19; the revised lines 384-386 on page 20; the added Fig. 5 in the revised manuscript).

Comment 9: *The statistics presented in Fig. 4d is too succinctly detailed. Please specify the number of sampled structures (which, I guess, corresponds to the one presented in Fig. 4c, but this is not explicitly written), and what the error bars represent (I guess, standard deviation?).*

Our Response: According to the Reviewer’s suggestion, we have clarified the specific number of sampled structures in the revised manuscript in Fig. 4(d). The number of sampled structures counted was 40, 15, and 10 in our original manuscript, respectively. As the Reviewer pointed out, the number of sampled structures for two- and three-molecule exciton strong coupling cases corresponds to the one presented in Fig. 4(c). On the other hand, the error bars represented in Fig. 4(d) are indeed standard deviations; we have added this description in the figure legend.

Corresponding revision: We have clarified the specific number of sampled structures in Fig. 4(d) and the meaning of the error bars in this figure (see the revised line 364-365 on page 17).

Comment 10: *In the introduction, the authors mentioned a former work with “a cuboid Au@Ag nanorod coated with dye molecule J-aggregate excitons”. At this stage in the text, for better readability, I would suggest relaxing the use of notations such as Au@Ag and describe the device in a few words instead.*

Our Response and Corresponding Revision: According to the Reviewer’s suggestion, we have relaxed the notation of Au@Ag as Au (core)-Ag (shell) nanorod and described the device in a few words. In addition, we have also relaxed the notation of 2D as two-dimensional in the current introduction. (see the revised lines 52, 53, and 59 on page 3 in the revised manuscript).

1) *On page 4, “CB[7]” was used without being defined before as cucurbit[7]uril molecule.*

Our Response and Corresponding Revision: We have defined cucurbit [7]uril molecule as CB[7] before it is used in our revised manuscript (See revised lines 94 and 95 on page 5 in the revised manuscript).

2) On page 8, I believe the statement “We set $N=1$ and $g_{dc}=56.6$ meV in calculations, and the rest of the parameters were extracted from the experimental measurements” is incorrect, since g_{dc} was extracted with a fit of SRS with Eq. (1). So, the only assumption is $N=1$.

Our Response and Corresponding Revision: As the Reviewer pointed out, the only assumption is $N = 1$ in our calculations. We have revised the sentence “We set $N=1$ and $g_{dc}=56.6$ meV in calculations, and the rest of the parameters were extracted from the experimental measurements” as “We set $N=1$ in calculations, and the rest of the parameters were extracted from the experimental measurements” (see revised lines 219 and 220 on page 11 in the revised manuscript).

3) In Eq. (1) (and several other times), h should be replaced by \hbar .

Our Response and Corresponding Revision: In Eq. (1) (and several other times), h should be \hbar . Our original Word version of the manuscript contains all \hbar of these equations. Maybe, it has something wrong when we uploaded the Word version of the manuscript and converted it to the PDF version via the online submission system. In our revised manuscript, we have corrected h as \hbar in Eq. (1) and other related equations and made sure it is displayed correctly in the corresponding PDF version (see the revised Eqs. (1)-(3) in the revised manuscript).

4) On page 12, I found the sentence “At the dye resonance, [...] strong coupling” a little too short, just referring to two other works. I think this point is interesting and deserve a little more explained physics. Why is the conductance of the molecule enhanced due to the coupling?

Our Response: According to the Reviewer’s suggestion, we have revised this sentence in the manuscript. We have added explained a little more physics about this interesting phenomenon. Research has shown that the exciton conductivity (mobility) in organic materials can be significantly enhanced when the molecules are strongly coupled to the mode field [*Nat. Mater.* 2015, 14, 1123-1129; *Phys. Rev. Lett.* 2015, 114, 196402]. Such an interesting phenomenon originates from the extended coherence of the polaritonic states induced by the hybridization of the exciton state with the mode field at resonance ($\omega_d = \omega_c$) [*Nat. Mater.* 2015, 14, 1123-1129]. Then, the enhanced exciton conductivity will largely improve the delocalization of the charge carriers at the dye resonance, leading to a remarkable decrease in the EF intensity of the mode field around the junction between the two Au nanoparticles. We have also added *Nat. Mater.* 2015, 14, 1123-1129 as ref. 50 in the revised manuscript.

Corresponding Revision: We have revised the sentence “At the dye resonance, [...] strong coupling” and added a little more physics about this interesting phenomenon (see the revised

lines 305-311 on page 15; the revised lines 312 and 313 on page 16; and the added ref. 50 in the revised manuscript).

5) In Fig. 4 a and b, the dashed lines, which I assume are Lorentzian fits, are not described.

Our Response and Corresponding Revision: The dashed lines in Fig. 4(a, b) are indeed Lorentzian fits. In the revised manuscript, we have added the descriptions of these dashed lines in the revised figure legend and caption of Fig. 4(a, b).

6) In Fig. 4 d, the green stars do not appear in the legend.

Our Response and Corresponding Revision: We have described the green stars in the figure caption of Fig. 4(d) but have not given it in the legend. According to the Reviewer's suggestion, we have added the green star and its description in the figure legend in the revised Fig. 4(d) (see the revised Fig. 4(d) in the revised manuscript).

7) In the Methods sections, the word "field" is written "filed" several times.

Our Response and Corresponding Revision: We have corrected the "field" as "filed" in the revised manuscript, and we have also rechecked all of our manuscripts carefully and revised all typo errors (see the revised lines 450-452, and 458 on page 23 in the revised manuscript).

8) In addition to ref. 45, I would suggest adding a more recent reference, that is even more relevant since it corresponds to TDDFT investigations of particle dimers coupled to single molecules: Kuisma et al., ACS Photonics 9, 1065 (2022).

Our Response and Corresponding Revision: We have cited the work of ACS Photonics 9, 1065 (2022) by Kuisma et al. as ref. 56 in the revised manuscript (see revised lines 663 and 664 on page 34 in the revised manuscript).

9) Fig. S12 from the supplementary info seems useless to me, since it is already shown as insets in Fig. 4 and b.

Our Response and Corresponding Revision: We have removed Fig. S12 (in the original SI) from the revised SI.

10) Related to Table S2, emitter numbers are shown for WS₂ monolayer and J-aggregate. The very idea of defining emitter numbers for excitons in these structures is debated, and I would recommend citing the work of Tserkezi et al., Rep. Prog. Phys. 83, 082401, 2020.

Our Response: According to the Reviewer's suggestion, we have read the review paper by Tserkezi et al. and cited this work as ref. 24 in the revised manuscript. Indeed, some debates exist on evaluating exciton numbers involved in strong coupling using the excitonic materials of J-aggregates or transition-metal dichalcogenide (TMD) monolayers. We have also given a little discussion of this debate about the exciton number estimation in the revised manuscript and the caption of Table S1 (we revised the order of the original Table S2 as Table S1). **From this debate, one can also see the novelty of this manuscript in realizing the deterministic**

single-molecule exciton strong coupling with plasmons, which is important in developing room-temperature quantum devices based on the plasmon-single exciton strong coupling.

Corresponding Revision: We have cited the work by Tserkezis et al. as ref. 24 in the revised manuscript. We have also given a little discussion of this debate about the exciton number estimation in the revised manuscript and the caption of Supplementary Table S1 (see the added ref. 24 in the revised manuscript and the revised caption of Supplementary Table S1).

REVIEWER COMMENTS

Reviewer #1 (Remarks to the Author):

The authors have greatly improved their manuscript with numerous complementary experiments that strongly support their claims. I want to congratulate them for the nice assembly, SERS and photobleaching experiments that they conducted.

I only have very minor points but, overall, the paper should now be accepted for publication in Nature Communications.

1-The overall manuscript should be proofread by a native English speaker as there is a large number of typos and some odd sentences.

2-The authors only provide quantitative statistical information in the methods section and in the supporting information. But this information should appear in the main text: after optimizing the assembly procedure, about 20-25% of the sample actually corresponds to dimers, out of which 15% are in strong coupling. This is clearly an improvement over previous studies but these numbers must be stated in the main text. In the end, about 3-4% of the scatterers in the darkfield images correspond to strongly-coupled dimers: it is an important fact that the readers should be clearly made aware of.

3-Table S1 gives a nice overview of prior strong coupling experiments but I find it surprising that the data of ref 18 are not included as they also correspond to gold nanodimers as used in this manuscript.

4-the sentence "arising from a dramatic increase in the luminous MB monomers" on page 6 is unclear. Please rephrase. Maybe: "this indicates an improvement of the quantum yield of the MB dyes when hosted by CB molecules"?

5-the sentence "single-nanoparticle absorption measurement has been remained a challenge in experiment at present, owing to the incident light is always scattered by nanoparticles" is unclear (and grammatically incorrect). There are a number of experimental methods to assess extinction (Chem. Soc. Rev., 2014,43, 3921-3956) or absorption (ACS nano 11 (2), 1412-1418) by single nanostructures but they are indeed more complicated than darkfield scattering. Please rephrase.

Reviewer #2 (Remarks to the Author):

The authors have taken care of the concerns raised in my previous review. They have provided new experimental data on the two-molecule PCA and temporal dynamics of the system supporting single-molecule evidence. The arguments on the single-molecule scale are now convincing.

Having successfully addressed the issues and thus improved the quality of the work, I recommend accepting the manuscript for publication.

In the revised version of their manuscript, the authors provided additional experimental and numerical data, as well as corrections in the main text and supplementary information. Overall, all the results presented in this manuscript are, in my opinion, very interesting and encouraging, and point in the right direction for strong coupling between confined light and single emitters under ambient conditions.

However, with regard to the publication of this manuscript in Nature Communications, I am mainly concerned by the following assertions, which I consider to be false:

- The molecule's position is deterministic.
- The transition dipole moment can perfectly align along the plasmonic field.
- Single exciton strong coupling with the plasmonic dimer mode is achieved.

Therefore, I have to reject the manuscript. Nevertheless, I recognize the effort and contribution made by the authors to the field of nanophotonics, and I do believe that further work might justify a resubmission.

Related to the positioning of the molecular exciton with the electric field of the dimer mode, the authors claim in Table S1 that their work provide both "*emitter position certainty*" and "*dipole orientation certainty*". Why would the position be known for certain, as compared to, e.g., single MB molecules in Au NPoM (ref. 7)? In comment 3 from the second reviewer, the relevant point was raised that dimers can have facets. The main concern here was related to the number of molecules present in the gap, but this also means that even if there is only one molecule, its position in space is not fully known.

Related to the alignment of the molecular exciton with the electric field of the dimer mode, a critical point was raised in comment 2 of the second reviewer: "*it is very likely for some parts of the MB molecule to extend beyond the hydrophobic portal of CB[7], as MB is lengthier*". This comment was associated with the question whether this could prevent the CB[7] to bind the two nanoparticles together. If this is indeed true that the CB[7]@single MB binds the two nanospheres together, then, as the authors said, "*the MB molecule chain could be bent or compressed to some extent in the gap of the AuND*" so that the C=O functional groups of the CB[7] can access the gold surface. **If the molecule is folded or compressed, how can we be sure that the transition dipole moment is always oriented along the axis of the dimer? In fact, how can we be sure that the transition dipole moment has the same value as the "free" molecule?** My feeling is that we can only relax this assumption and try to examine the coupling experiment at different illumination angles. Reading the manuscript, I get the impression that the authors are relying a little too much on previous work. In NPoM structures, the plasmonic field is oriented perpendicular to the substrate, while it is parallel in the case of AuNDs, for which it would be interesting to see polarization-resolved dark field scattering as e.g., in Stührenbger et al., Nano Lett. 2018, 18, 9, 5938–5945.

Another point, which is the most critical, relates to the single exciton strong coupling claim. **The energy difference of 116 meV observed between the two peaks in the scattering spectrum cannot be interpreted as strong coupling between a single MB molecule and the plasmonic mode of the nanodimer.**

This can be confirmed both by calculations and the measured scattering before and after photobleaching the MB molecules in the gap of the AuNDs, Figure S13 in the revised

supplementary information. Bleaching the molecule should eliminate the Rabi splitting in the case of strong coupling, but this is not what is observed here. In Figure S13a, we clearly see that the two peaks are still there and that a transparency dip has been suppressed. But such a transparency dip cannot be associated to strong coupling. In panels b and c, we see a peak being suppressed, **but no closing of the gap between the peaks is observed**. The suppressed peak is maybe due to Purcell-enhanced emission of the molecule, but this is definitively not strong coupling. To highlight my point, I have added vertical lines in Figure S13, see attached image. In addition, in panel b, an extra peak is seen, and remains after bleaching. This peak is likely to be part of the bare AuND. In fact, similar additional shoulders are seen in bare AuNDs in Figure S15a. Note that the green line has a shoulder that is not seen in simulations. Such features may be due to imperfections in the morphology?

To show more clearly that the energy difference is not strong coupling, let us do a simple calculation. The coupling strength g is given by equation (10) in the added reference 40. This equation may look intimidating, but in fact it can be linked with the Purcell factor at resonance F_P , yielding the much simpler form $g = \frac{1}{2}\sqrt{\gamma_{MB}\Gamma_d F_P}$. This formula can also be found in Lalanne et al., Laser Photonics Rev. 2018, 12, 1700113 and Gerard, Top. Appl. Phys. 2003, 90, 269. Here $\gamma_{MB} = n\mu_{MB}^2\omega_{MB}^3/(3\hbar\pi\epsilon_0c^3)$ is the spontaneous emission rate of the MB molecule in a dielectric medium (here $n = 1.27$ to account for surface roughness of the substrate) and Γ_d is the plasmon decay rate of 135 meV provided by the authors. Since we now have the Purcell factor at resonance $F_P = 3.2 \cdot 10^6$, we find a coupling strength $g \approx 43$ meV, that is below the extracted value of 56 meV by the authors. This new value is below the condition $2g > (\Gamma_d + \Gamma_c)/2$, but it is approaching the strong coupling regime, which is encouraging.

Now, looking at the simulations provided by the authors, with the MB molecule being modeled as a cylinder with a Lorentz permittivity, a radius of 0.75 nm and a height of 0.9 nm, **I do see an overestimation of the molecule transition dipole moment**. The authors mention that they took the value of $f = 0.27$ to account for the small volume of the molecule, even though the formula (R3) yields $f = 0.41$. But in fact, this formula does not depend on the volume of the emitter.

In a recent arxiv by the group of Kristensen, a formula rigorously establishing the link between the Lorentz permittivity of the effective medium of the emitter and its volume has been shown (see equation (14) in arxiv:2308.12957). If we use the volume of 1.59 nm³ of the CB[7] and $f = 0.27$, we find a dipole moment of about 0.15 e*nm, which largely overestimates the dipole moment of the molecule (0.09 e*nm).

In fact, why not use the volume of the MB molecule itself? Such molecules are about 1.4 nm long and 0.95 nm wide (Jia, Puqi, et al. Applied sciences 8.10 (2018): 1903). If we assimilate the MB molecule with a cylinder of height 1.4 nm and radius 0.95/2 nm, and we keep the dipole moment value of 0.09 e*nm, we can use the same formula to compute the oscillator strength, which is now $f \approx 0.16$. If the authors insist in using the volume of the CB[7], then we obtain $f \approx 0.10$... This is why this method, despite being widely used, is problematic: one has to carefully parametrize the Lorentz permittivity. This parametrization is not necessary with a point-dipole approach for the emitter.

The overestimation of the dipole moment shown above explain why the simulations are consistent with the experimental data, despite the sign-exciton strong coupling not being achieved, as seen in Figure 2.

Finally, I want to stress that the SERS recording shown in Figure 5 of the revised manuscript does not show any evidence of picocavity events. What the authors stress as “picocavity-like”

seems rather random. Picocavity events are evidenced by enhanced vibrational lines that appear sporadically, as shown e.g. in Figure 3b in ref. 54 of the revised manuscript. As explained in this reference, "*Variations in the adatom position produce a wandering of picocavity SERS emission, alongside completely stable nanocavity lines*". From the recording, I can see a number of lines, but I cannot find the stable lines of the nanocavity, making this recording difficult to interpret. Nevertheless, this could be an exciting research direction to explore with nanodimers, which I look forward to reading about.

Editorial Note: This reviewer's edited attached Figure S13 is on the next page.

A list of Changes Made in the Revised Version of Manuscript

Change 1. We have listed Xinyi Fan as a co-author in the present version to account for his important contributions to the optical measurement in the revised manuscript (see the revised line 3 on page 1, the revised line 18 on page 36 in the current manuscript).

Change 2. To improve the language of our manuscript, the manuscript has been revised by an English speaker Editor in *Editage* (the revised parts are highlighted in red).

Change 3. We have added the information on the yields of the AuNDs, which demonstrate strong coupling in the main text of our revised manuscript (see the revised lines 8–11 on page 11).

Change 4. We have added the number of dye molecules involved in the strong coupling achieved in ref. 18 (*i.e.*, ref.16 in the current manuscript) in the revised Supplementary Table S1 (see the revised Supplementary Table S1 on page S24 of the current Supplementary Information, SI).

Change 5. We have revised the sentence "*arising from a dramatic increase in the luminous MB monomers*" and given a corresponding explanation in the revised manuscript (see the revised lines 14–19 on page 6; the added refs. 37, 38 in the revised manuscript).

Change 6. We have revised the sentence "*single-nanoparticle absorption measurement has been remained a challenge in experiment at present, owing to the incident light is always scattered by nanoparticles*" (see the revised line 21 on page 15, the revised lines 1, 2 on page 16, the added refs. 50 and 51 in the revised manuscript).

Change 7. We have added our calculation method of mode volume in the manuscript "Numerical simulations" section. In this section, we have also introduced another calculation approach based on the quasinormal mode (QNM) theory. Correspondingly, we added the two works of *ACS Photonics* 8, 307–314 (2021) and *Adv. Opt. Photonics* 12, 612–708 (2020)] as refs. 64 and 65 in our current manuscript (see the revised lines 14–22 on page 25, the revised lines 1, 2 on page 26, the added refs. 64 and 65 in the revised manuscript).

Change 8. We have added the TEM images of more AuNDs hybridized with single CB[7]@single MB emitters in the revised Supplementary Fig. S4, from which one can see the gaps in AuNDs are almost free from facets. In this revised Fig. S4, we have also added the statistics of the gap distance observed in the hybrid AuNDs (see the revised Fig. S4 in the current SI).

Change 9. We have added additional experiment results and discussions on the polarization-resolved dark-field scattering measurements of the strongly coupled AuNDs in the revised manuscript and Supplementary Information (see the revised lines 16–21 on page 21, the revised

lines 1–3 on page 22, the added ref. 60 in the current manuscript; the added Supplementary Section S7 and Fig. S18 on pages S19 and S20 in the revised SI). We have also added *J. Phys. Chem. A* **119**, 9098–9108 (2015) as ref. 40 to show the dipole direction of the MB molecule in the revised manuscript (see the added ref. 40 in the revised manuscript).

Change 10. We have replaced the photobleaching data in the revised Fig. S13a,b with new measurements. We have also replaced the green line in the revised Fig. S15a with a much better one, and its corresponding SEM image was also replaced (see the revised Fig. S13a,b and Fig. S15a in the current SI).

Change 11. We have added the quantum mechanical model used in our manuscript in the revised Supplementary Information (see the added Supplementary Section S5 in the current SI).

Change 12. We have added green arrows to indicate the nanocavity SERS lines in the revised Fig. 5b in the current manuscript.

Replies to the Reviewer's Comments

Reviewer #1

The authors have greatly improved their manuscript with numerous complementary experiments that strongly support their claims. I want to congratulate them for the nice assembly, SERS, and photobleaching experiments that they conducted. I only have very minor points, but overall, the paper should now be accepted for publication in Nature Communications.

We deeply appreciate Reviewer #1 for his or her affirmative statements. We would love to provide the required data and address all the remaining concerns of Reviewer #1 as follows.

Comment 1: *The overall manuscript should be proofread by a native English speaker as there are a large number of typos and some odd sentences.*

Our Reply: According to the Reviewer's suggestions, we have invited a native English speaker Editor in *Editage* to help us improve the language of our manuscript.

Corresponding Revision: The language revisions can be found in the revised manuscript marked in red.

Comment 2: *The authors only provide quantitative statistical information in the methods section and in the supporting information. But this information should appear in the main text: after optimizing the assembly procedure, about 20-25% of the sample actually corresponds to dimers, out of which ~15% are in strong coupling. This is clearly an improvement over previous studies, but these numbers must be stated in the main text. In the end, about 3-4% of the scatterers in the darkfield images correspond to strongly coupled dimers; it is an important fact that the readers should be clearly made aware of.*

Our Reply: According to the Reviewer's instructions, we have added the quantitative statistical information of the Au dimers and the strongly coupled dimers with dye molecules in the main text of our revised manuscript.

Corresponding Revision: We have added these the quantitative statistical information in the section "*Single-exciton strong coupling in AuNDs with different detuning*" in the main text of our revised manuscript (see the revised lines 8-11 on page 11).

Comment 3: *Table S1 gives a nice overview of prior strong coupling experiments, but I find it surprising that the data of ref 18 are not included as they also correspond to gold nanodimers as used in this manuscript.*

Our Reply: According to the Reviewer's good suggestions, we have added the data of ref. 18 in the revised Table S1.

Corresponding Revision: We have added the number of dye molecules involved in the strong coupling achieved in ref.18 (*i.e.*, ref.16 in the current manuscript) in the revised Table S1 (see the revised Table S1 on page S24 of the current Supplementary Information).

Comment 4: *The sentence "arising from a dramatic increase in the luminous MB monomers" on page 6 is unclear. Please rephrase. Maybe: "This indicates an improvement of the quantum yield of the MB dyes when hosted by CB molecules"?*

Our Reply: According to the Reviewer's good suggestions, we have revised the sentence "arising from a dramatic increase in the luminous MB monomers" as "owing to the enhancement in the fluorescence quantum yield of these monomers hosted in CB[7] cavities". We have also explained this point: "The fluorescence quantum yield and brightness of MB molecules, which are well-known to have a low fluorescence quantum yield³⁷, are enhanced due to the low polarizability inside the CB[7] cavities and the suppression of some relaxation processes of the dye molecules³⁸" We have also added refs. 37, 38 in our current manuscript.

Corresponding Revision: We have revised the sentence "arising from a dramatic increase in the luminous MB monomers" and given a corresponding explanation in the revised manuscript (see the revised lines 14-19 on page 6; the added refs. 37, 38 in the revised manuscript).

Comment 5: *the sentence "single-nanoparticle absorption measurement has been remained a challenge in experiment at present, owing to the incident light is always scattered by nanoparticles" is unclear (and gramatically incorrect). There are a number of experimental methods to assess extinction (Chem. Soc. Rev., 2014, 43, 3921-3956) or absorption (ACS nano 11 (2), 1412-1418) by single nanostructures but they are indeed more complicated than darkfield scattering. Please rephrase.*

Our Reply: According to the Reviewer's good suggestions, we have revised the sentence "..., even though single-nanoparticle absorption measurement has been remained a challenge in experiment at present, owing to the incident light is always scattered by nanoparticles" as "..., even though the experimental single-nanostructure absorption measurement is more complex than the performed dark-field scattering^{50,51} because the incident light is always scattered by the nanostructures⁵²". We have also added the references of *Chem. Soc. Rev.* **43**, 3921–3956 (2014) and *ACS Nano* **11**, 1412–1418 (2017) as the added refs. 50 and 51, respectively, in the revised manuscript.

Corresponding Revision: We have revised the sentence "*single-nanoparticle absorption measurement has been remained a challenge in experiment at present, owing to the incident light is always scattered by nanoparticles*" in the current manuscript (see the revised line 21 on page 15, the revised lines 1, 2 on page 16, the added refs. 50 and 51 in the revised manuscript).

Reviewer #2

The authors have taken care of the concerns raised in my previous review. They have provided new experimental data on the two-molecule PCA and temporal dynamics of the system supporting single-molecule evidence. The arguments on the single-molecule scale are now convincing. Having successfully addressed the issues and thus improved the quality of the work, I recommend accepting the manuscript for publication.

Our Reply: We deeply appreciate Reviewer #2 for his or her recommendation of our manuscript to be published in *Nature Communications*.

Reviewer #3

In the revised version of their manuscript, the authors provided additional experimental and numerical data, as well as corrections in the main text and supplementary information. Overall, all the results presented in this manuscript are, in my opinion, very interesting and encouraging, and point in the right direction for strong coupling between confined light and single emitters under ambient conditions. However, with regard to the publication of this manuscript in Nature Communications, I am mainly concerned by the following assertions, which I consider to be false:

- The molecule's position is deterministic.*
- The transition dipole moment can perfectly align along the plasmonic field.*
- Single exciton strong coupling with the plasmonic dimer mode is achieved.*

Therefore, I have to reject the manuscript. Nevertheless, I recognize the effort and contribution made by the authors to the field of nanophotonics, and I do believe that further work might justify a resubmission.

We appreciate Reviewer #3 for confirming our results in our revised manuscript, the significance of our work, and all the constructive suggestions to make our claim even more robust. Related to the Reviewer's final three concerns: 1) *the molecule's position is deterministic*, 2) *the transition dipole moment can perfectly align along the plasmonic field*, 3) *single exciton strong coupling with the plasmonic dimer mode is achieved*, we have carefully examined all of them and cautiously responded as follows. We sincerely hope to convince Reviewer #3 to approve our contribution to this field, warranting its publication in Nature Communications.

Comment 1: *Related to the positioning of the molecular exciton with the electric field of the dimer mode, the authors claim in Table S1 that their work provides both "emitter position certainty" and "dipole orientation certainty". Why would the position be known for certain, as compared to, e.g., single MB molecules in Au NPoM (ref. 7)? In comment 3 from the second Reviewer, the relevant point was raised that dimers can have facets. The main concern here was related to the number of molecules present in the gap, but this also means that even if there is only one molecule, its position in space is not fully known.*

Our Reply: About the 1st question, "*Why would the position of the MB molecule emitter in our coupling system be known for certain, as compared to, e.g., single MB molecules in Au NPoM (ref. 7)?*" **We argue that this is mainly related to the difference between the two preparation methods used in our manuscript and in the ref. 7.** It can be seen in the SI of ref. 7 that single MB molecules embedded in the Au NPoM were achieved **passively**. It was realized as follows [*Nature* 535, 127–130 (2016)]: firstly, the Au-coated Si substrate was submerged in a 1 mM solution of CB[7] in de-ionized water overnight to deposit a layer of CB[7]

and CB[7]@MB complexes; then, the 40-60 nm Au nanoparticles are drop cast onto the coated film where physisorption takes place and are rinsed off with de-ionized water after 10s to remove excess particles; lastly, the substrate is then blown dry using nitrogen, and individual Au nanoparticles are physically adsorbed on the surface of the Au film bound with a layer of empty CB[7] and CB[7]@single MB complexes. **Therefore, the single MB molecules can be pressed under the individual Au nanoparticles, but the number and position of MB molecules embedded in the nanogap of Au NPoM are completely random (as shown in Fig. R1a), even if the Au nanoparticles are in a perfect sphere shape without any crystal facets. In fact, however, the Au nanoparticles used in ref. 7 demonstrate obvious facets (see Fig. R2).**

Figure R1. Schematics of the hybrid NPoM and AuND cavities integrated with a single MB molecule. **a** Schematic of the hybrid NPoM cavity containing a single MB with random position (the method used in ref. 7 in the manuscript). **b** The process flows to fabricate the hybrid AuND containing a single MB with deterministic position and dipole alignment (our work).

FIGURE REDACTED

Figure R2. Au nanoparticles used in ref. 7. Scanning electron microscope (SEM) image of drop cast Au nanoparticles used in ref. 7. This figure is taken from the ref. 7 in the revised manuscript.

Unlike the fabrication approach in ref. 7, we integrated single MB molecules in individual AuNDs using the **active method** of host-guest chemistry combined with the CB[7]-mediated self-assembly, as illustrated in Fig. R1b. Specifically, we first added individual CB[7]@single MB emitters into colloidal Au nanoparticles, then CB[7] can actively bind to the surface of an Au nanoparticle through its carbonyl-fringed portals on one side (Fig. 1b(II)). When another Au nanoparticle moved close to this CB[7]@single MB functioned Au nanoparticle, the carbonyl-fringed portals on the other side of the CB[7] cage can grasp and bind to this second nanoparticle via the formation of coordination bonds. Lastly, the hybrid AuND/CB[7]@single MB with a single MB molecule integrated into the gap center (*i.e.*, the position of $E_{F_{max}}$, Fig. R1a(III)) of the AuND is formed (see Supplementary Movie S1). Therefore, **we can see that such a precise location of a single MB emitter in the gap of AuND mainly arises from the unique CB[7]-mediated active self-assembly approach**, as R. W. Taylor et al. said, the CB[n]-mediated self-assembly "*opens the exciting possibility of positioning the guests in the very center of the intense confined electric field (hot spot) for optimal sensing...Such exquisite control over both the creation of numerous exact separations and precise electromagnetic modes, and the positioning of analyte molecules is unprecedented and has not been demonstrated in any other system...*" [*ACS Nano* 5, 3878–3887 (2011)].

Even if the Au nanodimers have small facets, this method can still integrate the single MB into the nanogap. Still, it is not necessarily at the gap center, just as Reviewer#3 pointed out. Our further numerical simulations demonstrate that the electric field (EF) distribution in the nanogap constructed by two small facets (for instance, a diameter of the facet $D = 6$ nm) is almost uniform and the maximum (Fig. R3), indicating that the single MB integrated in such a nanogap with small facets can still be seemed at the position of maximum EFs, even though the

MB does not necessarily locate at gap center. In this sense, this unique method can effectively integrate the single MB in the nanogap of the AuND at the position of the intense confined electric field, *i.e.*, hot spot [ACS Nano 5, 3878–3887 (2011); Nano Lett. 12, 5924–5928 (2012)], which is essential for achieving the single-exciton strong coupling.

Figure R3. Plasmonic EF distributions in the AuND with small facets. **a** The normalized EF as a function of the position away from the gap center along the vertical direction of the AuND (*y*-axis) with a 6-nm facet. **b** In-plane vertical EF distribution in the gap center of this AuND.

We have noticed this problem often encountered in the previous literature. In our manuscript, we have tried our best to fabricate the super spherical Au nanoparticles (Fig. R4a-d) using the method of Au³⁺ oxidation etching. From Fig. R4e-l, one can see that the nanogaps in our AuND/CB[7]@single MB hybrids are constructed by smooth Au nanoparticles almost without crystal facets. Even though it is inevitable that there remains a small part of AuNDs has nanogaps with small-size crystal facets, the Au nanoparticles employed in this work are already a significant improvement over those used in previous literature (for example, the above Fig. R2, *i.e.*, Fig. S4 in the SI of ref. 7). In ref. 7, the authors calculated the electric field distribution, mode volume of the plasmonic nanocavity (NPoM nanostructure), as well as the strong coupling between the gap plasmon resonances and dye molecules by modeling the Au nanoparticles as super spherical nanospheres. We argue that it is reasonable and acceptable to model our Au³⁺-etched Au nanoparticles as perfect nanospheres in our calculations and simulations. **Such super spherical Au nanoparticles facilitate us to precisely integrate a single MB molecule in a certain position of gap center in the AuND, by using the active method of CB[7]-mediated active self-assembly.**

Figure R4. The Transmission electron microscope (TEM) images of Au³⁺-etched Au nanoparticles with different sizes and the hybrid AuNDs with single MB molecules. a-d TEM images of the super spherical Au nanoparticles with different diameters of (a) 27.0 ± 2.2 nm, (b) 38.4 ± 1.8 nm, (c) 45.4 ± 0.8 nm and (d) 57.0 ± 2.8 nm, respectively. e-l TEM images of the representative Au NDs with a fixed gap of about 0.9 nm, which are constructed by super spherical Au nanoparticles via single CB[7]@single MB emitters (at a concentration of $\bar{n}_{MB} = 0.8$ CB[7]@single MB emitter and $\bar{n}_0 = 0.8$ empty CB[7] on average per Au nanoparticle).

Last but not least, regarding the number of molecules in the gap, *i.e.*, Comment 3, previously raised by Reviewer #2, who had been convinced by our explanations and believed in our arguments on the single-molecule scale. We want to re-examine and reply to this question by providing further concrete evidence, hopefully, to convince Reviewer #3 and future readers. In our sample fabrication, there was $\bar{n}_{MB} \sim 0.8$ CB[7]@single MB emitter and $\bar{n}_0 \sim 0.8$ empty CB[7] molecule per Au nanoparticle (the molar ratio of MB and CB[7] was 1:2, a half of CB[7] molecules were empty without MB, see Methods Section in the revised manuscript). If we take $\bar{n}_{MB} = 1$ and $\bar{n}_0 = 1$ for an example, *i.e.*, there are 2 CB[7]@single MB emitters and 2 empty CB[7] molecules, on average, can be bound to an AuND. Theoretically, one can estimate the possibilities for one and two CB[7]@single MB emitters embedded in the gap of the ND are $\sim 50\%$ (the nanodimer is formed due to CB[7] binding, either through the CB[7]@single MB emitter or the empty CB[7]) and 0.28%, respectively. To be specific, considering the cross-section area ($\sim 1.8 \text{ nm}^2$) of the CB[7] and the surface area ($\sim 5024 \text{ nm}^2$) of an Au nanoparticle with a diameter of 40 nm, the probability of two CB[7]@single MB emitters simultaneously

accommodated in a same gap (*i.e.*, located in the gap region with a cross-sectional area of $28.3 \text{ nm}^2 = 3.14 \times 3^2 \text{ nm}^2$, where has almost the same most intensive electric fields, Fig. 1e in the revised manuscript) can be estimated as $\sim 0.28\%$ according to probability calculation (Fig. R5a). Assuming that two small facets construct the nanogap in an AuND, each facet is 10 nm in diameter (*i.e.*, $3.14 \times 5^2 = 78.5 \text{ nm}^2$ in area, Fig. R5b), the possibility for two CB[7]@single MB emitters simultaneously embedded in this gap will be improved to $\sim 0.78\%$. Therefore, even in the case of crystal facets, the possibility of two MB molecules simultaneously embedded in the same nanogap remains very low. This conclusion of a single molecule in AuND has also been further verified by our experiments of two-analyte SERS and time-series SERS from individual AuND/CB[7]@single MB hybrids, as shown in Fig. 2d,e and Fig. 5 in our manuscript. **Therefore, such small facets do not affect the single molecule result and its precise location in the gap of AuND when the sample is treated with dye molecules at low concentrations.**

Figure R5. Schematics of two MB molecules simultaneously embedded in the gap of the AuND with and without facets. **a** Schematic of CB[7]@single MB emitters and empty CB[7] molecules bound on an AuND constructed by two perfect Au spherical nanoparticles. The possibility for two CB[7]@single MB emitters embedded in the gap of the nanodimer is $p = 50\% \times (C_3^1 \times 1/3 \times 28.3/5024) \approx 0.28\%$. **b** Schematic of CB[7]@single MB emitters and empty CB[7] molecules bound on an AuND constructed by two Au nanoparticles with small facets. The possibility for two CB[7]@single MB emitters embedded in the gap of the AuND is $p = 50\% \times (C_3^1 \times 1/3 \times 78.5/5024) \approx 0.78\%$.

Corresponding Revision: We have added the TEM images for more AuNDs in the Supplementary Fig. S4, from which one can see the gaps in AuNDs are almost free from obvious facets (see the revised Fig. S4 in the current Supplementary Information, SI).

Comment 2: *Related to the alignment of the molecular exciton with the electric field of the dimer mode, a critical point was raised in comment 2 of the second Reviewer: "it is very likely for some parts of the MB molecule to extend beyond the hydrophobic portal of CB[7], as MB is lengthier". This comment was associated with the question whether this could prevent the CB[7] to bind the two nanoparticles together. If this is indeed true that the CB[7]@single MB binds the two nanospheres together, then, as the authors said, "the MB molecule chain could be bent or compressed to some extent in the gap of the AuND" so that the C=O functional groups of the CB[7] can access the gold surface. If the molecule is folded or compressed, how can we be sure that the transition dipole moment is always oriented along the axis of the dimer? In fact, how can we be sure that the transition dipole moment has the same value as the "free" molecule? My feeling is that we can only relax this assumption and try to examine the coupling experiment at different illumination angles. Reading the manuscript, I get the impression that the authors are relying a little too much on previous work. In NPoM structures, the plasmonic field is oriented perpendicular to the substrate, while it is parallel in the case of AuNDs, for which it would be interesting to see polarization-resolved dark-field scattering as e.g., in Stührenbger et al., Nano Lett. 2018, 18, 9, 5938–5945.*

Our Reply: As Reviewer #3 pointed out, we had previously replied to the question, "*it is very likely for some parts of the MB molecule to extend beyond the hydrophobic portal of CB[7], as MB is lengthier, also seen in Fig.1. This further obstructs the binding of CB[7] to Au nanodimers*", which was raised by the Reviewer #2 in his or her previous Comment 2. In our response, we indeed argued that the MB molecule chain could be bent or compressed to some extent in the gap of the AuND so that the C=O functional groups of the CB[7] can access the gold surface and the hybrid AuND/CB[7]@single MB can be successfully constructed with a fixed interparticle separation of ~0.9 nm, as shown in Fig. S4 of our Supplementary Information. We feel pleased that Reviewer #2 has accepted our explanations; we would like to provide further detailed explanations regarding this question.

Figure R6. Schematic of a single MB molecule embedded in a CB[7] cage.

Fig. R6 shows that the four methyl groups ($-\text{CH}_3$, the groups in the dashed green ellipses) at two ends of the MB molecule are parts extending beyond CB's hydrophobic portal [7]. When the CB[7]@single MB emitter actively binds to the Au nanoparticle via the $\text{C}=\text{O}$ groups of CB[7], the four groups of $-\text{CH}_3$ will be bent first, making the chemical adsorption of $\text{C}=\text{O}$ to Au atoms more easily. However, we argue that such a fold or compression of the MB molecule will not significantly affect the transition properties of the MB molecule **because the transition at ~ 660 nm of the MB molecule is determined by its chromophore, i.e., the central ring of the MB molecule.** As reported in the literature, the absorption transitions at ~ 660 nm of the MB molecule are related to the electronic transition of $\text{S}_0\text{-S}_1$, i.e., HOMO-LUMO (see Fig. R7) [*J. Phys. Chem. A* 119, 9098–9108 (2015)]. Comparing HOMO and LUMO of the molecule, it can be seen that the redistribution of the electron density upon photoexcitation indeed "occurs in the central ring of the chromophore in the MB molecule" [*J. Phys. Chem. A* 119, 9098–9108 (2015); *Journal of Molecular Liquids* 336, 116369 (2021)]. Therefore, we argue that the somewhat fold or compression of the methyl groups at two ends of the embedded MB molecule will not significantly affect its chromophore, indicating that the transition dipole moment (TMD, **along the longitudinal direction of the MB molecule**, see Fig. R7) of the MB molecule will not be significantly influenced. **Therefore, it will not significantly influence the alignment of the transition dipole moment of the MB molecule when it is embedded in the gap of AuND.** Of course, it is also unobjective to say there is no influence on the transition dipole moment of the embedded MB molecule compared to that of the "free" molecule. **Thus, according to the Reviewer's suggestion, we have revised the statement of "perfectly aligning the excitonic dipole moment" as "a fine alignment of the excitonic dipole moment with the plasmonic field".** On the other hand, the fine alignment of the dipole moment of dye molecules (in CB[7]s) with the gap plasmonic mode has been verified and discussed in many previous investigations [*Nature* 535, 127–130 (2016); *Nature Photonics* 17, 865–871 (2023); *ACS Nano*. 5, 3878–3887 (2011)]. Thus, it is not just an assumption.

FIGURE REDACTED

Figure R7. Frontier orbitals for the transition in MB with the $\text{S}_0\text{-S}_1$ transition dipole moment (TDM) direction denoted. HOMO: Highest occupied molecular orbital; LOMO: Lowest unoccupied molecular orbital. The figure is taken from *J. Phys. Chem. A* 119, 9098–9108 (2015).

Lastly, we are grateful to Reviewer #3 for his/her suggestion on carrying out the polarization-resolved dark-field scattering measurements of the strongly coupled AuNDs with MB-molecule excitons (Fig. R8, *i.e.*, the added Fig. S18 in the revised Supplementary Information). As the Reviewer has pointed out, the plasmonic field is oriented perpendicular to the substrate in NPoM structures, while it is parallel to the dimer direction in AuNDs, for which it would be interesting and more suitable to see the polarization-resolved dark-field scattering of the plexcitons. In the revised manuscript, we have managed to perform the additional experiments of polarization-resolved dark-field scattering measurements of the strongly coupled AuNDs with MB-molecule excitons, as the Reviewer suggested. In our measurements, the linear polarizer in the collection path of the microscope was rotated, and a spectrum was collected at each angle from 0° to 90° , corresponding to the longitudinal (horizontal arrow) and transverse (vertical arrow) polarizations (the insets in Fig. R8), respectively.

Figure R8. Polarization dependence of plexcitonic properties on a single hybrid AuND strongly coupled with MB-molecule excitons. **a, b** Polarized scattering spectra for two cases of the strongly coupled AuNDs with MB-molecule excitons, which were detected at polarization angles of 0° , 15° , 30° , 45° , 60° , 75° , and 90° , respectively. The scale bar is 100 nm. **c, d** Relative scattering intensity of the UPB (blue balls) and LPB (red balls) modes as a function of the polarization angle for the two cases in **(a)** and **(b)**. Each polarized spectrum was normalized to its maximum scattering amplitude to compare the relative amplitudes of the UPB and LPB modes. The dashed circles in **(c)** and **(d)** means that the LPBs at these polarization angles are vanished.

Figure R8a, b gives the polarization-resolved dark-field scattering of the strongly coupled AuNDs with two and three MB-molecule excitons, respectively, which demonstrates a progressive emergence of the two plexcitonic states as the polarization is rotated from transverse (90°) to longitudinal (0°). As the polarization angle decreases from 90° to 0° , the coupling between the gap plasmon mode and exciton becomes stronger and stronger, and the LBP mode appears gradually on the red side of the exciton transition (660 nm/1.88 eV), which agrees with the observations reported in the literature [*Nano Lett.* 2013, 13, 3281–3286]. Interestingly, it is also found that the UBP and LPB modes have a different polarization dependence, as shown in Fig. R8c, d, which may effectively manipulate these two hybrid states. We will perform further systematic researches on this issue in the near future; we are grateful to Reviewer #3 for pointing us this interesting research direction based on the strongly coupled AuND/CB[7]single MB systems.

Corresponding Revision: We have added new experimental results and discussions on the polarization-resolved dark-field scattering measurements of the strongly coupled AuNDs with MB-molecule excitons in the revised manuscript and Supplementary Information (see the revised lines 16–21 on page 21, the revised lines 1–3 on page 22, the added ref. 60 in the current manuscript; the added Supplementary Section S7 and Fig. S18 on pages S19 and S20 in the revised SI). We have also added *J. Phys. Chem. A* **119**, 9098–9108 (2015) as ref. 40 to show the dipole direction of the MB molecule in the revised manuscript (see the added ref. 40 in the revised manuscript).

Comment 3: *Another point, which is the most critical, relates to the single exciton strong coupling claim. The energy difference of 116 meV observed between the two peaks in the scattering spectrum cannot be interpreted as strong coupling between a single MB molecule and the plasmonic mode of the nanodimer. This can be confirmed both by calculations and the measured scattering before and after photobleaching the MB molecules in the gap of the AuNDs, Figure S13 in the revised supplementary information. Bleaching the molecule should eliminate the Rabi splitting in the case of strong coupling, but this is not what is observed here. In Figure S13a, we clearly see that the two peaks are still there and that a transparency dip has been suppressed. But such a transparency dip cannot be associated to strong coupling. In panels b and c, we see a peak being suppressed, but no closing of the gap between the peaks is observed. The suppressed peak is maybe due to Purcell-enhanced emission of the molecule, but this is definitively not strong coupling. To highlight my point, I have added vertical lines in Figure S13, see attached image. In addition, in panel b, an extra peak is seen, and remains after bleaching. This peak is likely to be part of the bare AuND. In fact, similar additional shoulders*

are seen in bare AuNDs in Figure S15a. Note that the green line has a shoulder that is not seen in simulations. Such features may be due to imperfections in the morphology?

Our Reply: We appreciate the effort made by Reviewer #3 for carefully examining our photobleaching data in Fig. S13. We have demonstrated the photobleaching data (in Fig. S13) in reply to Comment 4 raised by Reviewer #1 in previous comments to verify that the observed energy splitting is indeed due to the MB molecule. After we provided these bleaching data for the observed energy splitting (Fig. S13 in the previous Supplementary Information), Reviewer #1 highly recognized us for our photobleaching experiments. However, we realize that there may remain some confusing points, according to the comments raised by Reviewer #3. The main doubt of the Reviewer arising from the left peak of the spectral Rabi splitting (SRS) in Fig. S13 may be due to the Purcell-enhanced emission of the MB molecule but not from the strong coupling. We would like to clarify as follows.

To give more clear pieces of evidence to support our conclusion in the manuscript, firstly, we reformed the photobleaching experiments of the hybrid AuNDs strongly coupled to single MBs, and the representative photobleaching data can be found in the following Fig. R9 (and the revised Fig. S13a,b in the current SI). From these new data, one can clearly see that the bleaching has successfully eliminated the spectral Rabi splitting in the cases of strong coupling. **The restored gap plasmon mode is not at any position of the two splitting peaks, proving that the observed energy splitting is due to the MB molecule and the observed SRS arising from the single exciton strong coupling.**

Secondly, related to the Reviewer's question, "*the suppressed peak is maybe due to Purcell-enhanced emission of the molecule*", we argue that **the suppressed peak is not the enhanced emission of the MB molecule because the suppressed peak is at ~616 nm, which is far away from the emission of the MB molecule at ~680 nm** (see Fig. S2a in SI). We guess that it means the Purcell-enhanced scattering of the molecule because what we have measured in Fig. S13 is scattering. Even so, we argue that it is also unrealistic because **the suppressed peak is still far away from the extinction of the MB molecule at ~660 nm**. Maybe our data in the original Fig. S13 mislead the Reviewer to think that the scattering spectra in Fig. S13 are a simple superposition of the enhanced scattering from the MB molecule and the scattering of the gap plasmon mode supported by AuND but do not from strong coupling. If the SRSs shown in Fig. S13a-c are a superposition of two Lorentz spectra (*one is the enhanced scattering spectrum of the MB, and the other is the scattering spectrum of AuND*), as the Reviewer has pointed out, there would be another phenomenon that mismatches with our observations. Namely, the anti-crossing energy dispersions of the UPB and LPB induced by strong coupling (Fig. 2b in our manuscript) cannot be observed, owing to the exciton transition energy (corresponding to the

suppressed peak in Fig. S13d-f, *i.e.*, the left peak as the black arrows in Fig. S13a-c) will keep unchanged during the gap plasmon mode detuning from it (Fig. R10). Especially when the two coupling subsystems are brought into resonance, there will be a single enhanced peak but not a spectral Rabi splitting. **It can be seen that such a dispersion is completely different from our experimental observations (Fig. 2b).** More importantly, the new data shown in Fig. R9 clearly show that the two splitting peaks all vanished after photobleaching. The transparency dip has been completely suppressed, **indicating that the suppressed peaks are not due to the Purcell-enhanced emission of the embedded molecule.**

Figure R9. New data for photobleaching of the strongly coupled AuNDs with single MB-molecule excitons. **a, b** Normalized scattering spectra of two hybrid AuNDs strongly coupled with single MB molecules, before (red curves) and after (green curves) photobleaching induced by laser illumination. Insets show SEM images of the measured hybrid Au NDs.

Figure R10. Theoretical spectra calculated for the superposition of two Lorentz spectra. **a** Theoretical spectra calculated for the superposition of two Lorentz spectra (one is the enhanced scattering spectrum of the MB molecule, and the other is the scattering spectrum of gap plasmon mode) with different detuning (δ). **b** Normalized theoretical spectra calculated for the superposition of the two Lorentz spectra ordered according to the detuning ($\delta = \hbar\omega_d - \hbar\omega_c$).

Lastly, we also want to explain further the photobleaching data observed in Fig. S13. Regarding the question "*In Figure S13a, we clearly see that the two peaks are still there and that a transparency dip has been suppressed, but such a transparency dip cannot be associated to strong coupling*" we argue that this case arises from strong coupling. From Fig. S13a(I), one can see that the SRS is about 190 meV, corresponding to a three-molecule-exciton strong coupling case. After bleaching, only a fraction of dye molecules was bleached (not a complete bleaching), leading to a reduction of the coupling strength between the gap plasmon mode and unbleached dye molecules, **manifested as a reduction of the SRS width and a suppression (not completely) of the transparency dip** (see Fig. S13a(II)), just as the Reviewer #1 has pointed out in the previous Comment 4. To give more clear evidences, we have replaced the photobleaching data in the revised Fig. S13a, b with the new measurements in above Fig. R9 a, b, respectively.

The original Figure S13. Photobleaching of the strongly coupled AuNDs with MB-molecule excitons. Scattering spectra of individual Au ND/CB[7]@MB hybrids before (I) and after (II) photobleaching induced by laser illumination. Insets show SEM images of the measured hybrid Au NDs.

Relate to the question "*In panels b and c, we see a peak being suppressed, but no closing of the gap between the peaks is observed*", we believe that the scattering spectrum shown in Fig. S13b(II) is a completely bleaching case of the strongly coupled system. That is to say, the gap between the two peaks in Fig. S13b(I) is closed (see the dashed gray line), indicating that the transparency dip in Fig. S13b(I) arising from the dye molecule's strong coupling with plasmons.

What's less desirable is that the restored gap plasmon mode in Fig. S13b(II) is just in the position of the right peak in Fig. S13b(I) (see dashed blue line), which may cause a misleading to the Reviewer#3 that the right peak in Fig. S13b(I) is the gap plasmon mode. A similar case is shown in Fig. S13c, despite the restored gap plasmon mode (Fig. S13b(II)) having a redshift compared to the position of the right peak in Fig. S13b(I)). Relate to the two cases shown in Figs. S13b(II) and S13c(II), **we argue that the restored gap plasmon mode has a redshift after photobleaching compared to the original gap plasmon mode supported by AuND in the strongly coupled systems.** It may arise from the laser-induced damage in the morphology of the AuND, including its nanogap, resulting in redshift or blueshift of the restored gap plasmon mode. Notice that such frequency shift of the restored plasmon mode can also be seen in some cases of the photobleaching measurements (G. Zengin et al., *Sci. Rep.* 3, 3074 (2013); *Phys. Rev. Lett.* 118, 237401 (2017)). Therefore, some works attempted other methods to determine the bare plasmon energy in the strongly coupled systems rather than photobleaching [*Nature* 535, 127–130 (2016); *Optica* 8, 1416–1423 (2021); *Nano Lett.* 22, 4686–4693 (2022); and this work].

Relate to the question, "*In addition, in panel b, an extra peak is seen, and remains after bleaching. This peak is likely to be part of the bare AuND. In fact, similar additional shoulders are seen in bare AuNDs in Figure S15a. Note that the green line has a shoulder that is not seen in simulations. Such features may be due to imperfections in the morphology?*", we agree with Reviewer's viewpoint. Indeed, the shoulder peak in the measured scattering spectrum in the original Fig. S13b is a part of the bare AuND; we think that it belongs to the quadrupolar resonance mode or the transverse resonance mode of the bare AuND, owing to the two resonance modes are very close in frequency in our AuNDs [*J. Phys. Chem. C*, 113, 4349–4356 (2009)]. However, the additional shoulder peaks in the bare AuNDs in Figure S15a are the quadrupolar resonance modes of the NDs because we calculated these scattering spectra using a plane wave with longitudinal polarization (see the right panel in Figure S15b). Despite it being much weaker than that of the gap plasmon resonance mode (i.e., the dipolar resonance mode of AuND), this shoulder peak can also be observed in some bleached coupling systems (the green arrow in Fig. S13b(II)). Of course, this weak mode may not be clearly seen in some cases, influenced by the size of Au nanoparticles and the microenvironment around the AuNDs. Relate to "*the green line has a shoulder that is not seen in simulations*", we guess it refers to the weak shoulder peak at ~750 nm of the green line in original Fig. S15a, which cannot be seen in simulations (Fig. S15b). We argue that this weak shoulder is an influence caused by the scattering of the microenvironment where the AuND resides. Of course, it also may be due to some imperfection in the morphology of the AuND, as the Reviewer has pointed out. In the

revised Fig. S15a, we have replaced the green line with a better one and the corresponding SEM image. According to the above discussions, one can see that the SRS of 116 meV observed in our manuscript is indeed due to the single-molecule exciton strong coupling.

The original Figure S15. Experimental and simulated scattering spectra of the bare AuNDs with different particle sizes. (a) Normalized scattering spectra of individual bare AuNDs with different particle sizes located on ITO-coated glass substrate show the plasmon resonance wavelength varying from 636 to 708 nm. The right panel gives the SEM images of the measured samples related to the scattering spectra shown in the left panel with the same line color as the dotted box around the SEM images. (b) Normalized scattering spectra of the bare AuNDs located on ITO-coated glass substrate (with a fixed gap distance of 0.9 nm) as a function of the particle size (diameter, D) varying from $D = 30$ to 54 nm, calculated using the FEM. The right panel gives the schemes of the calculated AuND structures related to the scattering spectra shown in the left panel with the same line color as the dotted box around these schemes

The revised Figure S15a. Normalized scattering spectra of individual bare Au NDs with different particle sizes. The AuNDs were located on an ITO-coated glass substrate, which show the plasmon resonance wavelength varying from 636 to 708 nm. The right panel gives the SEM images of the measured samples related to the scattering spectra shown in the left panel with the same line color as the dotted box around the SEM images.

Corresponding Revision: We have replaced the photobleaching data in the revised Fig. S13a,b with new measurements. We have also replaced the green line in the revised Fig. S15a with a much better one, and its corresponding SEM image is also replaced (see the revised Supplementary Fig. S13a, b and Fig. S15a in the current SI).

Comment 4: *To show more clearly that the energy difference is not strong coupling, let us do a simple calculation. The coupling strength g is given by equation (10) in the added reference 40. This equation may look intimidating, but in fact it can be linked with the Purcell factor at resonance F_P , yielding the much simpler form $g = 1/2 \sqrt{\gamma_{MB} \Gamma_d F_P}$. This formula can also be found in Lalanne et al., *Laser Photonics Rev.* 2018, 12, 1700113 and Gerard, *Top. Appl. Phys.* 2003, 90, 269. Here $\gamma_{MB} = n \mu_{MB}^2 \omega_{MB}^3 / (3 \hbar \pi \epsilon_0 c^3)$ is the spontaneous emission rate of the MB molecule in a dielectric medium (here $n = 1.27$ to account for surface roughness of the substrate) and Γ_d is the plasmon decay rate of 135 meV provided by the authors. Since we now have the Purcell factor at resonance $F_P = 3.2 \cdot 10^6$, we find a coupling strength $g \approx 43$ meV, that is below the extracted value of 56 meV by the authors. This new value is below the condition $2g > (\Gamma_d + \Gamma_c)/2$, but it is approaching the strong coupling regime, which is encouraging.*

Our Reply: We are very grateful to Reviewer #3 for recalculating the coupling constant (g) of our single-exciton strong coupling system using the point-dipole model. **We argue that the discrepancy of g values obtained via these two methods (the point-dipole model and extracting from the measured spectrum) is mainly because of insufficient consideration of actual experiments when using the point-dipole model.** It can be seen that the theoretical model that the Reviewer suggested is essentially equal to what we have used for calculating the local coupling interaction in nanostructures [*Phys. Rev. B* 87, 195138 (2013)], in which the coupling strength (g) has been given as: $g = \frac{1}{2} \sqrt{\Gamma_0 \cdot \Gamma_d \cdot M}$, where $\Gamma_0 = \omega_0^3 \mu_0^2 / 3 \pi \hbar \epsilon_0 c^3$ is the spontaneous emission (SE) rate of the two-level quantum emitter (QE) in vacuum [*Phys. Today* 42, 269–276 (1989); *Topics Appl. Phys.* 90, 269–315 (2003)] (for our experiment, the molecule is in a dielectric medium, $n = 1.26$, that is $\Gamma = n \omega_c^3 \mu_c^2 / 3 \pi \hbar \epsilon_0 c^3$), the same as that the Reviewer has suggested above); $M(\mathbf{r}_c, \omega = \omega_c, \boldsymbol{\mu}_c) = M_{peak}$ is the multiplication factor of the projected local

density of states (PLDOS) at the exciton transition position ($\omega = \omega_c$) and in the spatial position (\mathbf{r}_c) of the emitter [*i.e.*, the Purcell Factor., *Europhys. Lett.* **35**, 265 (1996); *Phys. Rev. E* **58**, 3896 (1998)]. When we substitute the parameters into the above formula, a smaller coupling constant of $g \approx 43$ meV can be obtained, corresponding to a very small SE rate of $\Gamma_0 \sim 2.57 \times 10^7$ Hz for the MB molecule in a dielectric medium of $n = 1.26$.

Here, we haven't considered the influence of the CB[7] cavity on the SE rate of the encapsulated MB molecule in the above calculations. Just as Reviewer#1 has pointed out in Comment 4 and our corresponding response, one can see that **the presence of the CB[7] cavity can augment the fluorescence quantum yield and brightness and, therefore, enhance the SE rate (Γ_0) of the encapsulated MB molecule.** It is well-known that MB molecules have a low fluorescence quantum yield [*Photodiagn. Photodyn. Ther.* **2**, 175–191 (2005)], but when these MB monomers are hosted in CB[7] cavities, their fluorescence quantum yield and brightness can be largely enhanced due to the low polarizability inside CB[7] cavities and the suppression of some relaxation processes of the dye molecules [*Int. J. Photoenergy* **7**, 133–141 (2005)]. Such a conclusion can also be verified by our fluorescence measurements of the CB[B]@single MB emitters in Fig. S2a in SI, which demonstrate the fluorescence intensity (*i.e.*, the fluorescence quantum yield) of the MB monomers encapsulated in CB[7] molecules was largely enhanced by more than three times (~ 3.6 times). **Therefore, we argue that the above estimated SE rate ($\Gamma_0 \sim 2.57 \times 10^7$ Hz) of the MB molecule is smaller than that embedded in CB[7] as we have performed in experiments, leading to an underestimation of the estimated coupling constant of $g \sim 43$ meV.**

We have been aware of this problem in our investigations. We have also developed a fully quantum mechanics approach for describing the strong coupling between a plasmon mode and a QE at ambient conditions, where we have considered the finite lifetimes of the plasmons and the QE in a realistic model by introducing the inelastic interactions between the coupling system and its environment (a continuum of modes) [*Phys. Rev. B* **103**, 235430 (2021)]. By using the methods of retarded Zubarev Green's functions, the extinction spectrum of the strongly coupled plasmon-QE system observed from the plasmon-channel can be given by,

$$\sigma(\omega) \propto -\text{Im} \frac{(\hbar\omega - \hbar\omega_c + i\Gamma_c/2)}{(\hbar\omega - \hbar\omega_d + i\Gamma_d/2) \cdot (\hbar\omega - \hbar\omega_c + i\Gamma_c/2) - g_{dc}^2}, \quad (\text{R1})$$

where Γ_d (Γ_c) is the damping linewidth of the plasmon mode (QE) at room temperature with considering the inelastic interactions between the coupling system and the environment, and these parameters can be determined by experimental measurements. By setting $d\sigma(\omega)/d\omega = 0$,

the analytical expressions of the SRS observed and their critical criteria can be respectively obtained as,

$$\hbar\Omega_R = 2\sqrt{g_{dc}(1+\Gamma_c/\Gamma_d)\cdot(g_{dc}^2+\Gamma_c\Gamma_d/4)^{1/2}-(g_{dc}^2+\Gamma_c\Gamma_d/4)\cdot\Gamma_c/\Gamma_d} \quad \text{if } g_{dc}^2 > \frac{\Gamma_c^2}{8(1+\Gamma_d/2\Gamma_c)} \quad (\text{R2})$$

Apparently, the SRS is usually not equal to the level splitting ($\hbar\Omega_L = 2\sqrt{g_{dc}^2 - (\Gamma_c - \Gamma_d)^2/16}$, if $g_{dc}^2 > (\Gamma_c - \Gamma_d)^2/16$), except for the case of $g_{dc} \gg \max\{\Gamma_d, \Gamma_c\}$. In the traditional atomic and solid-state systems operated at cryogenic temperatures and high vacuum, the losses (Γ_d, Γ_c) of the coupling system are very small, and strong coupling conditions can be well satisfied so that the SRS can faithfully reflect the level splitting. However, this condition can not be well satisfied, especially in the single emitter coupling at room temperatures, due to the large decays of plasmon mode and emitter in ambient conditions. **It should be mentioned that the above Eqs. (R1) and (R2) provide an effective and convenient approach to extract the coupling constant (g) based on the experimental measured SRS, Γ_d and Γ_c , which is not necessary to calculate the mode volume or Purcell factor at first.** Such extracting methods have been extensively employed in the previous literature [*Sci. Rep.* 3, 3074 (2013); *Nat. Commun.* 7, 11823 (2016); *Sci. Adv.* 4, eaar4906 (2018); *Nat. Commun.* 9, 4012 (2018); *Opt. Express.* 18, 23633 (2010)]. Therefore, the extracted value of $g_{dc} = 56.6$ meV based on the measured Γ_c is completely reasonable, reflecting the actual experiment conditions. It is also reasonable and understandable that there is some discrepancy between the g values calculated using the formula of $g = \sqrt{\Gamma_0 \cdot \Gamma_d \cdot M} / 2$ and that extracted from the above Eq. (R2), owing to the Γ_0 used in $g = \sqrt{\Gamma_0 \cdot \Gamma_d \cdot M} / 2$ is the SE rate of an ideal two-level system in an electromagnetic vacuum, without considering the influence of the interaction between QE and environment, including the CB[7] cavity. While Γ_c we used in our manuscript comes from the experimental measurements at ambient conditions (including the broadening of QE's linewidth induced by the interaction between the QE and environment), which is also a popular approach extensively used in recent experimental investigations [*Nature* 535, 127–130 (2016); *Nat. Commun.* 7, 11823 (2016); *Nano Lett.* 17, 3809–3814 (2017); *Phys. Rev. Lett.* 130, 143601 (2023); *Nano Lett.* 22, 4686–4693 (2022); *Nano Lett.* 17, 4689–4697 (2017)].

In short, the experimental measurements agree well with the quantum theory model and numerical simulations in our manuscript, demonstrating good self-consistency and certainty of the single-molecule exciton strong coupling achieved in our hybrid AuND/CB[7]@single MB system.

Corresponding Revision: We have added the quantum mechanical model used in our manuscript in the revised Supplementary Information (see the added Supplementary Section S1 in the current SI).

Comment 5: *Now, looking at the simulations provided by the authors, with the MB molecule being modeled as a cylinder with a Lorentz permittivity, a radius of 0.75 nm and a height of 0.9 nm, I do see an overestimation of the molecule transition dipole moment. The authors mention that they took the value of $f=0.27$ to account for the small volume of the molecule, even though the formula (R3) yields $f=0.41$. But in fact, this formula does not depend on the volume of the emitter. In a recent arxiv by the group of Kristensen, a formula rigorously establishing the link between the Lorentz permittivity of the effective medium of the emitter and its volume has been shown (see equation (14) in arxiv:2308.12957). If we use the volume of 1.59 nm^3 of the CB[7] and $f=0.27$, we find a dipole moment of about $0.15 \text{ e} \cdot \text{nm}$, which largely overestimates the dipole moment of the molecule ($0.09 \text{ e} \cdot \text{nm}$). In fact, why not use the volume of the MB molecule itself? Such molecules are about 1.4 nm long and 0.95 nm wide (Jia, Puqi, et al. Applied sciences 8.10 (2018): 1903). If we assimilate the MB molecule with a cylinder of height 1.4 nm and radius $0.95/2 \text{ nm}$, and we keep the dipole moment value of $0.09 \text{ e} \cdot \text{nm}$, we can use the same formula to compute the oscillator strength, which is now $f \approx 0.16$. If the authors insist in using the volume of the CB[7], then we obtain $f \approx 0.10$... This is why this method, despite being widely used, is problematic: one has to carefully parametrize the Lorentz permittivity. This parametrization is not necessary with a point-dipole approach for the emitter. The overestimation of the dipole moment shown above explain why the simulations are consistent with the experimental data, despite the single-exciton strong coupling not being achieved, as seen in Figure 2.*

Our Reply: As the Reviewer mentioned, there is another formula that has established a link between the transition dipole moment of the emitter and its oscillator strength, where the emitter's volume has been directly included (Eq. (14) in arxiv:2308.12957). Indeed, a smaller $f \approx 0.10$ can be estimated by using this formula with setting the volume of the CB[7] as 1.59 nm^3 and $\mu = 0.09 \text{ e} \cdot \text{nm}$. According to the Reviewer's suggestion, we restimulated the coupling AuND/CB[7]@single MB system by modeling the emitter as a dielectric cylinder of height 1.4 nm and radius $0.95/2 \text{ nm}$, where f was set as 0.16. From the simulated results in Fig. R11, one can see that when the gap of AuND enlarges to 1.4 nm, the gap plasmon mode has a blueshift to $\sim 630 \text{ nm}$, far detuning from the exciton absorption at $\sim 660 \text{ nm}$, which hinders the strong coupling between the AuND and dye molecule. It is also worth noting that the gap of 1.4 nm in individual AuND/CB[7] @single MB hybrids has seldom been observed in our morphology characterizations (see the following Fig. R12). Therefore, we decide to continue using the original model of the emitter (a dielectric cylinder with 0.75 nm in radius and 0.9 nm in length)

in our revised manuscript. As for the oscillator strength, f , our simulated results show that when we adopt $f = 0.27$ as R. Chikkaraddy *et al.* have used in *Nature* 535, 127–130 (2016), the simulated SRS ($\hbar\Omega_R$) can well reproduce our measurements.

Figure R11. Simulated scattering spectrum of the hybrid AuND/CB[7]@single MB with a cylinder height of 1.4 nm. The hybrid AuND/CB[7]@single MB is located on an ITO-coated glass substrate. A large detuning between the gap plasmon resonance (ω_d) and the exciton transition (ω_c) can be seen.

Figure R12. Statistics of gap distance (d) for the individual AuNDs hybridized with single CB[7]@single MB emitters. One can see that most gaps constructed by CB[7]@single MB emitter are at a fixed value of $d \sim 0.9$ nm, and the mean value of the observed gap distances is $d \sim 0.88 \pm 0.01$ nm.

Such a discrepancy between the transition dipole moment (μ) of the emitter used in simulations and the oscillator strength (f) estimated based on μ (according to Eq. (14) in

arxiv: 2308.12957) is very common [*Nano Lett.* 13, 3281–3286 (2013); *Nature* 535, 127–130 (2016); *Phys. Rev. Lett.* 114, 157401 (2015); *Phys. Rev. Lett.* 118, 237401 (2017)]. For example, for a single CdSe/ZnS quantum dot (QD) with a transition dipole moment 50 D (~ 1.04 e nm) [*PNAS.* 112, 12288 (2015); *Nano Lett.* 22, 4686–4693 (2022)] and a volume of ~ 300 nm³, one can obtain the f of this QD is only about 0.08 according to Eq. (14) in arxiv: 2308.12957. However, to reproduce the strong coupling between a single (CdSe/ZnS) QD and the plasmonic mode, K. Santhosh *et al.* employed $f = 0.6$ [*Nat. Commun.* 7, 11823 (2016)], H. Leng *et al.* used $f = 0.8$ [*Nat. Commun.* 9, 4012 (2018)], and J -Y Li *et al.* used $f = 0.85$ to simulate their strong coupling systems [*Nano Lett.* 22, 4686–4693 (2022)]. **It means that such a discrepancy does not mean that these experiments haven't entered the strong coupling regime.**

We argue that two reasons may have led to this discrepancy. The first reason is the applicability of Eq. (14) in arxiv: 2308.12957 in these experiments. Before the advent of the quantum mechanics model, H. A. Lorentz developed a classical model for the atomic polarizability description [*principles of nano optics*, 2ed, 2012, page: 249, 527]. The model Lorentz considers consists of a collection of harmonic oscillators for the electrons of an emitter. **In Lorentz's theory, the oscillator strength is a fitting parameter since there is no direct way to know how much an electron contributes to a particular atomic or molecule mode.** Later, with the development of quantum mechanics, the relationship between the oscillator strength and the transition dipole moment was deduced based on comparing quantum and classical model results. The oscillator strength is often written as [*Atom photon interaction: basic processes and applications*, page: 515-518, 524-526 (1992); *principles of nano optics*, 2ed, page: 523-527 (2012)]

$$f = \frac{2m\omega_{eg}}{\hbar} \left| \int \varphi_e^*(\mathbf{r}) \mathbf{d}(\mathbf{r}) \varphi_g(\mathbf{r}) d^3\mathbf{r} \right|^2 \quad (\text{R3})$$

From this formula, we can see that the oscillator strength is closely related to the wavefunctions of the emitter. For multi-electron emitter, we should consider the results of the weighted sum of multiple electron wave functions on the oscillator strength. The wavefunctions of the emitter are distributed within a certain volume in three-dimensional space, which we think depends on the emitter's volume. **Thus, it cannot be said that the formula we have used (*i.e.*, the Eq.**

(R3), $f_0 = \frac{(2\pi)^2 m_e \omega_c |\mu_c|^2}{3e^2 \hbar}$, in our previous response) does not depend on the volume of the

emitter. For example, suppose we use the volume of MB, $1.43 \text{ nm} \times 0.61 \text{ nm} \times 0.4 \text{ nm} \approx 0.35$ nm³ (Carbon 44, 1884-1890 (2006)). In that case, one can calculate $f \approx 0.45$ using Eq. (14) in arxiv: 2308.12957, agreeing well with our previous estimation of $f \approx 0.42$ using the formula (R3) in our previous response. From the above Eq. (R3), one can also see that the oscillator

strength (f) not only depends on the emitter's volume but also relates to the emitter's shape which can influence the spatial distribution of electron wave functions and finally affect the oscillator strength. **Therefore, it isn't easy to give a precise link between the transition dipole moment of an actual emitter used in the experiment and its oscillator strength only via the emitter's volume.**

The second reason is that Reviewer#3 has pointed out that although such a numerical simulation approach has been widely used, it is somewhat problematic. **In these simulations, the QE's oscillator strength is usually a qualitative and adjustable parameter used to reproduce the experimental observations, it is hard to be precisely quantified and determined in real experiments.** The discrepancies we listed in the above examples can further confirm this point. On the other hand, although this classical approach is suitable and convenient to simulate the optical response of the plasmon-exciton strong coupling systems, it cannot completely describe the interference and couplings between quasi-particles such as plasmons and excitons [*Nano Lett.* 2011, 11, 2318–2323]. Therefore, new theoretical tools have been developed to overcome such limitations [*Nano Lett.* 2011, 11, 2318–2323]. Thus, we have developed a fully quantum a fully quantum approach [*Phys. Rev. Lett.* 118, 237401 (2017); *Phys. Rev. B* 103, 235430 (2021)] to well describe the optical response of the strong coupling system in our manuscript.

Corresponding Revision: We have added the statistics of the gap distance in Supplementary Fig. S4 (see the revised Fig. S4(l) in the current SI).

Comment 6: *Finally, I want to stress that the SERS recording shown in Figure 5 of the revised manuscript does not show any evidence of picocavity events. What the authors stress as "picocavity-like" seems rather random. Picocavity events are evidenced by enhanced vibrational lines that appear sporadically, as shown e.g. in Figure 3b in ref. 54 of the revised manuscript. As explained in this reference, "Variations in the adatom position produce a wandering of picocavity SERS emission, alongside completely stable nanocavity lines". From the recording, I can see a number of lines, but I cannot find the stable lines of the nanocavity, making this recording difficult to interpret. Nevertheless, this could be an exciting research direction to explore with nanodimers, which I look forward to reading about.*

Our Reply: We are very grateful to Reviewer #3 for pointing out this exciting research direction to explore the picocavity behaviors with nanodimers. Although we have given a preliminary study of the coupling of picocavities in such hybrid AuNDs, it seems somewhat insufficient for a systemic investigation. Therefore, we have removed some discussion about the "picocavity " in the revised *Discussion* section of our current manuscript. Still, we retained

the statement, "...this interesting phenomenon indicates the possible formation of a picocavity^{57,58} in this hybrid AuND/CB[7]@single PF system". In the following, we try to provide our considerations in this aspect.

Related to the Reviewer's question: "*From the recording, I can see a number of lines, but I cannot find the stable lines of the nanocavity, making this recording difficult to interpret.*" In fact, one can see there are some stable SERS lines of the single PF molecule induced by nanocavity enhancement (see green arrows in the revised Fig. 5b in the current manuscript), even though these lines are not as stable as those shown in Figure 3b in the reference of *Nano Lett.* 22, 5859–5865 (2022). We argue that this difference mainly arises from the different hybrid systems used in these two works. Specifically, in our hybrid AuND with a single MB molecule, the signature of single-molecule behavior—"spectrum diffusion" is more significant, making the SERS lines induced by the nanocavity look somewhat unstable.

The revised Figure 5b. Time-dependent evolution of SERS signals from a hybrid AuND/CB[7]@single PF. The insets show SEM images of the corresponding measured samples. The scale bar is 50 nm. Each spectrum was collected for 2s, and the samples were excited at 632 nm. New vibrational lines between the blue arrows indicate the possible formation of a picocavity in the hybrid AuND. The green arrows indicate the nanocavity SERS lines.

Of course, the possibility of observing the picocavity behavior in such AuND/single-molecule strong coupling system may be lower than that in the hybrid NPoM system (with many molecules) used by Prof. Baumberg's group (see the following Fig. R13). In *Nano Lett.* 22, 5859–5865 (2022), we cannot clearly see how their NPoM/molecule system was prepared, here we can refer their another recent work that systematically investigated the picocavity behaviors using the same NPoM configuration and the same BPT molecules in the gap (*Nat.*

Commun. 14, 3291 (2023)). In their preparation section, they said that the self-assembled monolayer (SAM) of active molecules "is prepared on the Au surface by immersion in a 1mM analyte solution in anhydrous ethanol (>99.5%) for 22 h. The nanoparticles are dropcast onto the SAM for 30s before being rinsed with deionised water. The short time used for dropcasting ensures a low density across the sample so they can be individually observed in optical microscopy." Therefore, there were "a few hundred molecules in the ~10nm-wide optical field, highly-confined within the nanogap" of the NPoM cavity. When the focused laser spot (~1 μm) illuminates the 80-nm AuNP, there were "a few hundred molecules" in the gap contributed to the SERS emission, resulting in "completely stable nanocavity lines". **Therefore, in such an NPoM/molecule system, the stable SERS lines mainly come from many molecules in the nanocavity.** In contrast, the picocavity SERS emission comes from a single molecule attached to the gold adatom pulled from the facet into the gap (Fig. R13c), making the "variations in the adatom position produce a wandering of picocavity SERS emission, alongside completely stable nanocavity lines".

Figure R13. Schematic of the NPoM/multiple molecules system used in *Nat. Commun.* 14, 3291 (2023). These figures are taken from the reference of *Nat. Commun.* 14, 3291 (2023).

In our AuND/single molecule system, only a single molecule (or a very few molecules) in the gap of the single AuND can contribute to the SERS signals. Therefore, the SERS lines of the dye molecule induced by the nanocavity demonstrate a clearer single-molecule behavior, such as the significant "spectral diffusion" of vibrational peaks, leading to relatively less stable SERS lines of the single dye molecule in the gap. This indicates that the AuND/single molecule coupling system may be an ideal test bed for exploiting single-molecule SERS and single-molecule picocavity behaviors. It should also be mentioned that such unstable lines of the nanocavity can also be seen in Fig. S19 in ref. 7 in our manuscript shows the spectral diffusion of single-molecule SERS from different NPoM.

FIGURE REDACTED

Figure S19 in ref. 7. Spectral diffusion of single-molecule SERRS from different NPoM. This figure is taken from *Nature* 535, 127–130 (2016).

Nevertheless, we plan to conduct further systematic investigations in this research direction using our hybrid AuND with single molecules in the future, and we hope and expect to achieve better results.

Corresponding Revision: We have added green arrows to indicate the nanocavity SERS lines in the revised Fig. 5b in current manuscript.

REVIEWERS' COMMENTS

Reviewer #3 (Remarks to the Author):

In this new version, the authors have revised the manuscript with additional data, proper rephrasing regarding some claims, and statements made stronger with additional bibliography. I am particularly grateful and impressed by the additional measurements, namely:

- the repeated photobleaching experiment (revised Fig. S13), that seemed to me much better and convincing than the previous one.
- the polarization angle-resolved dark field scattering (revised Fig. S18) that, in my opinion, further strengthen the impact of this work.

Concerning the alignment and positioning of the molecule's dipole, I am fully aware of the difficulties encountered by the community in precisely knowing these parameters in the experiments, and I acknowledge the elements raised by the authors on this topic, which, personally, I find very interesting.

I want to underline, once again, the exciting prospect of picocavities interacting at the single molecule level in this experiment, Fig. 5. The authors have satisfactorily answered my comment related to this figure, and I look forward to further research in this direction.

Overall, the authors have brilliantly addressed my comments, and I want to congratulate them for this beautiful work. I do believe the manuscript is now much better, and I recommend publication in Nature Communications with no more delay.

Benjamin Rousseaux